



# Sesquiterpenes identified as key species for atmospheric chemistry in boreal forest by terpenoid and OVOC measurements

Heidi Hellén[1], Arnaud P. Praplan[1], Toni Tykkä[1], Ilona Ylivinkka[2], Ville Vakkari[1], Jaana Bäck[3], Tuukka Petäjä[2], Markku Kulmala[2] and Hannele Hakola[1]

[1]Finnish Meteorological Institute, P.O. Box 503, 0011 Helsinki, Finland

[2]Institute for Atmospheric and Earth System Research (INAR) / Physics, Faculty of Science, University of Helsinki, Finland, P.O. Box 64, FI-00014 University of Helsinki, Finland

[3]Institute for Atmospheric and Earth System Research (INAR) / Forest Sciences, Faculty of Agriculture and Forestry, University of Helsinki, Finland, P.O. Box 27, FI-00014 University of Helsinki, Finland

*Correspondence to*: Heidi Hellén (heidi.hellen@fmi.fi)

**Abstract.** Concentrations of terpenoids (isoprene, monoterpenes, sesquiterpenes) and oxygenated volatile organic compounds (OVOCs, i.e. aldehydes, alcohols, acetates and volatile organic acids) were investigated during two years at a boreal forest site in Hyytiälä, Finland, using in situ gas chromatograph-mass spectrometers (GC-MS). Seasonal and diurnal variations of terpenoid and OVOC concentrations as well as their relationship with meteorological factors were studied.

Of the studied VOCs, $C_2$-$C_7$ unbranched volatile organic acids (VOAs) were found to have the highest concentrations mainly due to their low reactivity. Of the terpenoids, monoterpenes (MTs) had highest concentrations at the site, but also 7 different highly reactive sesquiterpenes (SQTs) were detected. Monthly and daily mean concentrations of most terpenoids, aldehydes and VOAs were found to be highly dependent on the temperature. Highest exponential correlation with temperature was found for a SQT (β-caryophyllene) in summer. The diurnal variations of the concentrations could be explained by sources, sinks and vertical mixing. Especially the diurnal variations of MT concentrations were strongly affected by vertical mixing. Based on the temperature correlations and mixing layer height simple proxies were developed for estimating MT and SQT concentrations.

To estimate the importance of different compound groups and compounds for the local atmospheric chemistry, reactivity with main oxidants (OH, $NO_3$ and $O_3$) and production rates of oxidation products (OxPR) were calculated. MTs dominated OH and $NO_3$ radical chemistry, but SQTs had a major impact on ozone chemistry, even though concentrations of SQT were 30 times lower than MT concentrations. SQTs were the most important also for the production of oxidation products. Since SQTs have high secondary organic aerosol (SOA) yields, results clearly indicate the importance of SQTs for local SOA production.



## 1. Introduction

The boreal forest is one of the largest terrestrial biome in the world, forming an almost continuous belt around the northern
hemisphere. It is characterized by large VOC emissions with strong seasonal variations (Sindelarova et al. 2014). The boreal
zone is estimated to be a major source of climate-relevant biogenic aerosol particles produced from the reaction products of
primary emitted biogenic volatile organic compounds (BVOCs, Tunved et al., 2006). Isoprene, monoterpenes (MTs) and
sesquiterpenes (SQTs) are the main BVOCs emitted from the boreal forest. They are known to influence particle formation
and growth (e.g. Kulmala et al., 2013), the oxidation capacity of air (Peräkylä et al., 2014) and chemical communication by
plants and insects (Holopainen, 2004). Oxidized VOCs emitted from the vegetation include e.g. carbonyls, alcohols and
volatile organic acids (VOAs), but their emissions are less studied and they are also produced in the air from the reactions of
VOCs. Studies on total reactivity in the boreal forest air have suggested the presence of highly reactive unmeasured BVOCs
(Sinha et al. 2010, Nölscher et al. 2012, Praplan et al. 2018). Also in other vegetation zones fraction of unmeasured BVOCs
have been very high (up to 80%; Yang et al., 2016). Therefore better characterization of BVOC emissions and concentrations
in forested areas is needed.

In the air BVOCs readily react with atmospheric oxidants, and the photochemical oxidation of even small organic compounds
can lead to the formation of tens to hundreds of first generation products, which then undergo further oxidation and
transformation (Glasius and Goldstein, 2016). Thus, it will probably never be possible to identify all oxidation products of all
VOCs in the atmosphere. Therefore, detailed knowledge on the primary emitted compounds is crucial. Reaction rates, reactions
and SOA yields of different terpenoids vary a lot (Lee et al. 2006, Ng et al. 2017) and compound specific concentration data
are essential to our understanding on biosphere-atmosphere interactions as well as local and regional atmospheric chemistry.
Proton transfer reaction mass spectrometers (PTR-MSs) are often used for measurements of fluxes or concentrations of MTs
(Yuan et al. 2017) and there are already long data sets on ambient air concentrations of MTs measured by PTR-MSs even in
boreal forest (Lappalainen et al. 2009, Kontkanen et al. 2016). PTR-MS measurements of SQT concentrations are not common,
but there are some data available from the tropical forests (Kim et al. 2009 and 2010, Jardine et al. 2011). However, PTR-MS
is not able to separate different MTs or SQTs. Data on concentrations of individual MTs measured by gas chromatograph-
mass spectrometers (GC-MSs) are scarce and often only from short measurement campaigns (Yanez-Serrano et al. 2017,
Jardine et al. 2015, Jones et al. 2011 , Hakola et al. 2009 and 2003, Kesselmeier et al. 2002). Emissions of both MTs and SQTs
have been studied at various vegetation zones (Guenther et al. 2012), but to our knowledge there are only three studies
published on atmospheric concentrations of individual SQTs (Bouvier-Brown et al. 2009, Hakola et al., 2012 and Yee et al.
2018). Bouvier-Brown et al. (2009) measured ambient air concentrations of MTs and SQTs in a ponderosa pine forest in
California from 20[th] of Aug until 10[th] of Oct in 2007 with an in situ GC-MS and Hakola et al. (2012) measured MTs and SQTs
in the air of a Finnish boreal forest in 2011. However, due to losses in the inlets and less sensitive instrument, both were



missing β-caryophyllene, which is the main SQT known to be emitted by the pines (Hakola et al. 2006). Yee at al. (2018) conducted SQT measurements in the central Amazonia over four months in 2014 and found 30 different SQTs. However, their site was located 2.5 km away from the forest and they concluded that the most reactive compounds had already reacted away before arriving to the site. For example they did not detect β-caryophyllene even though they were able to find lots of its

5 reaction products. Emission chamber measurements are often suffering from losses of the most reactive SQTs on the chamber walls and inlet lines and canopy scale flux measurements are not available due to fast reactions and low concentrations of SQTs. Therefore, ambient concentration data is clearly needed to constrain the emissions of SQTs (Duhl et al. 2008). To our knowledge, we are here reporting the first quantitative measurements of ambient concentrations of β-caryophyllene, the main SQT emitted by the boreal trees (Hakola et al. 2006, Hakola et al. 2001).

Regarding oxygenated volatile organic compounds (OVOCs), studies on the emissions of small compounds (e.g. methanol, acetone, acetaldehyde, acetic acid) have been conducted (e.g. Aalto et al. 2004, Sindelarova et al., 2014), but knowledge of the biogenic sources and concentrations of the larger volatile carbonyls, alcohols ($C_5$-$C_{10}$) and volatile organic acids (VOAs) is very limited.

In this study ambient air measurements of individual BVOCs and OVOCs were conducted in 2015 and 2016 in boreal forest at SMEAR II site in Hyytiälä with in situ gas chromatograph-mass spectrometers (GC-MSs). VOC measurements are susceptible to technical failures and even with the intensive campaign for this study, our data does not cover the whole measuring period continuously. To be able to parametrize the concentrations, to understand their sources and to fill the gaps

20 in the data, we studied the dependence of concentrations on environmental factors. As temperature is the dominant factor controlling the emissions of these BVOCs from trees at Hyytiälä (e.g. Tarvainen et al. 2005, Hakola et al. 2006 and 2017), the main focus was set on temperature dependence. Based on the temperature correlations simplified proxies for estimating local concentrations were developed. To estimate the importance of the individual VOCs or VOC groups for the local atmospheric chemistry and secondary organic aerosol (SOA) production, reactivities and production rates of oxidation products were

25 calculated.

2. **Experimental**

2.1 **Measurement site**

Measurements were conducted in a boreal forest at SMEAR II (Station for Measuring Ecosystem-Atmosphere Relationships) in southern Finland. The SMEAR II station is a dedicated facility for studying the forest ecosystem-atmosphere relations (Hari and Kulmala 2005). The measurement station is located in Hyytiälä (61°51'N, 24°17'E, 181 m a.s.l) in a circa 55-year-old managed conifer forest. The continuous measurements at the site include leaf, stand and ecosystem-scale measurements of



greenhouse gases, pollutants (e.g. $O_3$, $SO_2$, $NO_x$) and many different aerosol, vegetation and soil properties. In addition, a full suite of meteorological measurements is currently collected.

The nearest vegetation to the measurement container is a homogeneous Scots pine forest (*Pinus sylvestris*, >60 %) where some
birches (*Betula* sp.), aspen (*Populus* sp.) and Norway spruces (*Picea abies*) grow below the canopy. The canopy height is about ≈20 m with an average tree density of 1370 stems (diameter at breast height > 5 cm) per hectare (Ilvesniemi et al., 2009). The understorey vegetation is formed by the different shrubs, grasses and moss species. The most common shrubs are lingonberry (*Vaccinium vitis-idea* L.) and bilberry (*Vaccinium myrtillus* L.), the most common mosses are Schreber's big red stem moss (*Pleurozium schreberi* (Brid.) Mitt.) and a dicranium moss (*Dicranum* Hedw. sp.) and the most common grasses
are wavy hair-grass (*Deschampsia flexuosa*) and small cow-wheat (*Melampyrum sylvaticum*). Anthropogenic influence at the site is low. The largest nearby city is Tampere with 200 000 inhabitants. It is located 60 km to the south-west of the site.

### 2.2 VOC measurements

Ambient air measurements of VOCs were conducted in 2011, 2015 and 2016. Data from the year 2011 have been published already earlier by Hakola et al. (2012). Concentrations were measured with three different in situ thermal desorpter-gas chromatograph-mass spectrometers (TD-GC-MSs), described hereafter as GC-MS1, GC-MS2 and GC-MS3. In 2015 and 2016 two different GC-MSs were used in parallel. The instruments were located in a container about 4 metres outside the forest in a gravel-bedded clearing. In 2015 and 2016 samples for GC-MSs were taken at the height of 1.5 m from an inlet reaching out
circa 30 cm from the container wall. In 2011 the GC-MS inlet was on the roof of a container at about 2.5 m.

GC-MS1 was used for the measurements of isoprene and individual monoterpenes in 2011 and May-July 2015. With GC-MS1 air was drawn through a 3m long stainless steel tube (od. ¼ inch) at the flow rate of 1 l min$^{-1}$. Tubes were heated to 120 ºC to avoid losses of terpenoids. The heated inlet system, which also destroys ozone, has been described by Hellén et al. (2012a).
VOCs in a 30-50 ml min$^{-1}$ subsample were collected in the cold trap of thermal desorption unit (ATD-400, PerkinElmer) packed with Tenax TA in 2011 and Tenax TA/Carbopack B in 2015 and analyzed in situ with a gas chromatograph (HP 5890, Agilent Technologies) with DB-1 column (60m, id. 0.25 mm, ft. 0.25 μm) and a mass selective detector (HP 5972, Agilent Technologies). One 60-min sample was collected every other hour. Detection limits were below 1 ppt for all MTs. Measurements with GC-MS1 have been described in detail by Hakola et al. (2012). The system was calibrated using liquid
standards in methanol solutions injected on Tenax TA-Carbopack B adsorbent tubes and analysed between the samples using the offline mode of the instrument. The stability of the mass spectrometer was followed by using tetrachloromethane as an"internal standard". The local concentration of tetrachloromethane in ambient air is stable, and thus it was possible to detect sampling errors or shifts in calibration levels by following its concentration.





GC-MS2 was used for the measurements of $C_5$-$C_8$ alcohols, $C_2$-$C_7$ volatile organic acids (VOAs) and MT sum in May-October 2015 and February-September 2016. Samples were taken every other hour. In the 3 m long fluorinated ethylene propylene (FEP) inlet (1/8 inch I.D.) and an extra flow of 2.2 L min$^{-1}$ was used to avoid losses of the compounds on the walls of the inlet tube. Samples were collected directly from this ambient air flow into the cold trap (U-T17O3P-2S, Markes International LTD,

Llantrisant, UK) of the thermal desorption unit (Unity 2 + Air Server 2, Markes International LTD, Llantrisant, UK). The sampling time was 60 min and the sampling flow through the cold trap 30 ml min$^{-1}$. Samples were analyzed in situ with a gas chromatograph (Agilent 7890A, Agilent Technologies, Santa Clara, CA, USA) and a mass spectrometer (Agilent 5975C, Agilent Technologies, Santa Clara, CA, USA) connected to the thermal desorption unit. The polyethylene glycol column used for separation was the 30-m DB-WAXetr (J&W 122-7332, Agilent Technologies, Santa Clara, CA, USA) with an inner

diameter of 0.25 mm and a film thickness of 0.25 µm. The system was calibrated using liquid standards in Milli-Q water (VOAs) and methanol (other VOCs) injected into adsorbent tubes filled with Tenax TA (60/80 mesh, Supelco, Bellefonte, USA) and Carbopack B (60=80 mesh, Supelco, Bellefonte, USA) and analysed by the same method samples. The stability of the mass spectrometer was followed by running gaseous field standards containing aldehydes and aromatic hydrocarbons after every 50th sample taken and using tetrachloromethane as an "internal standard". Used method has been described in more

detail by Hellén et al. (2017). Due to observed inter-conversion between MT isomers inside the instrument presumably during the pre-concentration step in the thermal desorption unit, only MT sum is reported. Similar behavior has been observed by Jones et al. (2011). However, contrary to their observations we did not find isomerization to be repeatable. They also mention that they were able to detect β-pinene in standards, but it was not detected in the ambient samples. In our tests β-pinene was suffering by the most degradation of all studied MTs and higher inter-conversion was observed after running several ambient

samples than directly after running adsorbent tube standards. With two other GCMSs, which we used, inter-conversion/degradation was not detected.

GC-MS3 was used for the measurements of individual MTs, SQTs, isoprene, 2-methyl-3-butenol (MBO) and $C_5$-$C_{10}$ aldehydes in April-November 2016. With GC-MS3 air was drawn through a 1m long fluorinated ethylene propylene (FEP) inlet (1/8 inch

I.D.) and 1 m long stainless steel tube (od. ¼ inch) at the flow rate of 1 l min$^{-1}$. Stainless steel tubes was used to destroy ozone and was heated to 120 ºC to avoid losses of terpenes (Hellén et al. 2012a). VOCs in a 40 ml min$^{-1}$ subsamples were collected for 30 minutes in the cold trap (Tenax TA/carbopack B) of thermal desorption unit (TurboMatrix, 650, Perkin-Elmer) connected to a GC (Clarus 680, Perkin-Elmer) coupled to a mass spectrometer (Clarus SQ 8 T, Perkin-Elmer). HP-5 column (60 m, id. 0.25 mm, film thickness 1 µm) was used for the separation. The system was calibrated using liquid standards in

methanol solutions injected on Tenax TA-Carbopack B adsorbent tubes and analysed between the samples using the offline mode of the instrument. The stability of the mass spectrometer was followed by running one adsorbent tube standard after every 50th sample taken and using tetrachloromethane as an "internal standard". Unknown SQTs were calibrated using the responses of β-caryophyllene.




Of the used instruments GC-MS3 was the most sensitive and it was able to detect very low concentrations of SQTs, much more than GC-MS1, and therefore in SQT data analysis we used only the 2016 data. Monthly means were calculated from all available data for each month (see number of data points in Table 1 and 2). Daily means were calculated for days with no missing data points starting at 8:00 (UTC+2) and ending at 8:00 (UTC+2) next day.

### 2.3 Calculation of formation rates of measured reaction products of MTs

For studying diurnal variation of measured reaction products of MTs net formation rates (NRF) were calculated. Production rate (PR), destruction rate (DR) and NFR of reactions products of MTs can be described by the equations below:

$$PR = \frac{d[product]}{dt} = k_{OH+MT}[MT][OH] \times yield + k_{O_3+MT}[MT][O_3] \times yield \qquad (1)$$

$$DR = \frac{-d[product]}{dt} = -K_{OH+product}[product][OH] - k_{O_3+product}[product][O_3] \qquad (2)$$

$NFR = PR + DR$ (3)

where $k_x$ is reaction rate with oxidant (x=OH or $O_3$), [MT, product or x] is the concentration of corresponding MT, product or oxidant and yields are yields of the product from the corresponding reactions obtained from Hakola et al. (1994).

**2.4  Reactivity calculations**

The total reactivity of the VOCs ($R_x$) was calculated by combining their respective concentrations (VOC$_i$) with the corresponding reaction rate coefficients ($k_{i,x}$).

$R_x = \sum[VOC_i] \, k_{i,x}$ (4)

This determines in an approximate manner relative role of compounds or compound classes in local OH, $NO_3$ and $O_3$ chemistry. The reaction rate coefficients for OH reactions are the same as used by Praplan et al. (2018), for $O_3$ reactions by Hakola et al. (2017) and $NO_3$ reactions by Ng et al. (2017). When experimental reaction rate coefficients were not available, they were
estimated with the AopWin™ module of the EPI™ software suite (https://www.epa.gov/tsca-screening-tools/epi-suitetm-estimation-program-interface, EPA, U.S.A). For the unknown SQTs an average reaction rate coefficients ($k_{OH}$=1.55 x 10$^{-10}$ cm$^3$ s$^{-1}$, $k_{O3}$=6.1 x 10$^{-15}$ cm$^3$ s$^{-1}$, $k_{NO3}$=8.72 x 10$^{-11}$ cm$^3$ s$^{-1}$) of the known SQTs were used. Due to lack of measured or estimated



reaction rate coefficients, these average values were used also for longicyclene in $O_3$ and $NO_3$ reactions and for β-farnesene in $NO_3$ reactions.

### 2.5 Calculation of total production rates of oxidation products

Production rates for oxidation products (oxPRs) were calculated for the reactions of different VOCs with the OH radicals, $O_3$ and $NO_3$ radicals by the Eq. (6).

$$OxPR = \frac{d[products]}{dt} = \sum [VOCi] \left( k_{OH+VOCi}[OH] + k_{O_3+VOCi}[O_3] + k_{NO_3+VOCi}[NO_3] \right) \tag{5}$$

where $k_i$ is reaction rate coefficient of a VOC with an oxidant (OH, $O_3$ or $NO_3$) and $[VOC_i]$ is the concentration of corresponding VOC or oxidant. Unknown SQTs were not taken into account in calculations. Of the sesquiterpenes reaction rate with $NO_3$ radicals was found only for β-caryophyllene and reactions of other SQTs were not considered in the calculations. Since hydroxyl radical concentrations were not measured directly, proxies were calculated from UVB radiation intensity, which is known to correlate strongly with OH radicals as described by Petäjä et al. (2009). $NO_3$ concentrations were calculated by assuming steady-state by its production from $O_3$ and $NO_2$ and removal by photolysis and oxidation reactions as decribed by Peräkylä et al. (2014). The only modification compared to Peräkylä et al. (2014) was that data for individual MTs were used and β-caryophyllene (main SQT at the site) was also considered as an additional sink.

20 Aerosol surface area needed for the calculation of $NO_3$ concentration was derived from the aerosol number size distribution in the range 3-1000 nm at the SMEAR II. It was obtained using two parallel differential mobility particle sizers (DMPS) (Aalto et al., 2001). Each DMPS system consists of Hauke-type differential mobility analyzer (DMA) and condensation particle counter (CPC). Each DMA separates sampled aerosol particles according to their electrical mobility, and selected particles are transported to the corresponding CPC, which grows them by condensing butanol on their surface, and counts their number with optical methods. By changing the strength of the electric field inside DMA, particles with different electrical mobilities can be selected and counted. The first DMPS measures particles with sizes between 3 and 10 nm and the second one between 10 and 1000 nm. Combining the spectra, the number size distribution of the whole size range is reached. One measurement cycle scanning all the sizes takes about 10 minutes. Charging the aerosol population to an equilibrium charge distribution with a bipolar charger enables the measurements of both neutral and charged particles.

The number of particles in a unit volume in certain size range can be determined as

$$N(D_p) = \int_{D_p}^{D_p+dD_p} n\left(\log D_p\right) d\log D_p \tag{6}$$





where $n(\log D_p)$ is the number density distribution representing the number of particles between diameter $D_p$ and $D_p + \mathrm{d}\,D_p$ per unit volume. When assuming that all the particles are spherical, the surface area distribution become

$$s\left(log D_p\right) = n\left(log D_p\right) \cdot \pi D_p^2 \tag{7}$$

Then the surface area of particles in the range $D_p - D_p + \mathrm{d}\,D_p$ obtained likewisely as

$$S\left(D_p\right) = \int_{D_p}^{D_p + dD_p} s\left(log D_p\right) dlog D_p \tag{8}$$

## 2.6 Complementary measurements

Meteorological data, $O_3$, NO and $NO_x$ concentrations were obtained from SmartSMEAR AVAA-portal (Junninen et al., 2009,
https://avaa.tdata.fi/web/smart). All the data used in this study are the one collected at the height of 4.2m from the mast inside the forest, except for the temperature at 125m for comparison.

Mixing layer height (MLH) was estimated from measurements with a Halo Photonics Stream Line scanning Doppler lidar, which is a 1.5 μm pulsed Doppler lidar with a heterodyne detector (Pearson et al., 2009). Range resolution of the lidar is 30 m and the minimum range of the instrument is 90 m. Operating specifications of the lidar are given in supplementary Table S1. At Hyytiälä wind profile was obtained from a 30° elevation angle conical scan, i.e. from a vertical azimuth display (VAD) scan. This VAD scan was configured with 23 azimuthal directions and integration time of 12 s per beam. A vertical stare of 12 beams and integration time of 40 s per beam was configured to follow the VAD scan. The VAD scan and 12-beam vertical stare were scheduled every 30 min at Hyytiälä; other scan types operated during the 30-min measurement cycle were not utilized in this study. The lidar data was corrected for a background noise artefact according to Manninen et al. (2016). After this correction a signal-to-noise-ratio threshold of 0.001 was applied to the data.

Turbulent kinetic energy (TKE) dissipation rate was calculated from the Doppler lidar measurements according to the method by O'Connor et al. (2010). A VAD-based proxy for turbulent mixing ($\sigma^2_{VAD}$) was calculated from the 30° elevation VAD scan according to the method by Vakkari et al. (2015). MLH was determined from the TKE dissipation rate and the VAD scan similar to Vakkari et al. (2015). Briefly, first a constant threshold of $10^{-4}$ $m^2$ $s^{-3}$ was applied to the TKE dissipation rate profile, i.e. MLH was taken as the last range gate where TKE dissipation rate was higher than $10^{-4}$ $m^2$ $s^{-3}$. If the TKE dissipation rate was below the threshold value at the first usable gate at 105 m above ground level (a.g.l.), i.e. MLH < 105 m, the $\sigma^2_{VAD}$ profile





was used to identify MLH. For $\sigma^2_{VAD}$ a constant threshold of 0.05 m$^2$ s$^{-2}$ was applied to determine MLH (Vakkari et al., 2015). With this approach MLH could be identified from 60 m a.g.l. to >2000 m a.g.l.; rainy periods were excluded from analysis. Values below 60 m were marked as 0 m.

**3.    Results and discussion**

**3.1 Seasonal and diurnal variations of concentrations**

The concentrations of most compounds measured with three different GC-MS instruments in 2011, 2015 and 2016 are at the
same level (Table 1 and 2). Of the measured compounds VOAs had highest concentrations during all months (Figure 1). Also 1-butanol and isopropanol concentrations were high, most likely because they are used in some instruments for aerosol measurements at the site. Even though concentrations of terpenoids are not as high as VOAs due to their high reactivity, they are expected to have highest impacts on local chemistry. For most of the studied compounds daily and monthly mean concentrations were highest during the warm summer months. For aromatic hydrocarbons, which are mainly emitted from
anthropogenic sources, concentrations were higher in winter.

Relative diurnal variation of most compounds was highest in June when mixing layer heights were highest (Figure 2 and 3). Concentrations of different compounds and compound classes are described in more detail in following sections.

**3.1.1 Concentrations of monoterpenes**

Of the terpenoids MTs had the highest concentrations with the mean MT SUM being 400, 440 and 430 pptv in summers (Jul-Aug) 2011, 2015 and 2016, respectively (Table 1). All MTs except p-cymene had a clear maximum in summer. MT concentrations have been measured earlier at this boreal forest site also with PTR-MSs (Lappalainen et al. 2009 and Kontkanen
et al. 2016). Those PTR-MS measurements were conducted close to the forest canopy at the height of 14 m (2006–2009) or 16.8 m (2010–2013). The median July MT concentration measured between 2006 and 2013 was 382 pptv. Large spatial differences in concentrations especially for terpenoids are expected depending on the sampling point at the site (Liebmann et al. 2018). In our study the sampling site was at the edge of the forest whereas in the previous studies by Lappalainen et al. (2009) and Kontkanen et al. (2016) above the canopy in the middle of the stand.

In our measurements MT concentrations had high peaks in May 2016 (12th at 3:04am and 14th at 4:10am-6:10am), 1st of June at 1:10am and 9th of September at 23:35pm, which were clearly deviating from the other data. Based on the wind directions, it is possible that these peaks are due to high concetrations coming from the site of operations of a sawmill, a woodmill and a pellet factory in Koreakoski, 5 km southeast of Hyytiälä. The influence of this factory on monoterpene concentrations have





been observed also in earlier studies of Eerdekens et al. (2009), Liao et al. (2011), Williams et al. (2011) and Hakola et al. (2012). These samples were not used in the further analysis.

Of the MTs α-pinene had clearly the highest concentrations (50% of the MT sum) followed by 3Δ-carene, β-pinene and
limonene (Figure 1 and Table 1). MT distribution was very similar for nighttime (PAR<50 µmol s$^{-1}$ m$^{-2}$) and daytime (PAR>50 µmol s$^{-1}$ m$^{-2}$) values, only 1,8-cineol and linalool had a bit higher fraction during the day. MT distribution observed here resembles the one of the emissions of local trees (Bäck et al. 2012, Hakola et al. 2006 and 2017).

Diurnal variability of MT concentrations at the site is driven by the vertical mixing; low values were measured during the day
when mixing was highest and the highest values during nights with lowest mixing (Figures 2 and 3). This has been observed also in earlier studies of MTs at this boreal forest site (Kontkanen et al. 2016 and Hakola et al. 2012). Similar diurnal variation was found by Bouvier-Brown et al. (2009) in a ponderosa pine forest in the Sierra Nevada Mountains of California. However, this observation of MTs is opposite to the diurnal variation of MT concentrations measured in the Amazon tropical rain forest by Yanez-Serrano et al. (2017). Light dependent emission found in the Amazonia (Jardine et al. 2015) could explain this. In
boreal forest emissions are strongly temperature dependent and may continue also during nights only with lower rates if temperature is high enough (e.g. Hakola et al. 2006).

Diurnal variation of concentrations was highest in June concomitant with highest variation of mixing layer height (Figure 2 and 3). Mean mixing layer heights during the day (at 12:00-16:00) in June and July were 1605 and 819 m, respectively. While
during the night (at 00:00-4:00) mean mixing heights in June and July were 87 and <60 m, respectively. Also monthly mean mixing layer height, which roughly describes the mean dilution volume of the emissions, was two times higher during our measurements in June than in July (Table1). Since the lidar that was used for the measurements of mixing layer heights, is not able to detect mixing layer heights <60 m, we used also temperature difference between the heights 125 m and 4.2 m to roughly describe the vertical mixing. The correlation of monthly mean diurnal variation of MT sum concentrations with temperature
difference at the site was high ($R^2_{MT}$=0.85 in July, Figure S1a). Also individually measured values had relatively good correlation with temperature difference ($R^2_{MT}$=0.46 in July, Figure S1b). 1,8-cineol and linalool were not following this general diurnal pattern of MTs giving indication of different sources. 1,8-cineol is the only MT which is known to have also clear light dependent emissions from Scots pines growing at the site (Hakola et al. 2006).

**3.1.2 Concentrations of sesquiterpenes**

At the moment there is very little data on atmospheric SQT concentrations and also emission data is much sparser than for MTs. In our measurements SQTs had similar seasonal variation as MTs, but their concentrations were much lower (Table 1). SQTs are very reactive and therefore their contribution to the local chemistry can still be significant. The highest 30-minute



mean for SQT sum (103 pptv) was detected on 25th of July at 3:15am coinciding with high temperature and shallow mixing layer. SQTs were not increased during the sawmill episodes of MTs. They are more reactive and if emitted they are probably depleted during the transport from the sawmill to the site.

Of the SQTs β-caryophyllene had the highest concentrations, but in summer also longicyclene, β -farnesene and 4 unidentified SQTs were detected (Figure 1). Night time (PAR<50 μmol s$^{-1}$ m$^{-2}$) and daytime (PAR>50 μmol s$^{-1}$ m$^{-2}$) distributions of SQT concentrations were very similar, only β-farnesene had a slightly higher fraction during the day. β-caryophyllene is emitted by the local pines and spruces (Hakola et al. 2006 and 2017) as well as from the forest floor (Hellén et al. 2006, Mäki et al. 2017). Aaltonen et al. (2011) and Mäki et al. (2017) have also detected longicyclene in forest floor emissions. Laboratory studies

have shown stress related emissions of β-farnesene (Blande et al. 2009, Petterson 2007), and it has been detected as well in local spruce emissions by Hakola et al. (2017).

The diurnal variation of most SQTs was similar to the variability of MTs, and the concentrations were largely driven by the vertical mixing (Figure 2 and 3). As for the MTs correlation of monthly mean diurnal variation of SQT concentrations with

temperature difference between the heights 125m and 4.2m was high (R$^2$$_{SQT}$=0.90 in July, Figure S1a) and  also individual measured values had relatively good correlation with temperature difference (R$^2$$_{SQT}$=0.48 in July, Figure S1b). Based on the modeling studies by Zhou et al. (2017) also a higher chemical sink during the day may have an effect on SQT concentrations. This was supported here by the higher relative diurnal variation of SQTs compared to MTs. The only exception was, β-farnesene, which had almost as high concentrations during the day as in the night, indicating different sources than for other

SQTs and MTs. An opposite diurnal variation was found by Bouvier-Brown et al. (2009) at ponderosa pine forest in California and they suggested that β-farnesene is having mainly light dependent sources.

### 3.1.3 Isoprene and 2-methyl-3-buten-2-ol concentrations

Isoprene and 2-methyl-3-buten-2-ol (MBO) concentrations were low (Table 1). Monthly means in 2016 were 0.3-18 pptv for isoprene and 0.1-30 pptv for MBO. Low levels were expected since the main local trees trees (Scots pine and Norway spruce) are MT emitters and have only minor emissions of isoprene and MBO (Tarvainen et al. 2005, Hakola et al. 2006 and 2017). Highest daily means were measured in July and August together with MTs and SQTs. Isoprene is known to have light dependent emissions (Ghirardo et al. 2010) while MBO emissions from local trees are mainly temperature dependent (Hakola

et al. 2006).

The diurnal variation of MBO was coinciding with the variation of MTs with high values during the night and low values during the day. This was expected due to temperature dependent emission of MBO. For isoprene clear change in diurnal variation was observed between early summer (April-June) and late summer (July-September) (Figure 2). In May and June



when emissions are still low due to the early growing season, lower daytime values were detected, but in July and August, daytime concentrations were clearly higher due to high light dependent emissions. Previously the gradually increasing isoprene emissions have been connected to the foliage growth period, and start when the effective temperature sum (ETS) reaches a threshold value , e.g. in tea-leafed willow the lower emissions of isoprene were found when ETS<400 degree days (Hakola et al. 1998). During our measurements in 2016 ETS reached value 400 on 23rd of June.

### 3.1.4 Concentrations of reaction products of terpenes

Methacrolein (MACR) is a reaction product of isoprene, but monthly means of MACR concentrations are not following the concentrations of its precursor isoprene (Table 1). MACR is known to have also anthropogenic sources (Biesenthal and Shepson, 1997) and in spring and autumn, when lifetimes are longer and biogenic emissions lower than in summer, anthropogenic influence is expected to be higher. In July and August when isoprene concentrations and biogenic emissions are highest monthly mean concentrations of MACR (4.8 and 3.3 ppt, respectively) were ca. 30% of the isoprene concentration. This is close to the yields of 25 % and 24 % measured in chamber experiments by Paulson et al. (1992) and Atkinson (1994), respectively.

Nopinone is a reaction product of β-pinene and its monthly mean concentration followed the variation of the MTs (Table 1). During the summer months concentration was 7-13 % of the concentration of its precursor β-pinene. In reaction chamber studies yields of nopinone in ozone reactions of β-pinene have been 19-23 % (Grosjean et al. 1993; Hakola et al. 1994 and Winterhalter et al. 2000) and in OH radical reactions 25-30 % (Calvert et al. 2011). Further reactions of the products affect also the ambient air concentrations and therefore chamber yields are not directly comparable to the concentrations.

Mean diurnal variation of nopinone was following the variations of its precursor (β-pinene) in April- June, but in July-September high values were observed also during the day (Figure 4). Nopinone is known to be produced both from OH radical and $O_3$ reactions of β-pinene (Hakola et al. 1994), but it is destroyed only in the OH radical reactions. Since the yields from the $NO_3$ radical reactions are not available from literature they cannot be considered. $NO_3$ reactions would increase the production especially during the night. Also deposition may have an effect (Zhou et al. 2017), but it was not taken into account here. Production rate (PR), destruction rate (DR) and net formation rate (NFR) of nopinone were calculated using the equations 1 to 3. Nopinone yields for OH radical and $O_3$ reactions obtained from Hakola et al. (1994) were 0.27 and 0.23, respectively. Change in the nopinone diurnal variation is explained by the balance between its sources and sinks. Concentrations followed closely the NFR variation (Figure 4). Nopinone is rather stable molecule, and has 5 times lower OH radical reactivity than β-pinene and in contrast to β-pinene, it is not reacting with $O_3$. Results indicate that in May and June, when there is already high light intensity and high OH radical concentrations, but the emissions of β-pinene are still low due to lower temperatures and early growing season, produced nopinone reacts away during the day and higher values are measured during the night, when



there are no OH radicals, but nopinone is still produced from O₃ reactions of β-pinene. In July and August higher emissions and faster reactions of β-pinene with OH radicals results in higher daytime concentrations of nopinone. In September emissions are already lower, but also the OH radical concentrations and the mixing layer heights are lower and higher nopinone concentrations are still detected during the day.

Reaction product of limonene, 4-acetyl-1-methylcyclohexene (4-AMCH), had very low concentrations and was detected only in June and July (Table 1). NFR for 4-AMCH was calculated by the same methods as for nopinone using the equations 1 to 3. Reaction rates of 4-AMHC with OH radical and O₃ are only 25 and 20 % lower than for its precursor. 4-AMCH yields used here for OH radical and O₃ reactions obtained from the Hakola et al. (1994) were 0.20 and 0.04, respectively. In a study by

Grosjean et al. (1993) yield from O₃ reaction was 0.02. The measured concentrations in the present study did not follow the diurnal variation of NFR especially in July, when highest concentrations were measured (Figure 4). However, production rate (PR) had aq similar diurnal pattern as concentrations. Studies on limonene reactions by Grosjean et al. 1993 and Hakola et al. 1994 did not take into account that 4-AMCH reacts almost as fast with the oxidants as limonene and real yields could be higher. When we increased yields in our calculations better agreement could be achieved. In figure 4d yields for OH radical

and O₃ reactions were increased by factor of 2 and 3, respectively. In August the sensitivity of the instrument was less than 50 % of the sensitivity in June/July due to a worse tuning of the MS and 4-AMCH was not detected even though calculations would indicate higher concentrations than in June.

### 3.1.5 Concentrations of volatile organic acids

VOAs had higher concentrations than terpenoids (Tables 1 and 2). Their atmospheric lifetimes (Calvert et al. 2011, Hellén et al. 2017) are much longer and therefore they can accumulate in the air and be transported for longer distances. They are expected to have both biogenic and anthropogenic sources and they are also produced in the air by the reactions of other VOCs (Ciccioli and Mannozzi, 2007). In this study the highest concentrations of VOAs in 2016 were measured already in June (Table

2). This was at least partly due to the different measurement periods for GCMS2 and GCMS3 in June and July. Days, when VOAs were measured with GCMS2, were at the end of the June when temperature (18 ºC) and PAR were higher compared to the VOC measurements with GCMS3 in June (temperature 12 ºC). Also the MT sum measured together with VOAs had the highest monthly mean in June.

The daily means of C₃-C₇ VOAs had higher correlation with MT sum (R=0.6-0.85) than with anthropogenic compounds e.g. toluene (R=0.0-0.31) indicating biogenic origin of these compounds either direct or through secondary production in the air. Only acetic acid showed some correlation with aromatic hydrocarbons (R=0.2-0.48). Since the lifetime of acetic acid is longest (1-2 weeks; Calvert et al. 2011) it is expected to be more influenced by the long-range transported anthropogenic emissions.





Daily means of hexanoic acid had very high correlation with 1-hexanol (R=0.97), but also with other $C_6$ compounds often called as green leaf volatiles (GLVs) i.e. hexanal (R=0.82) and cis-3-hexenol (R=0.83). This indicates that also hexanoic acid could be a GLV or that it is produced from GLVs in the air. Correlation of daily means of hexanoic acid with pentanoic and propanoic acids was also high (R=0.89 and 0.80, respectively) and with MT sum relatively high (R=0.77).

For smaller acids (acetic, propanoic and butanoic) daytime maxima were observed especially in July/August, but for pentanoic and hexanoic acids higher concentrations were observed during the nights (Figure 2). The mean diurnal variation of VOAs was not as strong as for the MTs and SQTs. Both direct biogenic emissions and production in the atmosphere are expected to be higher during the day, but since the mixing layer is also higher VOAs are more diluted. However, due to longer lifetimes

10 of these acids, mixing effect during the day is not as strong as for fast reacting terpenes since also dilution air may contain comparable amounts of these acids and they may even be produced in the upper parts of the mixing layer from the reactions of other VOCs. Also losses due to OH reactions during the day and dry/wet deposition during the nights are expected to affect the concentrations (Calvert et al. 2011). In canopy scale flux measurements by PTR-TOF downward fluxes of acetic acid have been detected especially during the nights (Schallhart et al. 2018).

### 3.1.6 Concentrations of $C_5$-$C_{10}$ aldehydes

$C_5$-$C_{10}$ aldehydes can be directly emitted or they can be produced in the air through oxidation of other compounds. Generally, the emissions are much lower than those of smaller aldehydes. Low emissions have been measured e.g. from grassland, but

20 emissions of trans-2-hexenal, and also 2-hexenylacetate and 2-hexenol from wounded and stressed plants can be significant (Fall 1999 and Hakola et al., 2001). Possanzini et al., (2000) found that larger aldehydes (heptanal, octanal) were emitted from citrus plants when exposed to ozone. There is also some evidence that e.g. nonanal can be produced when ozone attacks the fatty acids on leaf or needle surfaces (Bowman et al., 2003). Hakola et al. (2017) measured also $C_4$-$C_{10}$ aldehyde emissions from Norway spruce and found out that they were about the same magnitude as MT emissions during late summer.

Concentrations of $C_5$-$C_{10}$ aldehydes were low; monthly means remained <10 ppt (Table 1). In the measurements of Hellén et al. (2004) at the same site in March and April 2003 concentrations were slightly higher (12-16 pptv) but within the same order of magnitude. Diurnal variation of $C_5$-$C_{10}$ aldehydes were following the variation of isoprene with low daytime values in June and high values in July and August (Figure 2).

Daily means of hexanal were highly correlated with MTs and SQTs (R=0.90) in summer (Jun-Aug). Daily means of nonanal and decanal have highest correlation with β-farnesene in summer. Since β-farnesene emissions are related to stress this could indicate stress related source for them too.



In July 24-hour samples for analysis of carbonyls with a liquid chromatograph (LC, Praplan et al. 2018 in preparation) were collected concomitant with GCMS3 measurements, similarly to Praplan et al. 2017. From those samples also concentrations of <C$_5$ carbonyls were obtained. July mean for formaldehyde, acetaldehyde, acetone and butanal were 430, 270, 1820 and 50 pptv, respectively. Concentrations of these smaller carbonyls are much higher than for C$_5$-C$_{10}$ aldehydes. This is expected due
5  to longer lifetimes and larger emissions (Hellén et al. 2004).

**Table 1:** Mean concentrations of VOCs measured in summer (Jun-Aug) in 2011, 2015 and 2016 (GC-MS1 and GC-MS3) and monthly mean concentrations (pptv) in Apr-Nov 2016 with mean temperature (T), mean photosynthetic radiation (PAR), mean mixing layer height (MLH), mean mixing layer height between 0:00-4:00 (MLH$_{00-04}$) and mean mixing layer height between
10  12:00-16:00 (MLH$_{12-16}$) during the VOC measurements. N=number of measurements and DL=detection limit, '-'=missing value.

| pptv | DL | Summer (Jun-Aug) | | | 2016 | | | | | | | |
|---|---|---|---|---|---|---|---|---|---|---|---|---|
| | | 2011 | 2015 | 2016 | Apr | May | Jun | Jul | Aug | Sep | Oct | Nov |
| N | - | 267 | 244 | 240 | 163 | 244 | 61 | 114 | 125 | 246 | 84 | 187 |
| T (°C) | - | 15.3 | 14.1 | 14.7 | 2.9 | 13.3 | 12.0 | 17.8 | 14.3 | 11.1 | 2.0 | -1.6 |
| PAR (µmols$^{-1}$m$^{-2}$) | - | - | 382 | 389 | 227 | 413 | 492 | 374 | 301 | 229 | 38 | 15 |
| MLH (m a.g.l.) | - | - | - | - | 315 | 485 | 615 | 301 | 294 | 192 | 136 | 120 |
| MLH$_{00-04}$ (m a.g.l.) | - | - | - | - | 62 | 36 | 87 | 22 | 53 | 34 | 95 | 125 |
| MLH$_{12-16}$ (m a.g.l.) | - | - | - | - | 783 | 1262 | 1605 | 819 | 883 | 567 | 222 | 138 |
| Isoprene | 5.1 | 102 | 74 | 11 | 0.3 | 5.2 | 5.0 | 17.7 | 11.0 | 4.6 | 1.5 | 1.4 |
| MBO | 3.5 | - | 9.3 | 14 | 0.1 | 6.3 | 7.7 | 29.8 | 3.7 | 1.9 | 0.8 | 0.2 |
| α-Pinene | 1.1 | 192 | 248 | 224 | 10 | 110 | 136 | 365 | 173 | 72 | 4.8 | 7.7 |
| Camphene | 1.0 | 23 | 20 | 20 | 3.8 | 18 | 17 | 30 | 14 | 15 | 2.6 | 1.9 |
| β-Pinene | 0.2 | 53 | 35 | 37 | 1.0 | 18 | 20 | 60 | 31 | 15 | 0.8 | 0.7 |
| 3Δ-Carene | 0.8 | 85 | 79 | 86 | 3.8 | 51 | 65 | 136 | 58 | 24 | 1.2 | 2.3 |
| p-Cymene | 0.6 | 8 | 18 | 11 | 2.8 | 12.2 | 16.3 | 6.9 | 10 | 10 | 2.1 | 2.2 |
| 1,8-Cineol | 0.9 | 10 | 19 | 9.7 | 0.4 | 6.2 | 8.9 | 14 | 5.8 | 4.5 | 1.5 | 1.3 |
| Limonene | 1.0 | 23 | 14 | 30 | 0.6 | 6.2 | 9.2 | 62 | 21 | 9.7 | 0.4 | 0.7 |
| Terpinolene | 1.2 | 2 | 0.4 | 2.6 | 0.0 | 0.0 | 0.0 | 6.4 | 1.3 | 0.0 | 0.0 | 0.0 |
| Linalool | 1.6 | - | 5.6 | 0.8 | 0.0 | 1.1 | 1.1 | 1.0 | 0.3 | 0.1 | 0.4 | 0.2 |
| Myrcene | 0.5 | - | 4.2 | 5.3 | 0.3 | 1.7 | 2.0 | 10 | 3.8 | 1.4 | 0.2 | 0.2 |
| Bornylacetate | 0.6 | 0.6 | 0.6 | 1.1 | 0.0 | 0.7 | 0.7 | 1.8 | 0.7 | - | 0.2 | 0.1 |
| **MT SUM** | - | 398 | 442 | 427 | 22 | 223 | 274 | 689 | 318 | 151 | 13 | 17 |
| Longicyclene | 0.3 | - | - | 0.2 | 0.0 | 0.3 | 0.1 | 0.4 | 0.1 | 0.1 | 0.0 | 0.0 |
| β-Farnesene | 0.9 | - | - | 1.2 | 0.0 | 0.0 | 0.5 | 2.8 | 0.3 | 0.0 | 0.0 | 0.0 |
| β-Caryophyllene | 0.8 | - | - | 7.8 | 0.0 | 2.6 | 3.2 | 16 | 4.5 | 4.0 | 0.0 | 0.1 |
| SQT1 | 0.4 | - | - | 0.5 | 0.0 | 0.0 | 0.0 | 1.4 | 0.2 | 0.0 | 0.0 | 0.0 |
| SQT2 | 0.4 | - | - | 1.1 | 0.0 | 0.0 | 0.0 | 2.7 | 0.7 | 0.0 | 0.0 | 0.0 |
| SQT3 | 0.6 | - | - | 0.5 | 0.0 | 0.0 | 0.0 | 1.4 | 0.2 | 0.0 | 0.0 | 0.0 |
| SQT4 | 0.7 | - | - | 1.4 | 0.0 | 0.0 | 0.0 | 3.8 | 0.4 | 0.0 | 0.0 | 0.0 |





| | | | | 13 | 0.1 | 2.9 | 3.9 | 28 | 6.4 | 4.1 | 0.09 | 0.06 |
|---|---|---|---|---|---|---|---|---|---|---|---|---|
| SQT SUM | - | - | - | 13 | 0.1 | 2.9 | 3.9 | 28 | 6.4 | 4.1 | 0.09 | 0.06 |
| Nopinone | 0.8 | - | - | 3.8 | 0.7 | 2.1 | 1.6 | 7.7 | 2.0 | 1.2 | 0.5 | 0.5 |
| 4-AMCH | 0.9 | - | - | 0.2 | 0.0 | 0.0 | 0.2 | 1.9 | 0.0 | 0.0 | 0.0 | 0.0 |
| MACR | 0.3 | - | - | 3.8 | 4.1 | 3.8 | 3.4 | 4.8 | 3.3 | 5.0 | - | - |
| Pentanal | 0.9 | - | 41 | 6.8 | 5.2 | 11 | 6.1 | 8.1 | 6.3 | 3.8 | - | - |
| Hexanal | 0.4 | - | 30 | 9.1 | 3.5 | 9.1 | 6.5 | 12 | 8.6 | 3.2 | - | - |
| Octanal | 1.8 | - | 3.2 | 4.8 | 1.5 | 4.9 | 4.2 | 6.2 | 4.0 | 0.3 | - | - |
| Nonanal | 0.8 | - | 27 | 7 | 1.8 | 3.0 | 5.4 | 8.4 | 7.0 | 0.0 | - | - |
| Decanal | 1.8 | - | 19 | 7.1 | 0.4 | 3.0 | 6.8 | 10 | 4.1 | 0.5 | - | - |
| *trans*-2-Hexenal | 1.6 | - | 4.6 | 1.1 | 0.0 | 1.4 | 0.3 | 2.3 | 0.7 | 1.5 | - | - |

**Table 2:** Mean concentrations of studied VOCs measured in summer (Jun-Aug) 2015 and 2016 (GC-MS2) and monthly mean concentrations (pptv) in Feb-Sep 2016 with mean temperature (T) and photosynthetic radiation (PAR). N= number of measurements and DL=detection limit, '-'=missing value.

| pptv | DL | Summer(Jun-Aug) 2015 | Summer(Jun-Aug) 2016 | 2016 Feb | Mar | Apr | May | Jun | Jul | Aug | Sep |
|---|---|---|---|---|---|---|---|---|---|---|---|
| N | - | 615 | 218 | 43 | 56 | 92 | 240 | 31 | 81 | 106 | 218 |
| T (ºC) | - | 14.0 | 16.1 | -5.8 | -3.9 | 5.2 | 12.4 | 18.1 | 17.7 | 14.4 | 10.9 |
| PAR (µmols$^{-1}$m$^{-2}$) | - | 388 | 371 | 85 | 59 | 253 | 448 | 483 | 347 | 368 | 252 |
| MT SUM | - | 324 | 350 | 5.4 | 4.5 | 55 | 129 | 568 | 502 | 171 | 204 |
| Acetic acid | 280 | 1799 | 978 | 1530 | 714 | 1172 | 899 | 1723 | 1395 | 564 | 1418 |
| Propanoic acid | 22 | 127 | 76 | 46 | 26 | 55 | 51 | 225 | 90 | 29 | 87 |
| Butanoic acid | 14 | 45 | 68 | 37 | 14 | 41 | 37 | 114 | 76 | 56 | 74 |
| Pentanoic acid | 5 | 16 | 44 | 16 | 4.0 | 10 | 8 | 88 | 62 | 25 | 32 |
| Hexanoic acid | 7 | 15 | 11 | 3.7 | 1.3 | 1.9 | 0.5 | 35 | 15 | 2.3 | 3.3 |
| Heptanoic acid | 16 | 3.5 | 5 | 1.0 | 0.4 | 0.3 | 0.0 | 26 | 4.4 | 0.0 | 0.2 |
| Isopropanol | 11 | 122 | 228 | 9.4 | 1107 | 50 | 23 | 61 | 124 | 361 | 1097 |
| 1-Butanol | 6 | 138 | 365 | 169 | 48 | 306 | 339 | 508 | 304 | 311 | 277 |
| 1-Pentanol | 19 | 2.6 | 2.9 | 0.0 | 0.0 | 0.7 | 0.6 | 9.3 | 2.9 | 0.0 | 4.9 |
| 1-Hexanol | 3 | 1.5 | 2.2 | 0.0 | 0.0 | 0.2 | 0.2 | 5.2 | 3.1 | 0.1 | 1.0 |
| 1-Penten-3-ol | 2 | 0.5 | 0.8 | 0.0 | 0.0 | 0.0 | 0.1 | 0.5 | 2.0 | 0.0 | 0.1 |
| *trans*-3-Hexen-1-ol | 5 | 0.5 | 1.3 | 0.0 | 0.0 | 0.3 | 0.1 | 0.0 | 3.3 | 0.0 | 0.0 |
| *cis*-3-Hexen-1-ol | 4 | 0.0 | 0.4 | 0.0 | 0.0 | 0.0 | 0.0 | 0.9 | 0.4 | 0.0 | 0.1 |
| *trans*-2-Hexen-1-ol | 13 | 1.3 | 0.0 | 0.0 | 0.0 | 0.0 | 0.0 | 0.0 | 0.0 | 0.0 | 0.0 |
| *cis*-2-Hexen-1-ol | 12 | 0.2 | 0.0 | 0.0 | 0.0 | 0.0 | 0.0 | 0.0 | 0.0 | 0.0 | 0.0 |
| 1-Octen-3-ol | 3 | 0.1 | 0.1 | 0.0 | 0.0 | 0.0 | 0.0 | 0.2 | 0.2 | 0.0 | 0.2 |
| Butylacetate | 39 | 7.9 | 0.6 | 0.0 | 0.0 | 1.5 | 0.0 | 2.2 | 0.2 | 0.1 | 0.6 |



| | 8 | 0.0 | 0.0 | 0.0 | 0.0 | 0.0 | 0.0 | 0.0 | 0.0 | 0.0 | 0.0 |
|---|---|---|---|---|---|---|---|---|---|---|---|
| Hexylacetate | 8 | 0.0 | 0.0 | 0.0 | 0.0 | 0.0 | 0.0 | 0.0 | 0.0 | 0.0 | 0.0 |
| *cis*-3-Hexenylacetate | 6 | 0.8 | 0.2 | 0.0 | 0.0 | 0.0 | 0.0 | 0.5 | 0.3 | 0.0 | 0.0 |
| *trans*-2-Hexenylacetate | 7 | 0.0 | 0.0 | 0.0 | 0.0 | 0.0 | 0.0 | 0.0 | 0.0 | 0.0 | 0.0 |

There has been discussion on the formation of aldehydes in GC and PTR-MS instruments from organic peroxides (Rivera-Rios et al. 2014). However, we measured methacrolein and hexanal with both LC-UV and GCMS3 in July 2016 and the results were at comparable levels (methacrolein 4.7 and 4.8 pptv and hexanal 8.4 and 12 pptv, respectively). For pentanal even higher concentrations were obtained by LC-UV (July mean 45 ppt). In 2015 when GCMS1 was used, aldehyde concentrations were clearly higher than in 2016 (Table 1) and it is possible that the production from organic peroxides in GCMS1 could explain the difference. This indicates that hypothesis by Rivera-Rios et al. (2014) might be true for some GC instruments, but it is still unclear under which circumstances.

### 3.1.7 Concentrations of alcohols and acetates

$C_4$-$C_8$ alcohols and acetates (including GLVs) have generally very low concentrations; monthly means were mostly below detection limits (Table 2). The only exceptions were 1-butanol and isopropanol, which are both used in instrumentation at the site and have therefore higher concentrations from leaks and exhaust lines. As for the other BVOCs the highest concentrations of most alcohols and acetates were measured in summer. Most of the measured alcohols and acetates were GLVs, which are emitted due to herbivory or pathogen infection by almost every green plant (Scala et al. 2013) or due to physical damage of plants (Hakola et al. 2001).

### 3.2 Correlation of concentrations with temperature

Monthly and daily means of most of the studied BVOCs were found to be exponentially correlated with temperature. This temperature dependence is described in more detail for different compound groups as well as for individual BVOCs in the following sections.

### 3.2.1 Correlation of MT concentrations with temperature

Monthly mean MT concentrations had very strong exponential correlation with temperature ($R^2$=0.92, Figure 5a). The site is dominated by Scots pines, which have temperature and light-dependent emissions of MTs (Tarvainen et al. 2005). Correlation of photosynthetic active radiation (PAR) with monthly mean MT concentrations was also high ($R^2_{\text{Apr-Nov 2016}}$=0.73), but clearly lower than with temperature.





Daily means of MTs also correlate well with temperature ($R^2_{\text{Apr-Nov 2016}}$=0.83 and $R^2_{\text{Jun-Aug 2016}}$=0.88, Table 3 and Figure 5b). High exponential correlation of mean concentrations with temperature indicates that temperature has major effect on the seasonality of the concentrations and emissions and processes controlling them. In an earlier study by Lappalainen et al. (2009)

lower correlation ($R^2$=0.50) with temperature was found for PTR-MS data. However, they used only daytime medians. In our study 24-hour averages starting at 8:00 (UTC+2) and ending next day at 8:00 (UTC+2) have been used.

Temperature dependence of monoterpene emissions are often described by the Guenther algorithm (Guenther et al., 1993):

$E=E_S \times \exp(\beta\,(T-T_S))$                     (9)

, where $E_S$ is the standardized emission potential ($\mu g$ $gdw^{-1}$ $h^{-1}$), T is the leaf temperature ($^o$C), $T_S$ is the standard temperature of 30 $^o$C and $\beta$ is the temperature sensitivity ($^o$C$^{-1}$) of emissions. Often the value 0.09 $^o$C$^{-1}$ is used for $\beta$ to describe monoterpene emissions. In our monthly and daily mean concentration data temperature sensitivity was clearly higher ($\beta$=0.20 $^o$C$^{-1}$, Figure 5

and Table 3). The temperature affects also the vertical mixing of air and a lower mixing after warm sunny days is one probable reason for increased temperature sensitivity of concentrations. Even though the value 0.09 $^o$C$^{-1}$ is often used for $\beta$ to model emissions, it is known to vary (Hakola et al. 2006). Also here the temperature sensitivity of daily mean MT concentration for summer months ($\beta$=0.27 $^o$C$^{-1}$, Jun-Aug) was higher than for the whole growing season ($\beta$=0.20 $^o$C$^{-1}$, Apr-Nov).

To study temperature sensitivity of individual MTs, data from GC-MS3 was used. Exponential correlations of monthly means with temperature were found to have $R^2$>0.91 (Table 3 and Figure S2) for all monoterpenoids except 1,8-cineol ($R^2$=0.77), p-cymene ($R^2$=0.72), bornylacetate ($R^2$=0.71) and linalool ($R^2$=0.25). Tarvainen et al. (2005) found that in Scots pine emissions, 1,8-cineol was the only MT, which was both light and temperature dependent while others were only temperature dependent. *p*-Cymene has been detected e.g. in spruce emissions (Hakola et al. 2017), but it has also anthropogenic sources (Hakola et al.

2012). Linalool is known to be emitted by trees as a result of biotic stress (Petterson, 2007, Blande et al. 2009). Bornylacetate, linalool and 1,8-cineol have very low concentrations, which also results in higher uncertainty. For the MTs with high ($R^2$>0.91) temperature correlation, $\beta$-values of monthly means varied between 0.15 and 0.26 $^o$C$^{-1}$ being lowest for camphene and highest for $\beta$-pinene.

**3.2.2 Correlation of SQT concentrations with temperature**

As for the MTs also monthly and daily means of SQTs had very strong exponential correlation with temperature (Table 3, Figure S3). Temperature sensitivity of SQTs was even higher than for MTs. SQT emissions from Norway spruce (Hakola et al., 2017) and Scots pine (Tarvainen et al., 2005) are closely correlated with temperature, but SQT emissions may also be




influenced by light (Duhl et al. 2008). Especially daily mean β-caryophyllene concentrations had very high exponential correlation with temperature ($R^2_{Jun-Aug}$=0.96) supporting only temperature dependent emissions. Monthly means of SQT sum had also very high exponential correlation with temperature ($R^2$=0.97) indicating that also seasonality is driven by the temperature. For the other SQTs correlations were lower than for β-caryophyllene. Low concentrations with higher

5  measurement uncertainty and e.g. light and stress related emissions may have significant effects on the correlations. β-Farnesene is known to be emitted due to the biotic stress (Kännaste et al., 2009) and it has been shown to increase simultaneously with linalool in the emissions of Norway spruce and Scots pines (Hakola et al. 2006 and 2017). However, linalool and β-farnesene concentrations did not correlate in our data. Bouvier-Brown et al. (2009) suggested that at least in ponderosa pine forest β-farnesene emissions are both temperature and light dependent.

**Table 3:** Correlation of VOC concentrations with temperature at SMEAR II in 2016, intercept (a) of temperature dependence curve, temperature sensitivity (β) and temperature correlations ($R^2$) of monthly (Apr-Nov) and daily (Jun-Aug) mean concentrations and mixing layer height scaled concentration of individual measurements points ($C_{MLH}$). Fitted curves were exponent functions $y=ae^{\beta x}$, where y=concentration or MLH scaled concentration, x=temperature and β=temperature

15  sensitivity.

|  | Monthly mean (Apr-Nov) | | | Daily mean (Jun-Aug) | | | $C_{MLH}$ (Apr-Nov) | | |
|---|---|---|---|---|---|---|---|---|---|
|  | a (pptv) | β ($C^{-1}$) | $R^2$ | a (pptv) | β ($C^{-1}$) | $R^2$ | a | β ($C^{-1}$) | $R^2$ |
| Isoprene | 0.76 | 0.16 | 0.74 | 0.23 | 0.24 | 0.84 | 90 | 0.19 | 0.52 |
| MBO | 0.18 | 0.26 | 0.80 | 0.08 | 0.31 | 0.70 | 30 | 0.22 | 0.64 |
| α-Pinene | 6.07 | 0.23 | 0.95 | 1.93 | 0.30 | 0.85 | 750 | 0.17 | 0.63 |
| Camphene | 2.34 | 0.15 | 0.97 | 0.64 | 0.21 | 0.88 | 110 | 0.18 | 0.63 |
| β-Pinene | 0.68 | 0.26 | 0.97 | 0.45 | 0.27 | 0.82 | 80 | 0.19 | 0.65 |
| 3Δ-Carene | 1.86 | 0.25 | 0.95 | 0.59 | 0.30 | 0.88 | 190 | 0.19 | 0.66 |
| p-Cymene | 2.44 | 0.10 | 0.72 | 16.53 | -0.04 | 0.07 | 60 | 0.20 | 0.58 |
| 1,8-Cineol | 0.88 | 0.15 | 0.77 | 0.52 | 0.18 | 0.71 | 120 | 0.16 | 0.64 |
| Limonene | 0.45 | 0.25 | 0.91 | 0.14 | 0.34 | 0.91 | 40 | 0.21 | 0.61 |
| Linalool | 0.09 | 0.12 | 0.25 | 0.01 | 0.23 | 0.69 | 30 | 0.13 | 0.48 |
| Myrcene | 0.19 | 0.20 | 0.93 | 0.03 | 0.33 | 0.77 | 30 | 0.18 | 0.64 |
| Bornylacetate | 0.04 | 0.20 | 0.43 | 0.005 | 0.34 | 0.71 | 30 | 0.12 | 0.56 |
| **MT sum** | **14.38** | **0.22** | **0.96** | **5.57** | **0.27** | **0.88** | **1500** | **0.18** | **0.67** |
| MACR | 4.02 | 0.00 | 0.00 | 0.21 | 0.17 | 0.86 | 170 | 0.12 | 0.24 |
| Nopinone | 0.44 | 0.12 | 0.86 | 0.07 | 0.25 | 0.80 | 70 | 0.14 | 0.55 |
| Longicyclene | - | - | - | 0.003 | 0.28 | 0.69 | 10 | 0.13 | 0.4 |
| β-Farnesene | - | - | - | 0.003 | 0.37 | 0.83 | 140 | 0.09 | 0.13 |
| β-Caryophyllene | 0.04 | 0.34 | 0.87 | 0.019 | 0.37 | 0.96 | 50 | 0.18 | 0.51 |
| Other SQTs | - | - | - | 0.006 | 0.41 | 0.70 | 30 | 0.17 | 0.31 |
| **SQT sum** | **0.07** | **0.32** | **0.96** | **0.006** | **0.49** | **0.95** | **40** | **0.21** | **0.50** |
| Pentanal | 4.2 | 0.04 | 0.23 | 1.73 | 0.09 | 0.70 | 280 | 0.13 | 0.28 |
| Hexanal | 2.24 | 0.09 | 0.66 | 1.60 | 0.11 | 0.90 | 190 | 0.16 | 0.38 |



| | | | | | | | | | |
|---|---|---|---|---|---|---|---|---|---|
| Octanal | 0.65 | 0.11 | 0.22 | 1.12 | 0.09 | 0.36 | 100 | 0.15 | 0.26 |
| Nonanal | 0.31 | 0.14 | 0.07 | 1.88 | 0.08 | 0.70 | 100 | 0.15 | 0.26 |
| Decanal | 0.19 | 0.21 | 0.67 | 0.84 | 0.13 | 0.43 | 50 | 0.20 | 0.36 |
| *trans*-2-Hexenenal | 0.006 | 0.37 | 0.82 | 0.07 | 0.19 | 0.57 | 90 | 0.31 | 0.14 |
| Acetic acid | 947 | 0.01 | 0.02 | 107.3 | 0.13 | 0.22 | 57224 | 0.12 | 0.27 |
| Propanoic acid | 31.1 | 0.06 | 0.20 | 2.08 | 0.20 | 0.39 | 2728 | 0.14 | 0.36 |
| Butanoic acid | 27.7 | 0.06 | 0.49 | 23.45 | 0.06 | 0.31 | 2207 | 0.13 | 0.33 |
| Pentanoic acid | 3.48 | 0.15 | 0.58 | 0.84 | 0.23 | 0.65 | 573 | 0.16 | 0.41 |
| Hexanoic acid | 0.27 | 0.20 | 0.42 | 0.05 | 0.30 | 0.30 | 232 | 0.16 | 0.43 |

### 3.2.3 Correlation of isoprene and MBO concentrations with temperature

Isoprene emissions are both light and temperature dependent (Guenther et al. 1993, Ghirardo et al. 2010). Here correlation of isoprene daily mean concentrations with light and temperature activity factor (Guenther et al. 1993) was slightly lower ($R^2$=0.74) than for the temperature only ($R^2$=0.84, Figure S4). 2-Methyl-3-butenol (MBO) was somewhat better correlated with light and temperature activity factor ($R^2$=0.76) than with temperature only ($R^2$=0.70). This is in contrast with Scots pine emissions, where MBO has been found to be only temperature dependent (Tarvainen et al. 2005).

Even though diurnal variation of most MT, SQT and MBO concentrations are not following the ambient temperature, isoprene has the highest concentrations during the day and 30-minute mean concentrations have exponential correlation with ambient temperature (Figure S5, $R^2$=0.64).  Due to the close link between isoprene production and light, isoprene is produced and emitted from trees only during the light hours and is therefore detected in the air only during the day while MBO, MTs and

SQTs are emitted from storage pools inside the needles or leaves also during the night and due to lower vertical mixing ambient air concentrations are higher then (Ghirardo et al. 2010).

### 3.2.5 Correlation of terpenoid reaction product concentrations with temperature

Nopinone concentrations show very clear exponential correlation with temperature ($R^2_{daily}$=0.80) due to temperature dependence of its precursor (β-pinene) and faster production on warm and sunny summer days. Temperature sensitivity of nopinone daily means (β=0.25 °$C^{-1}$) is close to the sensitivity of its precursor β-pinene (β=0.27 °$C^{-1}$).

MACR, which is a reaction product of isoprene, has as high correlation ($R^2$=0.86) with temperature in summer as its precursor

isoprene ($R^2$=0.84) but the temperature sensitivity is a bit lower ($β_{isoprene}$=0.24 °$C^{-1}$ and $β_{MACR}$=0.17 °$C^{-1}$). Similar to its precursor, 30 minute mean concentrations of MACR have also low exponential correlation with temperature ($R^2$=0.32, Figure S5), but monthly means of MACR are not correlating with temperature ($R^2$<0.01, Table 3). MACR has also direct



anthropogenic sources (Biesenthal and Shepson, 1997) and in spring and autumn when biogenic emissions are lower, the influence of these sources is expected to be more important also due to longer lifetimes of VOCs in the atmosphere.

### 3.2.6 Correlation of OVOC concentrations with temperature

Since concentrations of most $C_5$-$C_{10}$ aldehydes are very close to detection limits, results are more scattered, but still clearly showing strong correlation with the temperature. Highest correlation of daily means in summer (Jun-Aug) was found for hexanal ($R^2$=0.90) and lowest for octanal ($R^2$ =0.36) and decanal ($R^2$ =0.43, Table 3 and Figure S5). Temperature sensitivities of aldehydes (0.08-0.13 °$C^{-1}$) were clearly lower than for terpenoids (0.18-0.67 °$C^{-1}$). Aldehydes have direct biogenic emissions

(Seco et al. 2007, Hakola et al. 2017), but they are also produced in the atmosphere by the oxidation of other VOCs. The correlation of daily mean concentrations of trans-2-hexenal with light and temperature activity factor (Guenther et al. 1993) was higher ($R^2$=0.71) than just with temperature ($R^2$=0.57) indicating light dependent source.

As for the isoprene and its reaction product (MACR), diurnal variation of pentanal and hexanal concentrations have also

correlation with temperature and temperature sensitivities for 30 minute mean concentrations ($\beta_{pentanal}$=0.07 °$C^{-1}$ and $\beta_{hexanal}$=0.08 °$C^{-1}$, Figure S5b) are close to the MACR ($\beta_{MCAR}$=0.06 °$C^{-1}$). This indicates that photochemical reactions could be an important source for these compounds as well.

A weak correlation with temperature was found also for VOAs, but it was lower than for most other studied VOCs (Table 3).

Due to the long lifetime of VOAs, background concentrations and anthropogenic sources are expected to have higher effect on concentrations and therefore their effect of local temperature dependent emissions and production in the air remains unclear. Correlation of daily means was highest for pentanoic acid ($R^2$=0.65, Table 3, Figure S6). Temperature sensitivity of butanoic acid daily means ($\beta$=0.06 °$C^{-1}$) was lower than for other VOAs. Butanol concentrations at the site are strongly affected by the emissions from particle counters used at the site and it is expected to produce butanoic acid.

### 3.2.7 Seasonality of temperature correlations

Variation of the daily mean concentrations is best explained by temperature in summer (Table 4). Also temperature sensitivity of MT, SQT and isoprene concentrations are highest during the summer months and lower in autumn and spring. In summer,

emissions from trees are expected to play a major role, but in spring and autumn relative impact of other emissions (e.g. sawmill emissions) increases.

In May values lower than expected by the overall temperature dependence were detected (Figure 5b). This is most probably explained by the beginning of the growing season (mean ETS<200) with lower emission potentials (Hakola et al. 2001 and





2012). In autumn (Sep-Nov), when values were more scattered (Figure 5b), emissions from fresh leaf litter are expected to have significant contribution to the concentrations (Hellén et al., 2006, Aaltonen et al., 2011, Mäki et al., 2017). Then MT sum was correlating (Figure S7) a bit better with soil humus layer temperature ($R^2$=0.87) than with ambient temperature ($R^2$=0.80), which also indicates the soil related sources. During colder months, when biogenic emissions are low, also anthropogenic

emissions have higher relative influence. This is detected by higher MT concentrations in November than expected by the general temperature correlation (Figure 5b). However, if daily means of all studied months are plotted together, correlation with temperature is relatively high ($R^2_{MT}$=0.83, $R^2_{SQT}$=0.67 $R^2_{ISOPRENE}$=0.68, Table 4).

**Table 4:** Characterization of the temperature dependence of isoprenoid concentrations with intercept (a), temperature

sensitivity (β) and correlation ($R^2$) of daily mean concentrations of MT sum and SQT sum measured at SMEAR II in different months in 2016. N=number of daily means. Fitted curves were exponent functions $y=ae^{\beta x}$, where y=concentration, x=temperature and β=temperature sensitivity.

|  | MT sum | | | | SQT sum | | | | Isoprene | | | |
|---|---|---|---|---|---|---|---|---|---|---|---|---|
|  | N | a | β (°C$^{-1}$) | $R^2$ | N | a | β (°C$^{-1}$) | $R^2$ | N | a | β (°C$^{-1}$) | $R^2$ |
| Apr | 9 | 8.16 | 0.18 | 0.43 | 5 | 0.17 | -0.05 | 0.11 | 2 | - | - | - |
| May | 13 | 1.19 | 0.33 | 0.67 | 13 | 0.10 | 0.24 | 0.82 | 13 | 0.31 | 0.19 | 0.70 |
| Jul | 8 | 1.19 | 0.26 | 0.92 | 8 | 0.09 | 0.33 | 0.91 | 8 | 0.09 | 0.29 | 0.87 |
| Aug | 7 | 6.53 | 0.37 | 0.82 | 7 | 0.001 | 0.63 | 0.92 | 7 | 0.14 | 0.31 | 0.94 |
| Sep | 15 | 29.2 | 0.12 | 0.18 | 15 | 1.48 | 0.05 | 0.02 | 15 | 0.89 | 0.14 | 0.60 |
| Oct | 5 | 13.4⁻ | -0.08 | 0.16 | - | - | - | - | 5 | 1.3 | -0.08 | 0.05 |
| Nov | 9 | 11.7 | 0.11 | 0.69 | - | - | - | - | - | - | - | - |
| Apr-Nov | 66 | 10.64 | 0.20 | 0.83 | 48 | 0.06 | 0.31 | 0.67 | 50 | 0.50 | 0.18 | 0.68 |

**3.2.8 Temperature sensitivities vs. vapor pressures**

Temperature sensitivities (β-values) of the most abundant terpenoids were found to be dependent on their vapor pressures (Figure 6). Vapor pressures have been estimated with the AopWinTM module of the EPITM software suite (https://www.epa.gov/tsca-screening-tools/epi-suitetm-estimation-program-interface, EPA, U.S.A). Higher β-values were

found for the terpenes with lower vapor pressure, higher boiling point and higher carbon number. This indicates that temperature sensitivity is driven by the volatility of the compounds. In addition to temperature sensitivities of monthly means shown in figure 6, also summertime daily means of terpenes had the same dependence on vapor pressures. However, camphene, p-cymene, 1,8-cineol and linalool did not show this dependence neither for monthly or daily means. For these compounds temperature sensitivity was lower than expected based on the volatility. The possible reasons for these differences are as

mentioned also in the previous sections: the concentrations of camphene and p-cymene are affected by the emissions of the closeby sawmill, 1,8-cineol has also light dependent emissions and linalool is emitted from plants due to stress.





Even though VOAs had lower correlation with temperature than terpenes (Table 3), dependence of temperature sensitivity on vapor pressures was also found for all other VOAs except butanoic acid, which is expected to be produced from the 1-butanol used in other instruments at the site. For $C_5$-$C_{10}$ aldehydes only monthly means had this dependence. Summertime daily means

of aldehydes had higher correlation with temperature, but still β-values did not follow the vapor pressures.

These dependencies can be used to estimate the kind of compound that could explain missing reactivity found by total reactivity measurements or to assist the identification of compounds in direct mass spectrometric methods.

**3.2.9. Simple proxies for estimating local BVOC concentrations**

Kontkanen et al. (2016) have developed MT proxies, which are used for calculating concentrations of the MT sum at the SMEAR II. The proxies are based on the temperature-controlled emissions from the forest ecosystem, the dilution caused by the mixing within the boundary layer and different oxidation processes. Our data shows that monthly and daily means of both

sum and individual MT and most other BVOC concentrations can be described relatively well using only the temperature (Table 3 and 4, Figure 5) and a simplified proxy for daily or monthly mean concentrations would be

$$[BVOC(monthly \ or \ daily)_i]_{proxy} = ae^{\beta T} \qquad\qquad (10)$$

where a and β are empirical coefficients found from the table 3 and obtained from the correlation of monthly and daily mean concentrations with temperature (Figure 7) and T is ambient temperature.

However, for describing diurnal variation mixing of air has to be taken into account. To roughly describe the dilution caused by the vertical mixing we multiplied the concentrations with mixing layer heights ($C_{MLH}$=[VOC ] x MLH) and studied the

correlation of these $C_{MLH}$ values with temperature (Table 3 and Figure 7). All individual measured data points available from the year 2016 were used except the cases when MLH was below the detection limit of LIDAR (< 60 m). Therefore the highest values during the most stable nights are missing. The correlation of $C_{MLH}$ values with temperature was best for the MTs ($R^2_{MTsum}$=0.67). The modeling study of Zhou et al. (2017) showed that variation of monoterpene concentrations is mainly driven by the emissions and mixing, while for faster reacting SQTs also oxidation plays a role. For oxygenated compounds

also production in the air and deposition have an effect on local concentrations and therefore correlation of $C_{MLH}$ values with temperature are lower than for MTs (Table 3). In our case proxy for the concentration of MT or SQT sum or an individual compound ($BVOC_i$,), when MLH> 60m, would be



$$[BVOC_i]_{proxy} = \frac{ae^{\beta T}}{MLH} \qquad\qquad (11)$$

where a and β are empirical coefficients found from the table 3 and obtained from the correlation of concentrations multiplied by the mixing layer height ($C_{MLH}$) with temperature (Figure 7), T is ambient temperature and MLH is mixing layer height.

### 3.3 Importance of studied BVOCs for local atmospheric chemistry

### 3.3.1 Reactivity of measured BVOCs

To describe the effects of different compounds and compound groups on the oxidation capacity of air we calculated OH, $NO_3$ and $O_3$ reactivities for studied BVOCs using the measured concentrations and reaction rates with different oxidants (Eq. 4). OH reactivity of MTs is clearly higher than for the any other VOC group at this boreal forest site showing the importance of the MTs for the local OH chemistry (Figure 8a). OH reactivity of monoterpenes is 10 times higher than the reactivity of SQTs even in July when SQT concentrations were highest. OH reactivity of other compound groups was minor (ca. 4% in July).

Based on additional measurements in July, also the contribution of <$C_5$ carbonyls (formaldehyde, acetaldehyde, acetone and butanal) was minor. However, even adding up the reactivity of all the BVOCs, anthropogenic VOCs and other OH reactive compounds measured at the site, OH reactivity is much lower (<50 %) than the total reactivity measured at the site by Sinha et al. (2010), Nölscher et al. (2012) and Praplan et al. (2018 in preparation).

Since $O_3$ is reacting only with unsaturated VOCs, of the measured VOCs only isoprene, most MTs, SQTs and unsaturated alcohols contribute to the $O_3$ reactivity. From May to September SQTs had major contribution to the $O_3$ reactivity (Figure 8b). Even though MT concentrations are ca. 50 times higher than SQT concentrations, $O_3$ reactivity given by the SQTs is about 3 times higher than the reactivity of MTs. Hakola et al. (2017) also showed the high importance of SQTs for the $O_3$ reactivity in the spruce emissions. This indicates that SQTs and escpecially β-caryophyllene (Figure S8d) have much higher effect for

example on local ozone deposition than MTs. Several studies have shown that measured ozone deposition fluxes cannot be explained by modelled stomatal and known non-stomatal sinks, such as reactions with measured VOCs in the gas phase (Clifton et al. 2017; Wolfe et al. 2011; Rannik et al. 2012). Higher than expected impact of the SQTs could explain at least part of the discrepancy.

Also $NO_3$ radicals are mainly reacting with unsaturated VOCs and MTs have clearly highest contribution to the $NO_3$ reactivity of BVOCs at the site (Figure 8c). Of the SQTs only β-caryophyllene was considered since reaction rate coefficients were not available for the others. However, β-caryophyllene had the highest concentrations of all SQTs, but still did not have significant



effect on NO$_3$ reactivity. Liebmann et al. (2018) measured total NO$_3$ reactivity at the site in September 2016 and BVOCs measured at the same time explain 70% of the reactivity during the night but only 40 % during the day.

Similar to concentrations of the individual MTs α-pinene had the highest contribution to the OH, O$_3$ and NO$_3$ reactivity (Figure
S8). However, the importance of limonene and especially terpinolene for the local chemistry was clearly higher than their contribution to the concentrations. In addition, limonene has higher SOA yield than MTs generally (Lee et al. 2006) and therefore it is expected to be more important for SOA production than concentrations indicate. Of the individual SQTs β-caryophyllene had a major contribution to OH reactivity and for the O$_3$ reactivity it had the highest (>60 % in June-August) contribution of all measured VOCs (Figure S8).

### 3.3.2. Oxidation products and SOA

Oxidation of VOCs, under various environmental conditions, produces a variety of gas- and particle phase products that are relevant for atmospheric chemistry and SOA production. To describe this we calculated production rates of oxidation products
(OxPRs) from isoprene, MT, SQT and OVOC reactions as described in section 2.5.

More oxidation products were produced from SQTs than MTs (Figure 8d). The contribution of OVOCs, aromatic hydrocarbons and isoprene was very low. SQTs were very important especially during summer nights (Figure 8 and 9). In daytime contributions of MTs and SQTs were equal during all other months except in July when SQTs were dominating even in the
middle of the day. In addition, photo-oxidation of SQTs in smog chamber experiments has been shown to generally result in a much greater aerosol yield than MTs (Hoffmann et al., 1996; Griffin et al., 1999, Lee et al. 2006) and therefore they are expected to have a strong influence on SOA production. However, these production rates describe very local situation and even though fast reactions of SQTs have very strong local effects also MTs react relatively fast producing secondary products in regional scale. Global emissions of SQTs have been estimated to be about 20 % of the MT emissions (Guenther et al. 2012),
but this is probably a low-end estimate since evidence for additional unaccounted SQTs and their oxidation products clearly exists (Yee et al. 2018).

Often α-pinene is used as a proxy for all BVOCs, but as shown in Figure 9 contribution of α-pinene to the total of oxidation reactions was relatively low (ca. 20 %). The most important individual reaction producing oxidation products at the site was
the reaction of β-caryophyllene with O$_3$. For SQTs contribution of OH and NO$_3$ reactions were very low especially during summer months (<2 %). Also for MTs O$_3$ reactions were the most important, OH radicals had about 30 % contribution during summer days and NO$_3$ reactions were important in nighttime. Peräkylä et al. (2014) stated that for MTs nighttime oxidation is dominated by the NO$_3$ radicals whereas daytime oxidation is dominated by the O$_3$. However, like in our study, O$_3$ was dominating also during the summer nights. If we take into account that also emissions and concentrations are highest during



the summer, ozone becomes the most important oxidant for the OxPR of MTs at night as well. For SQTs O₃ oxidation is clearly dominating the first step of the reactions. However, reaction products of MTs and SQTs, that have lost all double bonds, continue to react with OH and NO₃ and their total contribution is expected to be higher. It has been suggested that during nighttime reaction products of MTs build-up in the atmosphere and are oxidized after sunrise with OH radicals promoting

particle growth (Peräkylä et al. 2014). Our results suggest that this applies also for SQTs.

## 4. Conclusion

We have measured an exceptionally large dataset of VOCs in boreal forest including terpenoid compounds (isoprene, MTs,

SQTs), aldehydes, alcohols and organic acids during 26 months in three years period. The measurements revealed that of the terpenoids, MTs had the highest concentrations at the site, but we were able to measure also highly reactive SQTs, such as β-caryophyllene and other SQTs in ambient air due to the availability of an instrument with improved sensitivity. Our result indicat that in addition to terpenoids also most of the VOAs, aldehydes and alcohols have a biogenic origin either from direct emissions or by production from the other BVOCs in the air through oxidation reactions.

Temperature was the major factor controlling concentrations of BVOCs in the air of boreal forest. Both monthly and daily mean concentrations of MTs had very strong exponential correlation with temperature ($R^2_{monthly}$=0.92 and $R^2_{daily}$=0.88). SQT concentrations were even more strongly correlated with temperature and had higher temperature sensitivity than MTs. Especially monthly mean concentrations in 2016 were highly correlated with temperature ($R^2$=0.97). Results also indicate that

in spring and even more in autumn also other sources (e.g. needle and leaf litter) than temperature dependent emissions from main local trees have high impact on MT and SQT concentrations.

Temperature sensitivities of the most abundant terpenoids, aldehydes and VOAs within the same class of compounds were dependent on vapor pressures. This knowledge can be used to characterize the missing reactivity found in forests during total

reactivity studies (Yang et al., 2016) and to help with identification of the masses in direct mass spectrometric measurements of BVOCs and their reaction products.

We also evaluated the effect different BVOCs have on the local atmospheric chemistry and although MTs dominated OH and NO₃ radical chemistry, SQTs had a major impact on ozone chemistry, even though SQT concentrations are 30 times lower

than MT concentrations. These results indicate that SQTs have much higher effect on ozone deposition detected at the site than MTs. SQTs were also producing more oxidation products than MTs. Since products of SQTs are less volatile than MT oxidation products and SQTs are expected to have even higher impact on local SOA production. Both MT and SQT oxidation was dominated by ozone especially during summer. Oxidation of other VOC groups had very minor contributions to the





formation of oxidation products at the site. Our results clearly indicate that SQTs have to be considered in local SOA studies for example when interpreting results from direct mass spectrometric measurements or modelling SOA formation and growth.

**Acknowledgements**

The research was supported by the Academy of Finland via Academy research fellow project (Academy of Finland, project 275608) and the Center of Excellence in Atmospheric Sciences (grant no. 307331). Data providers of SmartSmear AVAA portal are gratefully acknowledged.

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





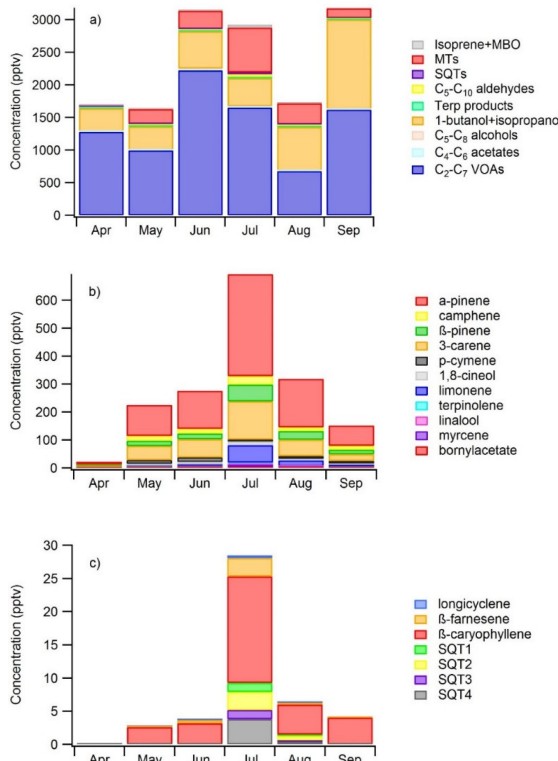

**Figure 1: Monthly mean concentrations of a) different compound groups, b) MTs and c) SQTs at SMEAR II in boreal**

5   **forest in 2016.**





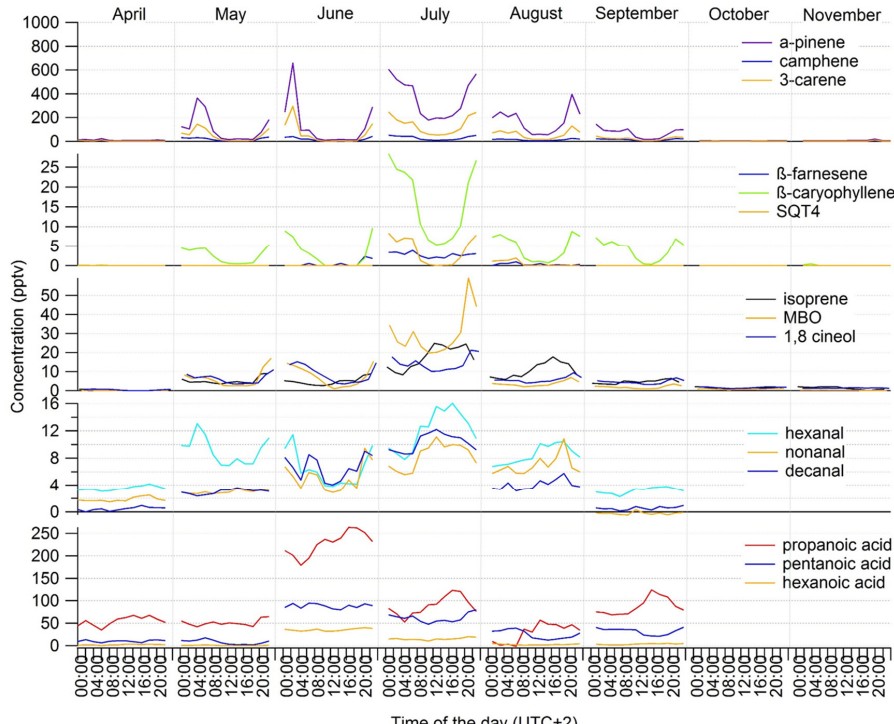

**Figure 2: Monthly mean diurnal variation of concentrations of different VOCs at SMEAR II station in 2016**



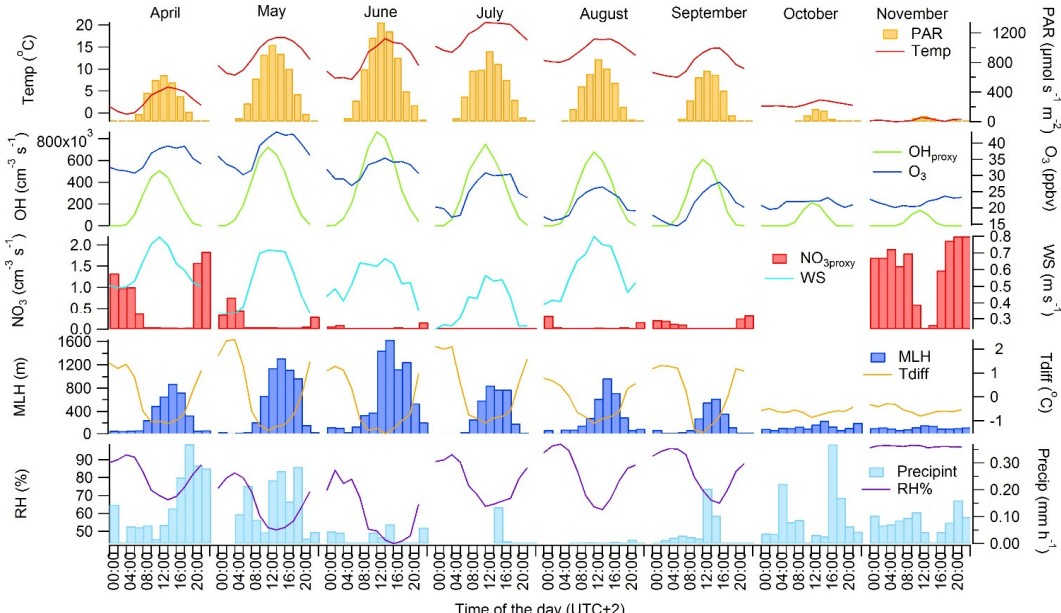

**Figure 3: Monthly mean diurnal variation of meteorological parameters and concentrations of main oxidants (OH radical, O₃ and NO₃ radical) in 2016 during the GCMS3 measurement periods. PAR=photosynthetic active radiation, WS=wind speed at the height of 8.4 m, MLH=mixing layer height, Tdiff=temperature difference between heights of 125 and 4.2 m, RH=relative humidity and precipint=intensity of precipitation. Wind speed data for Sep-Nov and NO₃ radical and RH data for October is missing.**





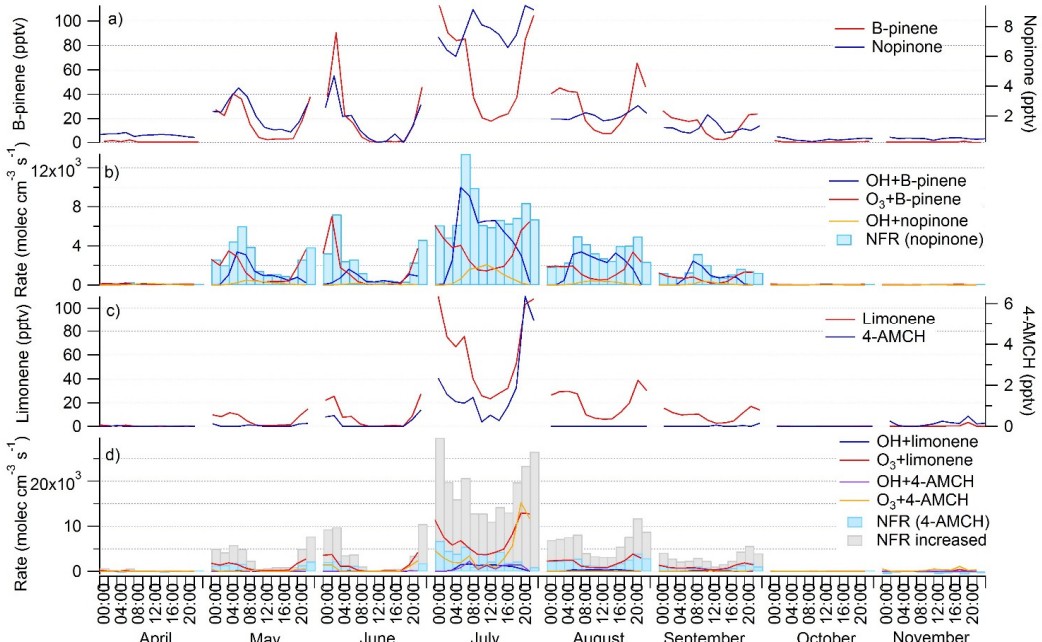

**Figure 4: Monthly mean diurnal variation of a) β-pinene and nopinone concentrations, b) production rates of nopinone from OH (OH+β-pinene) and O₃ (O₃+β-pinene) reactions, destruction rate of nopinone by OH reaction (OH+nopinone) and net formation rates (NFR) of nopinone, c) limonene and 4-AMCH concentrations and d) production rates of 4-AMCH from OH (OH+limonene) and O₃ (O3+limonen) reactions, destruction rates of 4-AMCH by OH (OH+nopinone) and O₃ (O3+4-AMCH) reactions, NRF and NRF with increased yields(NFR increased) of 4-AMCH.**



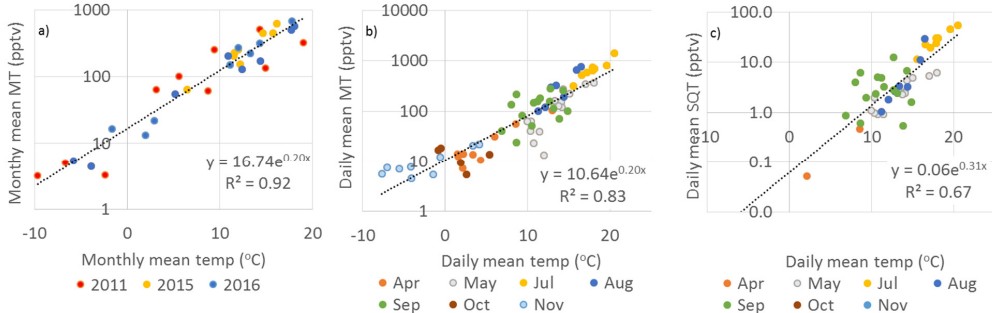

**Figure 5: Exponential correlation of temperature with a) monthly mean MT concentrations in 2011, 2015 and 2016, b) daily mean MT concentrations in 2016 and c) daily mean SQT concentrations in April-November 2016 measured at**
5 **SMEAR II.**

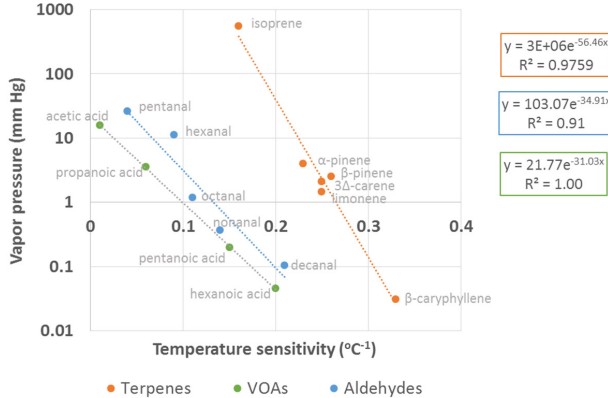

10 **Figure 6: Vapor pressure dependence of temperature sensitivities β (C$^{-1}$) of monthly mean concentrations measured at SMEAR II in 2016. Values for temperature sensitivities and exponent functions of temperature dependence of concentrations can be found in table 3.**





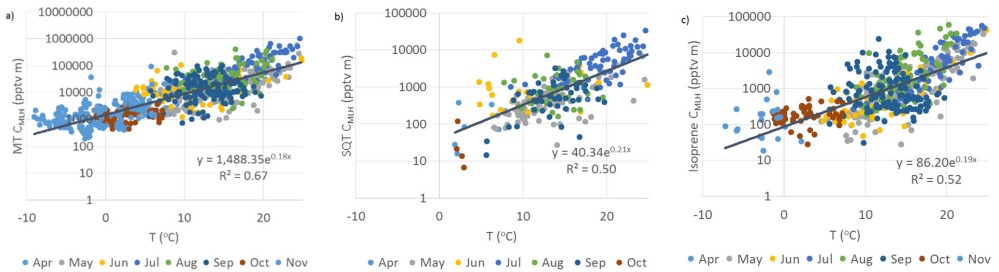

**Figure 7: Correlation of temperature with a) MT, b) SQT and c) isoprene concentrations multiplied by the mixing layer height (C$_{MLH}$) measured at SMEAR II station in 2016.**

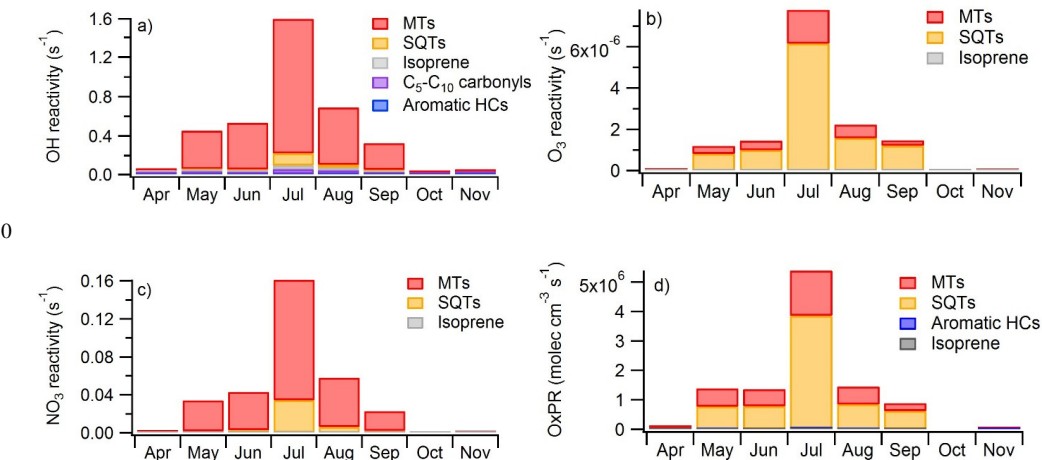

**Figure 8: a) OH reactivity, b) O₃ reactivity and c) NO₃ reactivity and d) secondary organic production rates (OxPR) of different VOC groups at SMEAR II during different months in 2016.**





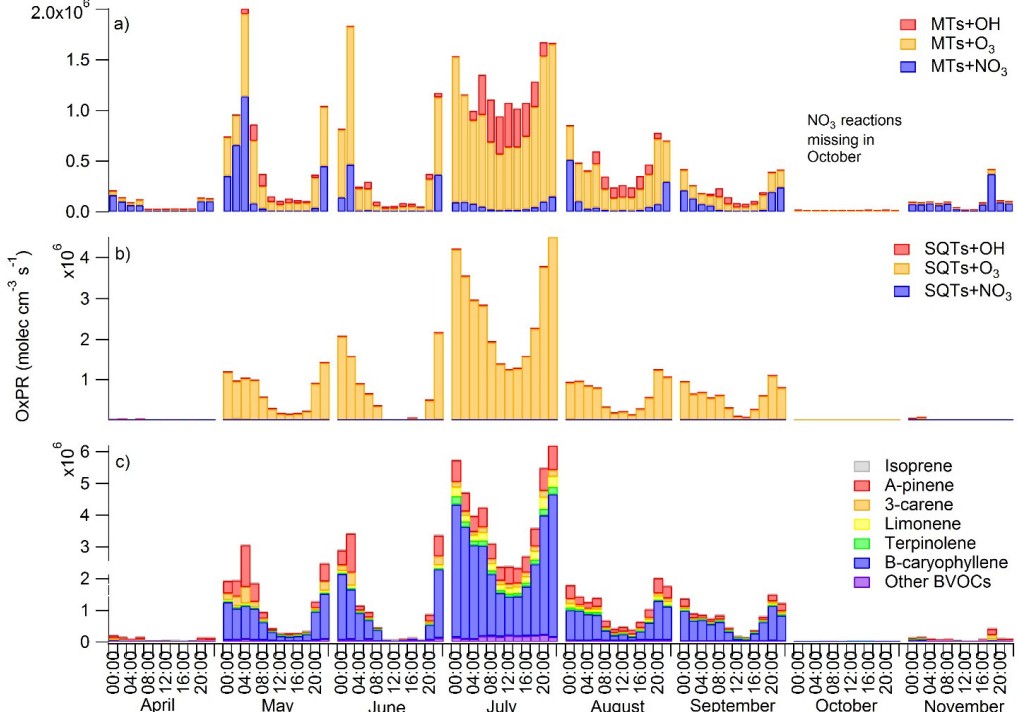

**Figure 9: Diurnal variation of production rates for secondary organic compounds (OxPR) from the reactions of a) MTs and b) SQTs with different oxidants (OH radical, O₃ and NO₃ radical) and c) contribution of individual BVOCs to the total OxPR during different months. NO₃ radical reactions are missing for the October since data for calculating its proxy is not available.**

