# Peer review of "Sesquiterpenes identified as key species for atmospheric chemistry in boreal forest by terpenoid and OVOC measurements"

_Atmospheric Chemistry and Physics, 2018_

## Referee Comment (RC1) · Anonymous Referee #1 · 12 Jun 2018

This study focuses on the measurement of VOCs (speciated monoterpenes and sequiterpenes and oxygenated species) performed for a period of two years in a Boreal forest. The measured data are compared with temperature to highlights trends and derive simple proxies to estimate the VOC concentrations. In addition, by using O3 data and proxies to estimate OH and NO3 radical concentrations, the reactivity of each class of compounds is calculated together with production rate of oxidation products.

This work contains a large amount of data and measurements of speciated sesquiterpens which are extremely sparse and limited. The manuscript is suitable for publication in ACP after the authors have considered the following points:

[Figure]

1. A more detailed comparison with measurements already performed at the site should be included. During the HUMPPA-COPEC 2010 campaign, for example, measurements and fluxes of VOCs were performed with GC and enclosed to the trees branches and the results have been published (Eerdekens et al., 2009; Yassaa et al., 2012; Hens et al., 2014). Putting the measurements from this study in the contest of what was previously observed at the same site would strengthen the conclusion of the manuscript. In addition, the results of the reactivity of the different VOC classes should be compared with a modelling study investigating the reactivity of VOC towards OH radicals in the specific site (Mogensen et al., 2011). A more structured comparison with previous measurements would be very interesting and could be included as a separate section (instead of scattered around the different compounds as it is at the moment) by possibly reducing the intercomparison of all species with temperature which occupies a large fraction of the discussion and could be partly moved to the supplementary information.

2. For the OH radical concentration determination, the authors refer to a publication by Petäjä et al. (2009). Unfortunately, this publication is not listed in the references but assuming they refer to the ACP paper, in that work there is no proxie described for the OH radical concentration. Proxies are given to determine the $H_2SO_4$ concentration. Therefore an explanation on how the OH radical concentration is calculated is needed. In addition, although it is true that there is a direct correlation between OH radical concentration and photolysis of $O_3$, a previous study which evaluated the OH radical budget at the same site (Hens et al., 2014) highlighted how the production from photolysis of ozone was actually marginal compared to other production paths. An error analysis study on the determination of the OH radical should be included. For the $NO_3$ radical concentration, a description of the formula is given (could be explicitly written as formula) but there is confusion regarding the value plotted in Fig. 3. The units for both OH and $NO_3$ radical concentrations are $cm^{-3} s^{-1}$ which is the unit for a production rate. Assuming the plots is showing concentrations (and if this is the case, please fix the unit on the axis), is the $NO_3$ concentration in the order of 0.5 pptv? How does it

compare with measurement of NO3 radicals performed at the same site (Liebmann et al., 2018)?

3. All the plots would be easier to follow if the x and y axis would be at the bottom and on the left of the figure, respectively (as for example figure 7b). Right now they are most of the time somehow in the figure making the reading of the numbers on the axis a bit difficult. The notation logarithm should also been added to the axis when necessary. Do the fit performed take into account the errors on both concentrations and temperatures (York fit, (York et al., 2004))? As temperature measurements are relative accurate, the error on the x-axis could be ignored but the error on the concentration of the different species should be taken into account for a proper analysis.

4. The manuscript, before publication, requires some careful language check. In particular articles are missing and the structure of the sentences is often confusing. Some examples are listed in the technical comments but they do not cover the entire manuscript.

Technical comments:

Title. Suggest "Long-term measurements of VOCs highlight the importance of sesquiterpenes for the atmospheric chemistry of a boreal forest".

Page 2, Line 8. Citation needed at the end of "…forest".

Page 2, Line 11. Citations needed at the end of "…studies".

Page 2, Line 14. More citations needed as examples of unmeasured BVOC.

Page 2, Line 14. "…Therefore a better characterization of…"

Page 2, Line 17. "…Once emitted, BVOCs readily…"

Page 2, Line 20. Suggest adding the word paths after reactions at the end of the sentence.

Page 2, Line 21. The expression "vary a lot" is for the spoken language. Suggest "…terpenoids are very different…"

Page 2, Line 22. "… essential to understand biosphere-atmosphere interactions…"

Page 2, Line 34. "…less sensitive instruments…"

Page 3, Line 1. "…to be emitted by the pine trees…"

Page 3, Line 4. The word lots can be substituted with the word many.

Page 3, Line 16. "…and 2016 in a boreal forest at the SMEAR II…"

Page 4, Line 16. "…in situ thermal-desorption gas…"

Page 4, Line 24. A small description of the heated inlet, although already described in details in another publication, would be beneficial to confirm the ability of the system to measure sesquiterpenes.

Page 4, Line 32. Remove the hooks for internal standard.

Page 5, Line 3. It is specified that an extra flow of 2.2 L min-1 was used. There is no indication of the flow drawn by the GC-MS2 (probably 1 L min-1 ?).

Page 5, Line 14. "…This method has been…"

Page 5, Line 16. " …A similar behavior…"

Page 5, Line 17. "…isomerization to be reproducible…"

Page 5, Lines 18-20. Sentence starting with "In our tests" and ending with "tube standards" is not clear and needs rephrasing.

Page 5, Line 20. "…Interconversion/degradation was not observed with the two other GC-MS used within this study…".

Page 5, Line 33. "What is identified by the authors as unknown sesquiterpenes? For which known sesquiterpenes was the instrument calibrated?

Page 6, Line 19. There are way more recent studies (compared to a study from 1994) on degradation of monoterpenes and products yields which should be considered for the calculation of PR.

Page 6, Line 30. It would be beneficial to have a little bit of background information on how the software derives rate coefficients for unknown VOC. Also, a list with all the VOCs species for which a derived rate coefficient was used (and the used rate coefficient) is needed.

Page 7, Line 15. The dependency of OH radical concentrations from UVB radiation was firstly showed by Rohrer and Berresheim (2006).

Page 9, Line 24. Please add the reference to measurements performed during the HUMPPA-COPEC campaign 2010 at the same site where monoterpenes were also measured (see general comment 1).

Page 10, Line 9. "...$\alpha$-pinene showed the highest concentration of the measured MT..."

Page 10, Line 25. Figure S1a) does not have any correlation plot but the mean diurnal variation and figure S1b) does not show any correlation factor equal to 0.85.

Page 11, Line 2. "...Concentrations of SQTs did not increased during the sawmill episode in contrast with what observed for MTs...". This is a suggestion of what the meaning of the sentence could be.

Page 11, Line 5. "...$\beta$-caryophyllene showed the highest concentrations among the measured SQTs followed by longicylene, $\beta$-farnesene and 4 unidentified SQTs detected only in the summer months (List of months)..."

Page 11, Line 8. A new study from the Amazonia forest (Bourtsoukidis et al., 2018) also shows large emissions of SQTs from soil.

Page 11, Line 10. "...$\beta$-farnesene, which was also detected in local..."

[Figure]

Page 11, Line 15. Same point as for Page 10, Line 25.

Page 11, Lines 16-17. The meaning of this sentence is not very clear. It is normal that sink terms (as much as production terms) will affect the concentration of a certain species. So what is the point the author is trying to make here?

Page 12, Line 9. "...but monthly averages of MACR..."

Page 12, Line 10. The possible anthropogenic origin of MACR should be discussed in more details. One paper is cited which, interestingly, suggests that a large source of MVK and MACR comes from car traffic. Would this be a reasonable source for the site? Are there more studies on anthropogenic sources for MACR? Why was MVK not measured?

Page 12, Line 18. "...During the summer month its concentration was...".

Page 12, Line 20. Please add the work by Kaminski et al. (2017).

Page 12, Lines 20-21. What is the message of the sentence starting with "Further reactions..."? Do the authors want to compare the yields and discuss possible differences between different studies?

Page 17, Line 21. "...BVOCs correlates exponentially with temperature...".

Page 18, Line 3. "...The high correlation with temperature observed indicates that temperature..."

Page 18, Line 22. The linalool is missing from the legend of figure S2b) and/or the data for bornylacetate are missing.

Page 19, Line 2. As most of the measured sesquiterpenes consists of $\beta$-caryophillene, it is not surprising that the sum of SQTs shows a high correlation with temperature.... Please use the greek letter (and not b) to label $\beta$-farnesene and $\beta$-caryophyllene in the figures.

Page 20, Lines 5-7. A little bit more discussion on the observed better correlation observed between temperature only and isoprene compared to temperature and light is needed as this is in contrast with what observed previously.

Page 20, Last paragraph. This concept was already discussed previously and, as there is no additional discussion about possible anthropogenic origins or how this is reasonable for such a remote site, it does not add anything from where it was previously discussed.

Page 22, Lines 2-6. The points made in this paragraph would probably not be so strong once a proper error analysis is introduced in the fit results. Though if, for example, a larger than expect concentration of MTs is observed in November and this is tentatively explained by anthropogenic origin of the monoterpenes, a more detailed discussion on which type of anthropogenic sources would produce which monoterpenes is needed.

Page 23, Line 20. Please use $\alpha$ instead of a.

Page 23, Line 21. Figure 7 does not seem to depict the results obtained when using equation 10 but more when using equation 11. It would be interesting to see how the proxies developed in this study compare with measurement of VOCs from previous studies at the same site.

Page 24, First 2 paragraphs. The addition of a table which includes the used rate coefficient for the different SQTs and MTs vs OH radicals, O3 and NO3 radicals would be beneficial for this section. The authors explain that SQTs have the largest contributions to the O3 reactivity despite the fact their concentration is $\sim$ 50 times lower than the concentration of MTs. This is not surprising as the rate coefficient of $\beta$-caryophyllene with O3 ($\sim$ 1e-14 cm3 s-1) is up to 3 order of magnitude faster than the rate with the main MTs measured at the site ($\alpha$-pinene, 9.4e-17 cm3 s-1, $\beta$-pinene, 1.9e-17 cm3 s-1, and carene, 4.8e-17 cm3 s-1). Similarly, as the rate coefficient with OH for different MTs and SQTs only ranges less than one order of magnitude, it is expected that MTs, as they are present in larger concentrations, dominate the OH reactivity. The point being,

it is not much surprising, that the SQTs dominates the O3 reactivity but rather it is important to underline the large absolute concentration of SQTs observed.

Page 25, Lines 5-6. Here it is a similar concept as before. Limonene and terpinolene both have relatively fast rate coefficient with OH radicals and O3 therefore, despite lower concentrations, they can have a large impact in the formation of secondary products.

Page 26, Line 29. Was the concentration of SQTs 30 or 50 times lower than the one of MTs? Anyway, as their reactivity depends on the product of concentration and rate coefficient, the sentence should be rephrase highlighting that due to the very fast rate coefficient of O3 with SQTs, a relatively (compared for example to monoterpenes) small concentration of SQTs can have a large impact on O3 deposition.

References

Bourtsoukidis, E., Behrendt, T., Yañez-Serrano, A. M., Hellén, H., Diamantopoulos, E., Catão, E., Ashworth, K., Pozzer, A., Quesada, C. A., Martins, D. L., Sá, M., Araujo, A., Brito, J., Artaxo, P., Kesselmeier, J., Lelieveld, J., and Williams, J.: Strong sesquiterpene emissions from Amazonian soils, Nature Communications, 9, 2226, doi:10.1038/s41467-018-04658-y, 2018.

Eerdekens, G., Yassaa, N., Sinha, V., Aalto, P. P., Aufmhoff, H., Arnold, F., Fiedler, V., Kulmala, M., and Williams, J.: VOC measurements within a boreal forest during spring 2005: on the occurrence of elevated monoterpene concentrations during night time intense particle concentration events, Atmospheric Chemistry and Physics, 9, 8331-8350, 2009.

Hens, K., Novelli, A., Martinez, M., Auld, J., Axinte, R., Bohn, B., Fischer, H., Keronen, P., Kubistin, D., Nölscher, A. C., Oswald, R., Paasonen, P., Petäjä, T., Regelin, E., Sander, R., Sinha, V., Sipilä, M., Taraborrelli, D., Tatum Ernest, C., Williams, J., Lelieveld, J., and Harder, H.: Observation and modelling of HOx radicals in a boreal

forest, Atmos. Chem. Phys., 14, 8723-8747, doi:10.5194/acp-14-8723-2014, 2014.

Kaminski, M., Fuchs, H., Acir, I. H., Bohn, B., Brauers, T., Dorn, H. P., Häseler, R., Hofzumahaus, A., Li, X., Lutz, A., Nehr, S., Rohrer, F., Tillmann, R., Vereecken, L., Wegener, R., and Wahner, A.: Investigation of the $\beta$-pinene photooxidation by OH in the atmosphere simulation chamber SAPHIR, Atmos. Chem. Phys., 17, 6631-6650, doi:10.5194/acp-17-6631-2017, 2017.

Liebmann, J., Karu, E., Sobanski, N., Schuladen, J., Ehn, M., Schallhart, S., Quéléver, L., Hellen, H., Hakola, H., Hoffmann, T., Williams, J., Fischer, H., Lelieveld, J., and Crowley, J. N.: Direct measurement of NO3 radical reactivity in a boreal forest, Atmos. Chem. Phys., 18, 3799-3815, doi:10.5194/acp-18-3799-2018, 2018.

Mogensen, D., Smolander, S., Sogachev, A., Zhou, L., Sinha, V., Guenther, A., Williams, J., Nieminen, T., Kajos, M. K., Rinne, J., Kulmala, M., and Boy, M.: Modelling atmospheric OH-reactivity in a boreal forest ecosystem, Atmos. Chem. Phys., 11, 9709-9719, doi:10.5194/acp-11-9709-2011, 2011.

Petäjä, T., Mauldin, R. L., Kosciuch, E., McGrath, J., Nieminen, T., Paasonen, P., Boy, M., Adamov, A., Kotiaho, T., and Kulmala, M.: Sulfuric acid and OH concentrations in a boreal forest site, Atmospheric Chemistry and Physics, 9, 7435-7448, 2009.

Rohrer, F., and Berresheim, H.: Strong correlation between levels of tropospheric hydroxyl radicals and solar ultraviolet radiation, Nature, 442, 184-187, doi:10.1038/nature04924, 2006.

Yassaa, N., Song, W., Lelieveld, J., Vanhatalo, A., Back, J., and Williams, J.: Diel cycles of isoprenoids in the emissions of Norway spruce, four Scots pine chemotypes, and in Boreal forest ambient air during HUMPPA-COPEC-2010, Atmospheric Chemistry and Physics, 12, 7215-7229, doi:10.5194/acp-12-7215-2012, 2012.

York, D., Evensen, N. M., MartÄśÌĄnez, M. L., and De Basabe Delgado, J.: Unified equations for the slope, intercept, and standard errors of the best straight line, American Journal of Physics, 72, 367-375, doi:doi:http://dx.doi.org/10.1119/1.1632486, 2004.

---

## Referee Comment (RC2) · Anonymous Referee #2 · 30 Jun 2018

This paper is represents many years of BVOC data at a boreal forest in Hyytiala, Finland. The major advancement was detection and measurement of reactive sesquiterpenes, particularly b-caryophyllene, in ambient air. There is also quantitative evidence of BVOC oxidation products, carbonyls, alcohols, and acids coming from the forest. Analyzed data showed predictable temperature dependencies and contribution to oxidative capacity of the atmosphere as well as SOA production.

This manuscript is very detailed and informative; I just have a few minor comments.

Sampling and Calibration

Because the major "breakthrough" is the ambient quantification of very reactive com-

pounds, I am interested in the sampling techniques and calibration that were used.

VOCs were calibrated using dilute liquid standards injected onto the adsorbent. Can you describe that a little further? What concentration ranges were used? Was it injected using a syringe or put into an air flow? Is this representative of the sampling technique (using liquid vs. gas-phase compounds and not accounting for losses in the sampling system)?

I understand the sampling of for GC-MS2 was a sub-sample from a larger flow (2.2 L/min). Please clarify the text; it took me awhile to figure this out. Part of the confusion is the use of "extra flow" terminology (pg 5, line 3)

Pg 5, line 14: "used method" does not make sense here

Pg 5, line 19: "suffering by the most degradation" needs to be re-phrased.

Sampling for GC-MS3 used two different types of inlets. Is that correct? Why was that done? I do not understand "stainless steel tube was used to destroy ozone" (pg 5, line 25). How was ozone destroyed?

Pg 5, line 26 omit "a" before 40 mL/min

Content

The idea that p-cymene (4-isopropyl toluene) is partially anthropogenic can be mentioned sooner (pg 9, line 23) to explain why it has a different pattern.

Pg 10, line 32: what is meant by the fact that MT data is more abundant? How is that different from the fact that there is "very little data on atmospheric SQT concentrations"?

Pg 12, line 10: elaborate a bit on the anthropogenic sources of MACR.

Pg 17, line 4: What is LC-UV?

It would be nice to have a table outlining your major BVOC species, their reaction rate

constants, and their vapor pressures. In other words, outline the data used to make the graphs.

Figure 2: why is the propanic acid so high in June?

Figure 5: There is a stronger correlation with monthly measurements vs. daily measurements of MT concentration and temperature. Why wasn't the monthly data included for SQT?

Wording

Be sure to fix the grammar throughout this manuscript. Here are a few examples.

When referring to PTR-MS and GC-MS, be careful of the verb agreement. You can either use the abbreviation to represent the instrument (e.g. gas chromatograph) or the technique (gas chromatography). I think you chose the former, but then you need to ensure there are articles (e.g. "the", "a", etc.) before the abbreviation and a corresponding verb.

Section 2.3: the text below the equations does not agree with the content of the equations. For example, "x" is not in the equation. I think all of the "ks" should be lowercase (reaction rate constants), but one was uppercase. Rephrase the explanation so that you don't use "yields are yields".

Pg 6, lines 28-29: rephrase this sentence; it is confusing

Pg 7, lines 12-13: rephrase sentence

Pg 9, line 22: insert a comma after "terpeniods"

Pg 13, line 12: change "aq" to "a"

Pg 14, lines 9-13: re-work this sentence because it is wordy. I don't understand the "also dilution air" phrase.

Pg 26, line 1: re-define "OxPR" here

Pg 26, line 13: "indicate" lost an "e"

Pg 26, line 28: change "effect" to "affect"

Figure 4: Define AMCH

Figures 5 & 7: move the y-axis to the edge of the graph (not at x=0) because it makes it difficult to read the y-axis values.

---

## Referee Comment (RC3) · Anonymous Referee #3 · 2 Jul 2018

The authors present seasonal measurements of monoterpenes, sesquiterpenes, and various oxygenated VOCs at a boreal forest site in Finland in 2011, 2015, and 2016. This study details summertime and monthly mean concentrations (April to Nov. 2016) of these species and monthly averages of their diurnal variabilities. The production of oxygenated species is investigated, and correlation of biogenic emissions with temperature are characterized. Lastly, the reactivities of these species with OH, O3, and NO3 are calculated.

This is a valuable dataset of underreported species including sesquiterpenes, small organic acids, and C6-C9 aldehydes in a forested environment. The paper is well or-

ganized, but the text will need to be carefully copy edited prior to publication. I recommend publication after the other detailed reviewer comments and the following points are addressed:

Section 3: Was there any dependence on humidity or an increase in MT or SQT emissions after rain events? Also, a time series and more complete summary of relevant statistics would be a great addition, even if it is in the supplement. P3 L17: Better to directly state that the GC's used in this study had technical difficulties rather than stating that all VOC measurements are "susceptible to technical failures." P4 L32: The use of "followed by" and "following" should be replaced by the more accurate terminology "characterized" and "measuring," respectively. P5 L1: Are the MT sum from GCMS2 presented? If so, a comparison to the individually summed MT from GCMS3 should be presented in the supplement. Also, with no ozone trap described for this instrument, I would suspect that the measurements will suffer from artifacts. P6 L10-15: Why is NO3 not included in these calculations? P7 L7: Equation (5) P9 L10: Avoid using "level" in place of the more accurate terms "mixing ratio" or "concentration." P11 L26: "trees" are listed twice P12 L9: Why was methyl vinyl ketone (MVK) not measured? P15 Table 1: Is MLH0-4 and MLH12-16 in local time? P24 L25 and P26 L30: I'm not sure that "deposition" is the correct term here. I think that "destruction" is the proper term.

---

## Author Response (AR1)

**Responses to author comments:**

**Anonymous Referee #1**

5

Thank you for the very good comments. We have considered them and we have improved our manuscript based on them as explained in more detail here:

- 10 This study focuses on the measurement of VOCs (speciated monoterpenes and sequiterpenes and oxygenated species) performed for a period of two years in a Boreal forest. The measured data are compared with temperature to highlights trends and derive simple proxies to estimate the VOC concentrations. In addition, by using O3 data and proxies to estimate OH and NO3 radical concentrations, the reactivity of each
- 15 class of compounds is calculated together with production rate of oxidation products. This work contains a large amount of data and measurements of speciated sesquiterpens which are extremely sparse and limited. The manuscript is suitable for publication in ACP after the authors have considered the following points:
- 20 1. A more detailed comparison with measurements already performed at the site should be included. During the HUMPPA-COPEC 2010 campaign, for example, measurements and fluxes of VOCs were performed with GC and enclosed to the trees branches and the results have been published (Eerdekens et al., 2009; Yassaa et al., 2012; Hens et al., 2014). Putting the measurements from this study in the contest of
- 25 what was previously observed at the same site would strengthen the conclusion of the manuscript. In addition, the results of the reactivity of the different VOC classes should be compared with a modelling study investigating the reactivity of VOC towards OH radicals in the specific site (Mogensen et al., 2011). A more structured comparison with previous measurements would be very interesting and could be included as a separate
- 30 section (instead of scattered around the different compounds as it is at the moment) by possibly reducing the intercomparison of all species with temperature which occupies a large fraction of the discussion and could be partly moved to the supplementary information.
- 35 -There are only very few earlier measurements at SMEARII of the compounds we have measured. Earlier measurements include PTR-MS monoterpene sum and some isoprene measurements (Eerdekens et al. 2009, Ruuskanen et al., 2009, Kontkanen et al. 2016 and Lappalainen et al. 2008). Hens et al., presented time series of monoterpene measurements, but there were no means or medians calculated to compare with. Yassaa et al. (2012) presented median values and diurnal cycles of isoprene, carene and α- and β-pinene in mid July to mid August. We added a supplement Table (S3) on
   40 these earlier measurements and more discussion on monoterpenes to the second paragraph of the section 3.1.1 and on
- isoprene to the section 3.1.3.

-We also added discussion on the study of Mogensen et al. (2011) to the manuscript into section 3.3.1

- 45 2. For the OH radical concentration determination, the authors refer to a publication by Petäjä et al. (2009). Unfortunately, this publication is not listed in the references but assuming they refer to the ACP paper, in that work there is no proxie described for the OH radical concentration. Proxies are given to determine the H2SO4 concentration. Therefore an explanation on how the OH radical concentration is calculated is needed.
- 50 In addition, although it is true that there is a direct correlation between OH radical concentration and photolysis of O3, a previous study which evaluated the OH radical budget at the same site (Hens et al., 2014) highlighted how the production from photolysis of ozone was actually marginal compared to other production paths. An error

analysis study on the determination of the OH radical should be included. For the NO3 radical concentration, a description of the formula is given (could be explicitly written as formula) but there is confusion regarding the value plotted in Fig. 3. The units for both OH and NO3 radical concentrations are cm-3 s-1 which is the unit for a production

5 rate. Assuming the plots is showing concentrations (and if this is the case, please fix the unit on the axis), is the NO3 concentration in the order of 0.5 pptv? How does it C2 compare with measurement of NO3 radicals performed at the same site (Liebmann et al., 2018)?

10 -Description on how OH radical concentration was calculated was added as well as reference to Rohrer and Berresheim (2006)

-Reference Petäjä et al. (2009) was added to the reference list

15 -Incorrect units of OH and NO3 mixing ratios were corrected in Fig 3.

-Measured NO3 in Liebmann et al (2018) was below the detection limit (1.3 pptv) all the time during their measurements in September 2016. Mean of calculated NO3 in September during our measurements was 0.12 pptv, which is also wellbelow their detection limit. We observed our 30 minute averages exceeding their detection limit only three times (N=220) during September.

- We think that error analysis of OH calculations should have been conducted for example in the publication of Petäjä et al. (2009), and is beyond the scope of our study. However, we added to the manuscript a comment on measured OH radical concentrations being clearly lower than estimated by this method as shown by Petäjä et al. (2009).

25

20

3. All the plots would be easier to follow if the x and y axis would be at the bottom and on the left of the figure, respectively (as for example figure 7b). Right now they are most of the time somehow in the figure making the reading of the numbers on the axis a bit difficult. The notation logarithm should also been added to the axis when

- 30 necessary. Do the fit performed take into account the errors on both concentrations and temperatures (York fit, (York et al., 2004))? As temperature measurements are relative accurate, the error on the x-axis could be ignored but the error on the concentration of the different species should be taken into account for a proper analysis.
- 35 -The axis of the figures were corrected and notes on the logarithmic scales were added. The values in the figure were not logarithms of the concentrations, only the scale in the y-axis was logarithmic. We added a note to the figure captions, whenever logarithmic axes were used.

-We now added uncertainties of the concentration values shown in the plots as error bars. Uncertainties were calculated
 as combined uncertainty of the measurement points using the propagation method. Measurement uncertainty was calculated as combined uncertainty given by the precision and systematic errors (calibration standard preparation and sampling flow).

The manuscript, before publication, requires some careful language check. In
 particular articles are missing and the structure of the sentences is often confusing.
 Some examples are listed in the technical comments but they do not cover the entire manuscript.

- A language check was done by a native speaker.

50

Technical comments:

Title. Suggest "Long-term measurements of VOCs highlight the importance of sesquiterpenes for the atmospheric chemistry of a boreal forest". -This is a good suggestion. We changed the title accordingly.

Page 2, Line 8. Citation needed at the end of "...forest".

-A reference to Guenther et al. (2012) was added

Page 2, Line 11. Citations needed at the end of "...studies".

10 -A reference to Mellouki et al. (2015) was added

Page 2, Line 14. More citations needed as examples of unmeasured BVOC.

15 -As stated in the review article by Yang et al. (2016) as well as in Sinha et al. (2010), Nölscher et al. (2012), and Praplan et al. (2018), cited in the manuscript, these unmeasured VOCs are not known and unmeasured fraction of BVOCs is based on the studies of total reactivity.

Page 2, Line 14. ". . . Therefore a better characterization of. . ."  $20\,$

-This has been corrected.

5

Page 2, Line 17. "...Once emitted, BVOCs readily..."

25 - This has been corrected.

Page 2, Line 20. Suggest adding the word paths after reactions at the end of the sentence.

30 - This has been corrected.

Page 2, Line 21. The expression "vary a lot" is for the spoken language. Suggest "...terpenoids are very different..."

- This has been corrected.

35

45

Page 2, Line 22. "... essential to understand biosphere-atmosphere interactions..."

40 - This has been corrected.

Page 2, Line 34. "...less sensitive instruments..."

- This has been corrected.

Page 3, Line 1. "... to be emitted by the pine trees. ..."

- This has been corrected.

50 Page 3, Line 4. The word lots can be substituted with the word many.

- This has been corrected.

Page 3, Line 16. ". . . and 2016 in a boreal forest at the SMEAR II. . . "

- This has been corrected.

5 Page 4, Line 16. "... in situ thermal-desorption gas..."

- This has been corrected.

Page 4, Line 24. A small description of the heated inlet, although already described in details in another publication, would be beneficial to confirm the ability of the system to measure sesquiterpenes.

- A more complete description of ozone removal was added.

15 Page 4, Line 32. Remove the hooks for internal standard.

- This has been corrected.

Page 5, Line 3. It is specified that an extra flow of 2.2 L min-1 was used. There is no indication of the flow drawn by the GC-MS2 (probably 1 L min-1 ?).

- The sampling flow of 30 ml/min is mentioned a couple of sentences earlier in the sampling description

Page 5, Line 14. "... This method has been..."

25 - This has been corrected.

30

Page 5, Line 16. "...A similar behavior..."

- This has been corrected.

Page 5, Line 17. "... isomerization to be reproducible..."

35 - This has been corrected.

Page 5, Lines 18-20. Sentence starting with "In our tests" and ending with "tube standards" is not clear and needs rephrasing.

40 - This has been rephrased.

Page 5, Line 20. ". . . Interconversion/degradation was not observed with the two other GC-MS used within this study. . . ".

45 - This has been corrected.

Page 5, Line 33. "What is identified by the authors as unknown sesquiterpenes? For which known sesquiterpenes was the instrument calibrated? C4

50

- The sentence 'Calibration solutions contained all individual studied compounds except for the SQTs there were only longicycene,  $\beta$ -farenesene,  $\beta$ -caryphyllene and  $\alpha$ -humulene.' was added to the method description.

Page 6, Line 19. There are way more recent studies (compared to a study from 1994) on degradation of monoterpenes and products yields which should be considered for the calculation of PR.

5

More studies are discussed in the results section and therefore reference to Hakola et al. (1994) was removed. There
are several recent studies available on reactions of MTs, but to our knowledge, yields for the compounds we studied
(nopinone and 4-AMCH) are not included. Especially for 4-AMCH, very few studies with yields are available.

- 10 Page 6, Line 30. It would be beneficial to have a little bit of background information on how the software derives rate coefficients for unknown VOC. Also, a list with all the VOCs species for which a derived rate coefficient was used (and the used rate coefficient) is needed.
- 15 We use the ChemiSpider's implementation of the software available online (www.chemspider.com). Description of this was corrected to the manuscript and the list of the used reaction rate coefficients was added as supplementary table S1.

Page 7, Line 15. The dependency of OH radical concentrations from UVB radiation was firstly showed by Rohrer and Berresheim (2006).

- More information on the calculation of OH mixing ratios as well as this reference were added to the manuscript.

Page 9, Line 24. Please add the reference to measurements performed during the 45 HUMPPA-COPEC campaign 2010 at the same site where monoterpenes were also

measured (see general comment 1).

- Our purpose here was to discuss other long-term measurements and therefore the sentence was rephrased to show this more clearly. However, HUMPPA-COPEC measurements are also referred now in the sections 3.1.1 and 3.1.3.

30

Page 10, Line 9. ". . . \_-pinene showed the highest concentration of the measured MT. . . "

- -This has been corrected.
- 35

Page 10, Line 25. Figure S1a) does not have any correlation plot but the mean diurnal variation and figure S1b) does not show any correlation factor equal to 0.85.

- The reference to Figure S1a) was removed.

40

Page 11, Line 2. "...Concentrations of SQTs did not increased during the sawmill episode in contrast with what observed for MTs...". This is a suggestion of what the meaning of the sentence could be.

45 -This has been corrected.

Page 11, Line 5. "...\_-caryophyllene showed the highest concentrations among the measured SQTs followed by longicylene, \_-farnesene and 4 unidentified SQTs detected only in the summer months (List of months)..."

- 50
- This has been corrected.

Page 11, Line 8. A new study from the Amazonia forest (Bourtsoukidis et al., 2018)

also shows large emissions of SQTs from soil.

- This has been added.

5 Page 11, Line 10. "...\_-farnesene, which was also detected in local..."

- This has been corrected.

Page 11, Line 15. Same point as for Page 10, Line 25.

- The reference to Figure S1a) was removed.

Page 11, Lines 16-17. The meaning of this sentence is not very clear. It is normal that sink terms (as much as production terms) will affect the concentration of a certain species. So what is the point the author is trying to make here?

- Based on the modelling study of Zhou et al. (2017), the dilution due to mixing has a much stronger effect on local MT concentrations than the chemical sink. For SQTs, the chemical sink is also important during the day and therefore the relative diurnal variation of SQTs was stronger than for MTs. We clarified this in the manuscript

20

10

15

Page 12, Line 9. ". . .but monthly averages of MACR. . . "

- This has been corrected.

25 Page 12, Line 10. The possible anthropogenic origin of MACR should be discussed in more details. One paper is cited which, interestingly, suggests that a large source of MVK and MACR comes from car traffic. Would this be a reasonable source for the site? Are there more studies on anthropogenic sources for MACR? Why was MVK not measured?

30

50

- MVK was not quantified due to a lack of calibration standard and low concentrations.

Page 12, Line 18. "... During the summer month its concentration was...".

35 - This has been corrected.

Page 12, Line 20. Please add the work by Kaminski et al. (2017).

**- This has been added.**

40 Page 12, Lines 20-21. What is the message of the sentence starting with "Further reactions..."? Do the authors want to compare the yields and discuss possible differences between different studies?

45 - This sentence was removed.

Page 17, Line 21. "... BVOCs correlates exponentially with temperature...".

```
- This has been corrected.
```

Page 18, Line 3. ". . . The high correlation with temperature observed indicates that temperature. . ."

**- This has been corrected.**

5

10

25

40

Page 18, Line 22. The linalool is missing from the legend of figure S2b) and/or the data for bornylacetate are missing.

- Linalool was added to the legend.

Page 19, Line 2. As most of the measured sesquiterpenes consists of \_-caryophillene, it is not surprising that the sum of SQTs shows a high correlation with temperature. . . . Please use the greek letter (and not b) to label \_-farnesene and \_-caryophyllene in the "

figures.

- This sentence and the corresponding figures were corrected.

15 Page 20, Lines 5-7. A little bit more discussion on the observed better correlation observed between temperature only and isoprene compared to temperature and light is needed as this is in contrast with what observed previously.

- We explanded the discussion on the significance of this observation. Due to the low concentrations and only small difference in correlations, we cannot still make the conclusions that concentrations would be only temperature driven.

Page 20, Last paragraph. This concept was already discussed previously and, as there is no additional discussion about possible anthropogenic origins or how this is reasonable for such a remote site, it does not add anything from where it was previously discussed.

- The sentence on the anthropogenic origin was deleted.

Page 22, Lines 2-6. The points made in this paragraph would probably not be so strong once a proper error analysis is introduced in the fit results. Though if, for example, a larger than expect concentration of MTs is observed in November and this is tentatively explained by anthropogenic origin of the monoterpenes, a more detailed discussion on which type of anthropogenic sources would produce which monoterpenes is needed.

35 - The word 'anthropogenic' was changed to 'close-by sawmill' to make the sentence more understandable. These sawmill emissions have been discussed in the manuscript for example in section 3.1.1.

- The sentence 'Then MT sum was correlating (Figure S7) a bit better with soil humus layer temperature (R2=0.87) than with ambient temperature (R2=0.80), which also indicates the soil related sources.' and figure S7 were removed since this small difference is not significant.

- As now shown by the Figure 5b, high values are just within the measurement uncertainty. This is now mentioned in the manuscript.

45 Page 23, Line 20. Please use \_ instead of a.

- This has been corrected.

- Page 23, Line 21. Figure 7 does not seem to depict the results obtained when using equation 10 but more when using equation 11. It would be interesting to see how the
- proxies developed in this study compare with measurement of VOCs from previous studies at the same site.

- The reference to equation 10 was corrected in Figure 5.

- We think that this kind of comparison would be of interest for a complete different manuscript, where different proxies and data sets could be compared properly. It would need the original datasets to get the 24 hour means from 8:00 to 8:00 and mixing layer height data for diurnal variations. LIDAR measurements have not been available earlier at SMEARII, which increases the difficulty of such comparison. In addition most of the earlier studies are only from short campaigns and would contain only very few 24-hour or monthly points.

- 10 Page 24, First 2 paragraphs. The addition of a table which includes the used rate coefficient for the different SQTs and MTs vs OH radicals, O3 and NO3 radicals would be beneficial for this section. The authors explain that SQTs have the largest contributions to the O3 reactivity despite the fact their concentration is \_50 times lower than the concentratrion of MTs. This is not surprising as the rate coefficient of \_-caryophyllene with
- 15 O3 (\_ 1e-14 cm3 s-1) is up to 3 order of magnitude faster than the rate with the main MTs measured at the site (\_-pinene, 9.4e-17 cm3 s-1, \_-pinene, 1.9e-17 cm3 s-1, and carene, 4.8e-17 cm3 s-1). Similarly, as the rate coefficient with OH for different MTs and SQTs only ranges less than one order of magnitude, it is expected that MTs, as they are present in larger concentrations, dominate the OH reactivity. The point being,
- 20 it is not much surprising, that the SQTs dominates the O3 reactivity but rather it is important to underline the large absolute concentration of SQTs observed.

- We added a table of used rate coefficients to the Supplementary Material (Table S1). Our main point here is that we are now able to measure also the concentrations of SQTs, but nevertheless, we think that by using these reactivities, we can show more clearly that even these low concentrations of SQTs are very important for the local atmospheric chemistry.

This is not directly seen by simply comparing MT and SQT concentration levels.

Page 25, Lines 5-6. Here it is a similar concept as before. Limonene and terpinolene both have relatively fast rate coefficient with OH radicals and O3 therefore, despite lower concentrations, they can have a large impact in the formation of secondary products.

- We rephrased the sentence.

35 Page 26, Line 29. Was the concentration of SQTs 30 or 50 times lower than the one of MTs? Anyway, as their reactivity depends on the product of concentration and rate coefficient, the sentence should be rephrase highlighting that due to the very fast rate coefficient of O3 with SQTs, a relatively (compared for example to monoterpenes) small concentration of SQTs can have a large impact on O3 deposition.

40

25

- SQT concentration was corrected to '50 times lower than MTs'.

- The sentence was rephrased.

45

**Anonymous Referee #2**

5

Thank you for the very good comments. We have considered them and we have improved our manuscript based on them as explained in more detail here:

This paper is represents many years of BVOC data at a boreal forest in Hyytiala, Finland.

The major advancement was detection and measurement of reactive sesquiterpenes, particularly b-caryophyllene, in ambient air. There is also quantitative evidence of BVOC oxidation products, carbonyls, alcohols, and acids coming from the forest. Analyzed data showed predictable temperature dependencies and contribution to oxidative capacity of the atmosphere as well as SOA production. This manuscript is very detailed and informative; I just have a few minor comments.

**15 Sampling and Calibration**

Because the major "breakthrough" is the ambient quantification of very reactive compounds, I am interested in the sampling techniques and calibration that were used. VOCs were calibrated using dilute liquid standards injected onto the adsorbent. Can you describe that a little further? What concentration ranges were

20 used? Was it injected using a syringe or put into an air flow? Is this representative of the sampling technique (using liquid vs. gas-phase compounds and not accounting for losses in the sampling system)?

- In our system we inject liquid standards into the tubes prior flusing them with a flow of clean nitrogen for 10 minutes to remove the methanol/water used as solvents. For most of the studied compounds, no gaseous standards are available

- 25 since they are not stable in the gas bottles. However, we have compared the gas and liquid standard methods for the main monoterpenes and aromatics and have optimized this way our liquid standard method, getting good results in the comparisons. For volatile organic acids (VOAs) the method has been compared with PTR-MS measurements in Hellén et al. (2017). We added a better description of the calibration method to the manuscript.
- 30 I understand the sampling of for GC-MS2 was a sub-sample from a larger flow (2.2 L/min). Please clarify the text; it took me awhile to figure this out. Part of the confusion is the use of "extra flow" terminology (pg 5, line 3)

- 'extra flow' was corrected to 'inlet flow' and the text was clarified.

35 Pg 5, line 14: "used method" does not make sense here

- It was changed to 'this method'.

Pg 5, line 19: "suffering by the most degradation" needs to be re-phrased.

- The sentence was rephrased.

Sampling for GC-MS3 used two different types of inlets. Is that correct? Why was that done? I do not understand "stainless steel tube was used to destroy ozone" (pg 5, line 25). How was ozone destroyed?

45

40

- In 2012 we published an article on this ozone destruction method (Hellén et al. 2012a). This is now better explained in the manuscript.

Pg 5, line 26 omit "a" before 40 mL/min

- 50
- This has been corrected.

**Content:**

The idea that p-cymene (4-isopropyl toluene) is partially anthropogenic can be mentioned sooner (pg 9, line 23) to explain why it has a different pattern.

**5 - We added a sentence about this at the suggested place.**

Pg 10, line 32: what is meant by the fact that MT data is more abundant? How is that different from the fact that there is "very little data on atmospheric SQT concentrations"?

10 - There is very little data on ambient concentrations and on emissions. These are two different type of data.

Pg 12, line 10: elaborate a bit on the anthropogenic sources of MACR.

- An explanation has been added.

Pg 17, line 4: What is LC-UV?

- It has been corrected to 'LC', which is actually explained earlier in the manuscript.

It would be nice to have a table outlining your major BVOC species, their reaction rate constants, and their vapor 20 pressures. In other words, outline the data used to make the graphs.

- The list of the used reaction rate coefficients and vapor pressures were added as supplementary table S1.

25

15

Figure 2: why is the propanic acid so high in June?

- The highest concentrations of all VOAs were measured in June together with the highest temperatures. VOAs were measured with GC-MS2 and the measurement period was different than for GC-MS3. This is explained in the manuscript in section 3.1.5.

30

Figure 5: There is a stronger correlation with monthly measurements vs. daily measurements of MT concentration and temperature. Why wasn't the monthly data included for SQT?

**35 - A figure including monthly mean SQTs has been added.**

**Wording**

Be sure to fix the grammar throughout this manuscript. Here are a few examples.

- 40 When referring to PTR-MS and GC-MS, be careful of the verb agreement. You can either use the abbreviation to represent the instrument (e.g. gas chromatograph) or the technique (gas chromatography). I think you chose the former, but then you need to ensure there are articles (e.g. "the", "a", etc.) before the abbreviation and a corresponding verb.
- 45 - A language check was done now by a native speaker for the whole manuscript.

Section 2.3: the text below the equations does not agree with the content of the equations. For example, "x" is not in the equation. I think all of the "ks" should be lowercase (reaction rate constants), but one was uppercase. Rephrase the explanation so that you don't use "yields are yields".

50

- This section has been corrected.

Pg 6, lines 28-29: rephrase this sentence; it is confusing

- This has been rephrased.

Pg 7, lines 12-13: rephrase sentence

**- This has been rephrased.**

Pg 9, line 22: insert a comma after "terpeniods"

**10 - This has been added.**

5

Pg 13, line 12: change "aq" to "a"

```
- This has been changed.
```

Pg 14, lines 9-13: re-work this sentence because it is wordy. I don't understand the "also dilution air" phrase.

- The sentence was rephrased.

20 Pg 26, line 1: re-define "OxPR" here

- We re-defined "OxPR" here.

Pg 26, line 13: "indicate" lost an "e"

- 'e' has been added.

Pg 26, line 28: change "effect" to "affect"

30 - This has been changed.

Figure 4: Define AMCH

```
- AMCH is now defined.
```

35

25

Figures 5 & 7: move the y-axis to the edge of the graph (not at x=0) because it makes it difficult to read the y-axis values.

- This has been corrected.

40

**Anonymous Referee #3**

Thank you for the very good comments. We have considered them and we have improved our manuscript based on them as explained in more detail here:

5

10

The authors present seasonal measurements of monoterpenes, sesquiterpenes, and various oxygenated VOCs at a boreal forest site in Finland in 2011, 2015, and 2016. This study details summertime and monthly mean concentrations (April to Nov. 2016) of these species and monthly averages of their diurnal variabilities. The production of oxygenated species is investigated, and correlation of biogenic emissions with temperature are characterized. Lastly, the reactivities of these species with OH, O3, and NO3 are calculated. This is a valuable dataset of underreported species including sesquiterpenes, small organic acids, and C6-C9 aldehydes in a forested environment. The paper is well or-ganized, but the text will need to be carefully copy edited prior to publication. I recommend publication after the other detailed reviewer comments and the following points are addressed:

15

25

30

35

Section 3: Was there any dependence on humidity or an increase in MT or SQT emissions after rain events? Also, a time series and more complete summary of relevant statistics would be a great addition, even if it is in the supplement.

Due to the strong effect of the mixing layer height, we were not able to detect any effects of the humidity. At this site the
 relative humidity also follows the similar diurnal cycle as the mixing layer height and MT concentrations. No correlation between daily means of relative humidity/rain events and MT or SQT concentrations were found either.

- We feel that showing the time series for this long and varying data is unnecessary and would produce only very unclear figures. However, the whole data set is available on request from the authors and we modified figure 1 to show 'box and whisker'-plots of different compound groups, so that the variability of the data is more visible. We also added a partial time series of MTs as a supplement figure S1.

P3 L17: Better to directly state that the GC's used in this study had technical difficulties rather than stating that all VOC measurements are "susceptible to technical failures."

- This has been corrected.

P4 L32: The use of "followed by" and "following" should be replaced by the more accurate terminology "characterized" and "measuring," respectively.

- This has been corrected.

P5 L1: Are the MT sum from GCMS2 presented? If so, a comparison to the individually summed MT from GCMS3 should be presented in the supplement. Also, with no ozone trap described for this instrument, I would suspect that the measurements will suffer from artifacts.

- Sampling times for GC-MS2 and GC-MS3 were different, so that a direct comparison is not possible. Nevertheless, we added times series of both instruments as a supplement figure S1.

45

- In our inlet test with 50 ppb of O3 (Hellén et al. 2012), no severe losses of MTs have been observed even though most SQTs were lost.

50 P6 L10-15: Why is NO3 not included in these calculations?

- We were unable to find any published yields of nopinone or 4-AMCH from the reactions with NO3.

P7 L7: Equation (5)

5 - This has been corrected.

P9 L10: Avoid using "level" innplace of the more accurate terms "mixing ratio" or "concentration."

- Whenever possible we replaced the term 'level'.

P11 L26: "trees" arenlisted twice

- This has been corrected.

15 P12 L9: Why was methyl vinyl ketone (MVK) not measured?

- MVK was not included in our calibration standards. However, it is not expected to have a high impact due to the very low emissions of isoprene at the site.

20 P15 Table 1: Is MLH0-4 and MLH12-16 in local time?

- Local winter time (UTC+2) is used throughout the manuscript. This is now properly mentioned in the manuscript.

P24 L25 and P26 L30: I'm not sure that "deposition" is the correct term here. I think that "destruction" is the proper term.

The first mention of "deposition" was changed to "destruction", but in the second instance, the whole sentence was modified based on the comment by the reviewer 1.

30

35

10

**Manuscript with corrections:**

Long-term measurements of volatile organic compounds highlight the importance of sesquiterpenes for the atmospheric chemistry of a boreal forest Sesquiterpenes identified as key species for atmospheric chemistry in boreal forest by terpenoid and OVOC measurements

10 Heidi Hellén1, Arnaud P. Praplan1, Toni Tykkä1, Ilona Ylivinkka2, Ville Vakkari1, Jaana Bäck3, Tuukka Petäjä2, Markku Kulmala2 and Hannele Hakola1

1Finnish Meteorological Institute, P.O. Box 503, 0011 Helsinki, Finland

15 2Institute for Atmospheric and Earth System Research (INAR) / Physics, Faculty of Science, University of Helsinki, Finland, P.O. Box 64, FI-00014 University of Helsinki, Finland

3Institute for Atmospheric and Earth System Research (INAR) / Forest Sciences, Faculty of Agriculture and Forestry, University of Helsinki, Finland, P.O. Box 27, FI-00014 University of Helsinki, Finland

20 Correspondence to: Heidi Hellén (heidi.hellen@fmi.fi)

Abstract. The cConcentrations of terpenoids (isoprene, monoterpenes (MTs), sesquiterpenes (SQTs)) and oxygenated volatile organic compounds (OVOCs, i.e. aldehydes, alcohols, acetates and volatile organic acids (VOAs)) were investigated during two-2 years at a boreal forest site in Hyytiälä, Finland, using in situ gas chromatograph-mass spectrometers (GC-MS).

25 Seasonal and diurnal variations of terpenoid and OVOC concentrations as well as their relationship with meteorological factors were studied.

Of the studied VOCs examined,  $C_2$ - $C_7$  unbranched\_volatile organic acids (VOAs) were found to haveshowed the highest concentrations, mainly due to their low reactivity. Of the terpenoids, monoterpenes (MTs) had showed the highest concentrations at the site, but also seven7 different highly reactive sesquiterpenes (SQTs) were also detected. The mMonthly

30 and daily mean concentrations of most terpenoids, aldehydes and VOAs were found to be highly dependent on the temperature. The hHighest exponential correlation with temperature was found for a SQT (β-caryophyllene) in summer. The diurnal variations of in the concentrations could be explained by sources, sinks and vertical mixing. TEspecially the diurnal variations of in MT concentrations were strongly affected by vertical mixing. Based on the temperature correlations and mixing layer height simple proxies were developed for estimating the MT and SQT concentrations.

To estimate the importance of different compound groups and compounds for thein local atmospheric chemistry, reactivity with main oxidants (hydroxyl radical (OH), nitrate radical (NO3) and ozone (O3)) and production rates of oxidation products

5 (OxPR) were calculated. The MTs dominated OH and NO3 radical chemistry, but the SQTs greatlyhad a major impact-on ozone ed  $Q_4^3$  chemistry, even though the concentrations of SQT were 30 times lower than the MT concentrations. SQTs were also the most important also for the production of oxidation productsproduction of oxidation products. Since the SQTs have showed high secondary organic aerosol (SOA) yields, the results clearly indicate the importance of SQTs for local SOA production.

10

**1. Introduction**

The boreal forest is one of the largest terrestrial biome in the world, forming an almost continuous belt around the northern Northern hemisphere/Hemisphere. It is characterized characterised by large volatile organic compound (VOC) emissions with 15 strong seasonal variations (Sindelarova et al. 2014). The boreal-Boreal zone Zone is estimated believed to be a major source of climate-relevant biogenic aerosol particles produced from the reaction products of primary emitted biogenic volatile organic compounds VOCs (BVOCs, Tunved et al., 2006). Isoprene, monoterpenes (MTs) and sesquiterpenes (SQTs) are the main reactive BVOCs emitted from the boreal forest (Guenther et al. 2012). They are known to influence particle formation and growth (e.g. Kulmala et al., 2013), the oxidation capacity of air-the atmosphere (Peräkylä et al., 2014) and chemical 20 communication by plants and insects (Holopainen, 2004). Oxidized Oxygenated VOCs (OVOCs) emitted from the vegetation include e.g. carbonyls, alcohols and volatile organic acids (VOAs), but their emissions are less studied and they are also produced in the air atmosphere from the reactions of VOCs (Mellouki et al., 2015). Studies on total reactivity in the atmosphere of boreal forests air have suggested the presence of highly reactive unmeasured BVOCs (Sinha et al. 2010, Nölscher et al. 2012, Praplan et al. 2018). [Also in other vegetation zones, the fraction of unmeasured BVOCs has also ve been very high (up 25 to 80%; Yang et al., 2016). Therefore, better characterization characterisation of BVOC emissions and concentrations in

forested areas is needed.

In the airOnce emitted BVOCs readily react with atmospheric oxidants, and the photochemical oxidation of even small organic compounds can lead to the formation of tens to hundreds of first generation products, which then undergo further oxidation and transformation (Glasius and Goldstein; 2016). Thus, it will probably never be possible to identify all oxidation products of all VOCs in the atmosphere. Therefore, detailed knowledge on the primary emitted compounds is crucial. The rReaction rates, reaction pathss and secondary organic aerosol (SOA) yields of different the various terpenoids vary a lot are very different (Lee et al. 2006, Ng et al. 2017) and compound specific concentration data are essential to an our-understanding of ing on biosphere-atmosphere interactions as well as local and regional atmospheric chemistry.

Proton transfer reaction mass spectrometers (PTR-MSs) are often used for measurements of fluxes or concentrations of MTs (Yuan et al. 2017) and there are already long data sets on ambient air concentrations of MTs measured by PTR-MSs even in boreal forest (Lappalainen et al. 2009, Kontkanen et al. 2016). PTR-MS measurements of SQT concentrations are not

- 5 commonhave often not been done, but there are some data available from the tropical forests (Kim et al. 2009, and 2010, Jardine et al. 2011). However, PTR-MSs areis not able to distinguishseparate different the various MTs or SQTs. Data on the concentrations of individual MTs measured by gas chromatograph-mass spectrometers (GC-MSs) are scarce and often available only from short measurement campaigns (Kesselmeier et al. 2002, Hakola et al. 2003, 2009, Jones et al. 2011, Yassaa et al. 2012, Jardine et al. 2015, Yanez-Serrano et al. 20172018, Jardine et al. 2015, Jones et al. 2011, Hakola et al. 2009 and
- 10 2003, Kesselmeier et al. 2002). Emissions of both MTs and SQTs have been studied at in various vegetation zones (Guenther et al. 2012), but to our knowledge there are only three studies published are available on the atmospheric concentrations of individual SQTs (Bouvier-Brown et al. 2009, Hakola et al., 2012 and Yee et al. 2018). Bouvier-Brown et al. (2009) measured the ambient air concentrations of MTs and SQTs in a ponderosa pine forest in California from 20th-of August until 10th-of October in 2007 with an in situ GC-MS and Hakola et al. (2012) measured MTs and SQTs in the air of a Finnish-boreal forest
- in Finland in 2011. However, due to losses in the inlets and less sensitive instruments, both studies were missing  $\beta$ -15 caryophyllene, which is the main SQT known to be emitted by-the pine treess (Hakola et al. 2006). Yee at et al. (2018) conducted SQT measurements in the central Amazonian rain forest for over four-4 months in 2014 and found 30 different SQTs. However, their site was located 2.5 km away from the forest and they concluded that the most reactive compounds had already reacted away before arriving to at the site. For example, they did not detect  $\beta$ -caryophyllene even though they were
- 20 able to find lots-many of its reaction products. Emission chamber measurements are-often suffering from losses of the most reactive SQTs on the chamber walls and inlet lines, and canopy-scale flux measurements are often not available due to the fast rapid reactions and low concentrations of SQTs. Therefore, ambient concentration data is are clearly needed to constrain the emissions of SQTs (Duhl et al. 2008). To our knowledge, we We report are here reporting what we believe are the first quantitative measurements of the ambient concentrations of  $\beta$ -caryophyllene, the main SQT emitted by the boreal forest trees (Hakola et al. 20062001, Hakola et al. 20012006). 25

30

Regarding oxygenated volatile organic compounds (OVOCs), studies on of the emissions of small compounds (e.g. methanol, acetone, acetaldehyde, acetic acid) have been conducted (e.g. Aalto et al. 20042014, Sindelarova et al., 2014). However but knowledge of the biogenic sources and concentrations of the larger volatile carbonyls, alcohols (C5-C10) and volatile organic acids (VOAs) is very limited.

In this study, the ambient air measurements of individual BVOCs and OVOCs were conducted in 2015 and 2016 in a boreal forest at the Station for Measuring Ecosystem-Atmosphere Relationships (SMEAR II) site in Hyytiälä with in situ gas chromatograph mass spectrometers (GC-MSs). We experienced technical difficulties In the VOC measurements, -are

Formatted: English (United States) Formatted: English (United States) Formatted: English (United States) susceptible to technical failures and even with the intensive campaign for this study, our data does not cover the whole entire measuring period continuously. To be able to parametrize parametrize the concentrations, to understand their sources and to fill the gaps in the data, we studied the dependence of the concentrations on environmental factors. As Since temperature is the dominant factor controlling the emissions of these BVOCs from trees at Hyytiälä (e.g. Tarvainen et al. 2005, Hakola et al. 2006, and 2017), the main focus was set on temperature dependence. Based on the temperature correlations simplified proxies for estimating local concentrations were developed. To estimate the importance of the individual VOCs or VOC groups for the local atmospheric chemistry and secondary organic aerosol (SOA) production, reactivities\_-and production rates of

oxidation products the production rates of oxidation products (OxPRs) were calculated.

**10 2. Experimental**

**2.1 Measurement site**

The mMeasurements were conducted in a boreal forest at SMEAR II-(Station for Measuring Ecosystem Atmosphere
 Relationships) in southern Finland. The SMEAR II station is a dedicated facility for studying the forest ecosystem-atmosphere relationships (Hari and Kulmala 2005). The measurement station is located in Hyytiälä (61°51'N, 24°17'E, 181 m a.s.labove sea level) in an eirea app\_55-year-old managed coniferous forest. The continuous measurements at the site include leaf\_, stand\_ and ecosystem-scale measurements of greenhouse gases, pollutants (e.g. ozone (O3), sulphur dioxide (SO2), nitrogen oxide (NOx)) and many different aerosol, vegetation and soil properties. In addition, a full suite of meteorological measurements is
 eurrently was collected.

The vegetation nearest vegetation to the measurement container is was a homogeneous Scots pine forest (*Pinus sylvestris L.*) forest; (>60 %)\_-where some birches (*Betula* sp.), aspen-poplar (*Populus* sp.) and Norway spruces (*Picea abies*) grow below the canopy. The canopy height is about ≈20 m with an average tree density of 1370 stems (diameter at breast height > 5 cm)
per hectare (Ilvesniemi et al., 2009). The understorey vegetation is formed by the different comprised of various shrubs, grasses and moss species. The most common shrubs are lingonberry cowberry (*Vaccinium vitis-idea* L.) and bilberry (*Vaccinium myrtillus* L.), the most common mosses are Schreber's big red stem moss (*Pleurozium schreberi* (Brid.) Mitt.) and a dicranium moss (*Dicranum* Hedw. sp.) and the most common grasses are wavy hair-grass (*Deschampsia flexuosa(L.)*, Trin.) and small cow-wheat (*Melampyrum sylvaticum* L.). Anthropogenic influence at the site is low. The largest nearby city is Tampere with 200 000 inhabitants. It is located 60 km to the south-west of the site.

**2.2 VOC Volatile organic compound measurements**

Ambient air measurements of VOCs were conducted in 2011, 2015 and 2016. Data from the year-2011 have beenwere published already-earlier by Hakola et al. (2012). The cConcentrations were measured with three different in situ thermaldesorptpion er-gas chromatograph-mass spectrometers (TD-GC-MSs), described hereafter as GC-MS1, GC-MS2 and GC-MS3. In 2015 and 2016 two different GC-MSs were used in parallel. The instruments were located in a container about 4 metres outside the forest in a gravel-bedded clearing. In 2015 and 2016 samples for the GC-MSs were taken at the a height of

1.5 m from an inlet reaching out eirea app. 30 cm from the container wall. In 2011 the GC-MS inlet was on the roof of a container at about 2.5 m.

The GC-MS1 was used for the measurements of isoprene and individual monoterpenes MTs in 2011 and May-July 2015. With

- 10 the GC-MS1 air was drawn through a 3m long stainless steel tube (outer diameter o.d. ¼ inch) at the a flow rate of 1 l min-1. The tFubes were heated to 120 °C to avoid losses of terpenoids. Stainless steel (grade 304 or 316) inlet line heated to 120 °C destroy O3 and losses for SQTs and MTs are negligible. Tthis method The heated inlet system, which also destroys ozone, ishns been described in detail by Hellén et al. (2012a). Removal of O3 from the inlet flow before collection of the sample is essential for avoiding losses of the very O3-reactive compounds (e.g. β-caryophyllene). VOCs in a 30-50 ml min-1 subsample
- 15 were collected in the cold trap of thermal desorption unit (ATD-400; PerkinElmer Inc., Waltham, MA, USAATD 400, PerkinElmer) packed with Tenax TA in 2011 and Tenax TA/Carbopack B in 2015 and analyzed in situ with a gas chromatographGC (HP 5890; Agilent Technologies Inc., Santa Clara, CA, USAHP 5890, Agilent Technologies) with DB-1 column (60m, inner diameter i.d. 0.25 mm, ft-film thickness 0.25 μm) and a mass-selective detector (HP 5972, Agilent Technologies). One 60-min sample was collected every other second hour. The dDetection limits were below 1 ppt for all
- 20 MTs. Measurements with the GC-MS1 have beenwere described in detail by Hakola et al. (2012). The system was calibrated using liquid standards in methanol solutions -injected on Tenax TA-Carbopack B adsorbent tubes and analysed between the samples using the offline mode of the instrument. The stability of the mass spectrometerMS was followed characterizedby using tetrachloromethane as an \_\_\_\_\_\_. The local concentration of tetrachloromethane in the ambient air is-was stable, and thus it was possible to detect sampling errors or shifts in calibration levels by following measuring its concentration.
- 25

The GC-MS2 was used for the measurements of  $C_5$ -- $C_8$  alcohols,  $C_2$ -- $C_7$  volatile organic acids (VOAs) and the MT sum in May--October 2015 and February--September 2016. Samples were taken every other second hour. In the 3 m long fluorinated ethylene propylene (FEP) inlet (1/8 inch I.Di.d.)-and, an extra-inlet flow of 2.2 L min-1 was used to avoid losses of the compounds on the walls of the inlet tube. The sSamples were collected directly-as sub-samples from this ambient air flow into

30 the cold trap (U-T1703P-2S, Markes International LTDLtd, Llantrisant, Wales, UK) of the thermal desorption unit (Unity 2 + Air Server 2, Markes International LTD, Llantrisant, UK). The sampling time was 60 min and the sampling flow through the cold trap 30 ml min-1. Heated stainless steel tubing (1 m) was used for O2 removal (Hellén et al., 2012a). Samples were analyzed analyzed in situ with a gas chromatographGC (Agilent 7890A, Agilent Technologies, Santa Clara, CA, USA) and a mass spectrometerMS (Agilent 5975C, Agilent Technologies, Santa Clara, CA, USA) connected to the thermal desorption

| -  | Formatted: Superscript |
|----|------------------------|
| -{ | Formatted: Subscript   |
| -{ | Formatted: Subscript   |
| -{ | Formatted: Subscript   |

unit. The polyethylene glycol column used for separation was the 30-m DB-WAXetr (J&W 122-7332, Agilent Technologies, Santa Clara, CA, USA) with an inner diameteri.d. of 0.25 mm and a film thickness of 0.25 µm. The system was calibrated using liquid standards in Milli-Q water (VOAs) and methanol (other VOCs) injected into adsorbent tubes filled with Tenax TA (60/80 mesh, Supelco. Inc., Bellefonte, PA, USA) and Carbopack B (60=/280 mesh, Supelco, Bellefonte, USA) and analysed

- 5 by the same method as samples. Liquid standards were injected into the clean nitrogen flow, which was flushed through the tubes for 10 min to remove the methanol/Milli-Q water used as a solvents. The stability of the mass spectrometerMS was followed by running gaseous field standards containing aldehydes and aromatic hydrocarbons after every 50th sample taken and using tetrachloromethane as an "internal standard", Used This method has beenwas described in more further detail by Hellén et al. (2017). Due to the inter-conversion observed inter conversion between the MT isomers inside the instrument
- 10 presumably during the pre-concentration step in the thermal desorption unit, only the MT sum is reported. SSimilar-imilar behaviorbehaviour has beenwas observed by Jones et al. (2011). However, in contrastry to their observations we did not findobserved that isomerization-isomerisation to bewas not repeatablereproducible. They Jones et al. (2011) also mentioned that they were able to detect β-pinene in the standards, but it was not detected in the ambient samples. In our tests, interconversion was highest for β-pinene was suffering by the most degradation of all studied MTs and higher inter conversion 15 was observed after running several ambient samples than directly after running adsorbent tube standards.
- 13 was observed after running several ambient samples than directly after running adsorbent tube standards. Interconversion/degradation was not observed with the other two GC-MS used in this study With two other GCMSs, which we used, inter-conversion/degradation was not detected.
- The GC-MS3 was used for the measurements of individual MTs, SQTs, isoprene, 2-methyl-3-butenol (MBO) and C5-C10
  aldehydes in April-November 2016. With With the GC-MS3 air was drawn through a 1m long fluorinated ethylene propylene (FEP) inlet (1/8 inch i.dLD.) and 1 m long stainless\_steel tubinge (o.d. ¼ inch) at the a flow rate of 11 min-1. Stainless steel tubeings heated to 120 °C was used to destroy ozoneO2 and was heated to 120 °C to avoid losses of terpenes. The O3 removal method wass described in further detail in (Hellén et al. (2012a). VOCs in a-40 ml min-1 subsamples were collected for 30 minutes in the cold trap (Tenax TA/carbopack B) of the thermal desorption unit (TurboMatrix, 650, Perkin-Elmer) connected
- 25 to a GC (Clarus 680, Perkin-Elmer) coupled to an mass-spectrometer MS (Clarus SQ 8 T, Perkin-Elmer). An HP-5 column (60 m, id. 0.25 mm, film thickness 1 µm) was used for the separation. The system was calibrated using liquid standards in methanol solutions injected into a clean nitrogen flow, flushed through theon Tenax TA-Carbopack B adsorbent tubes and analysed between the samples using the offline mode of the instrument. The stability of the mass spectrometerMS was followed by running one adsorbent tube standard after every 50th sample-taken and uusing tetrachloromethane as an "internal standard".
- 30 The calibration solutions contained all the individual compounds studied except for thesome SQTs. there were oOnly longicycene, β-farenesene, β-caryphyllene and α-humulene are included in the calibration standards. Unknown SQTs were calibrated using the responses of β-caryphyllene.

| -{ | Formatted: Superscript |
|----|------------------------|
| 1  | Formatted: Subscript   |
| 1  | Formatted: Subscript   |

Of the used-instruments used, the GC-MS3 was the most sensitive, and it was able to detect very low concentrations of SQTs, much more than the GC-MS1, and therefore only the 2016 data is used in SQT data analysis-we used only the 2016 data. Monthly means were calculated from all the available data for each month (see number of data points in Table 1 and 2). The dDaily means were calculated for days with no missing data points starting at 8:00 (UTC+2) and ending at 8:00 (UTC+2) the next day.

**2.3 Calculation of formation rates of measured reaction products of MTsmonoterpenes**

For studyingTO determine the diurnal variation of in the measured reaction products of MTs net formation rates (NFRsRF)
were calculated. Production rate (PR), destruction rate (DR) and NFR of reactions products of MTs can be described by the equations below:

$$PR = \frac{d[product]}{dt} = k_{0H+MT}[MT][0H] \times yield + k_{0_3+MT}[MT][0_3] \times yield$$
(1)

$$\quad DR = \frac{-d[product]}{dt} = -\frac{\kappa}{k_{0H+product}}[product][0H] - k_{0_3+product}[product][0_3]$$
(2)

$$NFR = PR + DR$$
(3)

where k\* is the reaction rate coefficient of the MT or product with oxidant (hydroxyl radical (x=OH) or O3) and, [MT, product,
 or xOH or O3] is the concentration of the corresponding MT, product or oxidant. In addition and yields are, the yields of the products from the corresponding reactions were used obtained from Hakola et al. (1994).

**2.4 Reactivity calculations**

25 The total reactivity of the VOCs ( $R_x$ ) was calculated by combining their respective concentrations (individual VOC (VOC*i*)) with the corresponding reaction rate coefficients ( $k_{i,x}$ ).

$$R_x = \sum [VOC_i] k_{i,x}$$

5

30 This determines in an approximate manner the relative role of compounds or compound classes in local OH, nitrate radical (NO3) and O3 chemistry. The experimentally determined reaction rate coefficients listed earlier The reaction rate coefficients for OH reactions are the same as used by Praplan et al. (2018), for O3-reactions by Hakola et al. (2017) and NO3 reactions by by Ng et al. (2017). were used. When the experimental reaction rate coefficients were not available, they were estimated with the

(4)

values from ChemSpider database (www.ChemSpider.com; the Royal-Society of Chemistry) estimated by -the AopWinTM module of the EPITM software suite (https://www.epa.gov/tsca-screening-tools/epi-suitetm-estimation-program-interface, EPA, U.S.A) as implemented online by the ChemSpider (www.ChemSpider.com; the Royal Society of Chemistry) were used. The estimation method used by AOPWIN is based upon the structure-activity relationship. UsedAll reaction rate coefficients are listed in the sSupplementary Material (Table S1). For the unknown SOTs an-average reaction rate coefficients (kOH=1.55

5 are listed in the sSupplementary Material (Table S1). For the unknown SQTs an-average reaction rate coefficients ( $k_{OH}$ =1.55 x 10-10 cm3 s-1,  $k_{O3}$ =6.1 x 10-15 cm3 s-1,  $k_{NO3}$ =8.72 x 10-11 cm3 s-1) of the known SQTs were used. Due to the lack of measured or estimated reaction rate coefficients, these average values were also used also for longicyclene in the O3 and NO3 reactions and for β-farnesene in the NO3 reactions.

**10 2.5 Calculation of total production rates of oxidation products**

Production rates for oxidation products (OoxPRs) were calculated for the reactions of different the various VOCs with the OH radicals, O3 and NO3 radicals by the Eq. (65).

15
$$OxPR = \frac{d[products]}{dt} = \sum [VOCi] (k_{OH+VOCi}[OH] + k_{O_3+VOCi}[O_3] + k_{NO_3+VOCi}[NO_3])$$
 (5)

where  $k_{+}$  is the reaction rate coefficient of a VOC with an oxidant (OH, O3 or NO3) and [VOCi, OH, O2 or NO2] is the concentration of the corresponding VOC or oxidant. Unknown SQTs were not taken into account in the calculations. Of the sesquiterpenes-SQTs the reaction rate with NO3 radicals was found only for  $\beta$ -caryophyllene and while the reactions of other SQTs were not considered in the calculations.

Since hydroxyl OH radical concentrations were not measured directly, proxies were calculated from the ultraviolet B (UVB) radiation intensity (Eq. 6), which is known to correlate strongly with OH radicals as first described by Rohrer and Berresheim (2006) and later evaluated by the observations at SMEAR II by Petäjä et al. (2009) and Hens et al. (2014).

(6)

25

30

20

 $[OH]_{proxy} = 5.62 \times 10^5 \times UVB^{0.62}$

Comparison with the measurements has shown that even though the variation in concentrations was quite similar, this method results in concentrations three times higher than those of the measurements (Petäjä et al. 2009). Therefore, the OxPRs of the OH radical reactions were actually expected to be lower than those presented in this study.

The NO3 concentrations were calculated by assuming a steady-state by its production from  $O_3$  and  $NO_2$  and removal by photolysis and oxidation reactions as described by Peräkylä et al. (2014). The only modification <del>compared to Peräkylä et al.</del>

| 1  | Formatted: Not Highlight |
|----|--------------------------|
| -  | Formatted: Not Highlight |
| Υ  | Formatted: Not Highlight |
| ľ  | Formatted: Not Highlight |
| () | Formatted: Not Highlight |
| Ì  | Formatted: Not Highlight |
| () | Formatted: Not Highlight |
|    | Formatted: Not Highlight |

(2014) was that the data for individual MTs were used and  $\beta$ -caryophyllene (main SQT at the site) was also considered as an additional sink.

The aAerosol surface area needed for the calculation of the NO3 concentration was derived from the aerosol number size

- 5 distribution in the range 3--1000 nm at the SMEAR II. It was obtained, using two parallel differential mobility particle sizers (DMPS) (Aalto et al., 2001). Each DMPS system consisteds of a Hauke-type differential mobility analyzer analyser (DMA) and condensation particle counter (CPC). Each DMA separates separated the sampled aerosol particles according to their electrical mobility, and the particles selected particles are transported to the corresponding CPC, which grows grew them by condensing butanol on their surface, and counted their number with optical methods. Particles with different electrical mobilities can be selected and counted. The first DMPS measures-measured particles with sizes between 3 nm and 10 nm and the second one-between 10 and 1000 nm. In cCombining the spectra, the number size distribution of the whole entire size range is was reached. One measurement cycle scanning all the sizes takes required about 10 minutes. Charging the aerosol population to an equilibrium charge distribution with a bipolar charger enables enabled the measurements of both
- 15 neutral and charged particles.

The number of particles in a unit volume in certain size range can be determined as

$$N(D_p) = \int_{D_p}^{D_p + dD_p} n(\log D_p) d\log D_p \tag{67}$$

where  $n(\log D_p)$  is the number density distribution representing the number of particles between diameter  $D_p$  and  $D_p + d D_p$ per unit volume. When assuming that is all the particles are assumed to be spherical, the surface area distribution become

$$s(log D_p) = n(log D_p) \cdot \pi D_p^2 \tag{78}$$

Then the surface area of the particles in the range  $D_p - D_p + d D_p$  obtained likewisely similarly as

$$S(D_p) = \int_{D_p}^{D_p + dD_p} s\left(\log D_p\right) d\log D_p \tag{89}$$

30

**2.6 Complementary measurements**

The Mmeteorological data,  $O_3$ , NO and NOx concentrations were obtained from SmartSMEAR AVAA-portal (Junninen et al., 2009, https://avaa.tdata.fi/web/smart). All the data used in this study are the one ose collected at the a height of 4.2m from the mast inside the forest, except for the temperature, which was collected at 125m for comparison.

- 5 The mMixing layer height (MLH) was estimated from measurements with a Halo Photonics Stream Line scanning Doppler lidar, which is a 1.5 µm pulsed Doppler lidar with a heterodyne detector (Pearson et al., 2009). The rRange resolution of the lidar is 30 m and the minimum range of the instrument is 90 m. Operating specifications of the lidar are given in supplementary Table S1S2. The wind profile aAt Hyytiälä wind profile was obtained from a 30° elevation angle conical scan, i.e. from a vertical azimuth display (VAD) scan. This VAD scan was configured with 23 azimuthal directions and integration time of 12
- 10 s per beam. A vertical stare of 12 beams and integration time of 40 s per beam was-were configured to follow the VAD scan. The VAD scan and 12-beam vertical stare were scheduled every 30 min at Hyytiälä; other scan types operated during the 30min measurement cycle were not utilized\_utilised in this study. The lidar data was corrected for a background noise artefact according to Manninen et al. (2016). After this correction a signal-to-noise-ratio threshold of 0.001 was applied to the data.
- 15 The t‡urbulent kinetic energy (TKE) dissipation rate was calculated from the Doppler lidar measurements according to the method by O'Connor et al. (2010). A VAD-based proxy for turbulent mixing (σ2VAD) was calculated from the 30° elevation VAD scan according to the method by Vakkari et al. (2015). The MLH was determined from the TKE dissipation rate and the VAD scan in am manner similar to that of Vakkari et al. (2015). Briefly, first a constant threshold of 10-4 m2 s-3 was first applied to the TKE dissipation rate profile, i.e. the MLH was taken as the last range gate where the TKE dissipation rate was higher than 10-4 m2 s-3. If the TKE dissipation rate was below the threshold value at the first usable gate at 105 m above ground level (a.g.l.), i.e. MLH < 105 m, the σ2VAD profile was used to identify the MLH. For σ2VAD a constant threshold of 0.05 m2 s-2 was applied to determine the MLH (Vakkari et al., 2015). With this approach, the MLH could be identified from 60 m a.g.l. to >2000 m a.g.l.; rainy periods were excluded from the analysis. Values below 60 m were marked as 0 m.

**25 3. Results and discussion**

**3.1 Seasonal and diurnal variations of in concentrations**

The concentrations of most compounds measured with three different GC-MS instruments in 2011, 2015 and 2016 are at the same levelwere similar (Tables 1 and 2). Of the compounds measured the eompounds-VOAs had-showed the highest concentrations during all months (Fig.ure 1, Table 1 and 2). Also The 1-butanol and isopropanol concentrations were also high, most likely because they are-were used in some instruments for aerosol measurements at the site. Even though the concentrations of the terpenoids are were not as high as those of the VOAs due to their high reactivity, they are were expected to have show the greatest highest impacts on local chemistry. For most of the compounds studied compounds daily and monthly mean concentrations were highest during the warm summer months. For aromatic hydrocarbons, which are mainly emitted from anthropogenic sources, the concentrations were higher in winter.

The rRelative diurnal variation of in most compounds was were highest in June when mixing layer heights the MLHs were highest (Figs.ure 2 and 3). The cConcentrations of different the various compounds and compound classes are described in more further detail in following sections.

**3.1.1 Concentrations of monoterpenes**

- 10 The MTs Of the terpenoids MTs hadshowed the highest concentrations of the terpenoids, with the mean MT SUM sum being 400, 440 and 430 pptv in summers (July-August) 2011, 2015 and 2016, respectively (Table 1). All the MTs except p-cymene had-showed a-clear maximum-maxima in summer. PpP-Cymene is also known to have anthropogenic sources (Hakola et al. 2012). The 4-variations in the MT sum measured with the GCMS2 and CGMS3 were similar (Figure S1), but their direct comparison was not possible, due to the different sampling times.
- 15

Long-term MT concentration measurementss havere previously ve been measured conductedearlier at this boreal forest site also with PTR-MSs (Lappalainen et al. 2009, and Kontkanen et al. 2016). Theeose PTR-MS measurements were conducted close to the forest canopy at the heights of 14 m (2006–2009) or 16.8 m (2010–2013). The median July MT concentration measured between 2006 and 2013 was 382 ppty. In previous studies in March 2003, 2005 and 2006, the MT concentrations

- 20 measured in short campaigns with PTR-MSs have been higher thatn in our measurements in April 2016 (Table S3). Yassaa et al. (2012) measured concentrations values-lower than our in a campaign in July--August 2010. Large spatial differences in concentrations especially for terpenoids are-were expected depending on the sampling point at the site (Liebmann et al. 2018). In our study the sampling site was at the edge of the forest whereas in the-previous studies by Lappalainen et al. (2009) and Kontkanen et al. (2016) it was above the canopyin the upper canopy level in the middle of the stand and in-Yassaa et al. (2012)
- 25 above at the canopy at a height of 24m, where it is expected due to transport and chemistry that concentrations are lower compared to our measurements.

In our measurements, the MT concentrations had showed high peaks in May 2016 (12th-12 May at 3:04am and 14th-14 May at 4:10am\_-6:10am), 1st of June at 1:10am and 9th-of September at 23:355pm, which were clearly deviateding from the other data. Based on the wind directions, it is possible that these peaks may have been are due to high concentrations coming from

30 data. Based on the wind directions, it is possible that these peaks may have been are due to high concentrations coming from the site of operations of a sawmill, a woodmill and a pellet factory in Koreakoski, 5 km southeast of Hyytiälä. The influence of this factory on monoterpene the MT concentrations have has also been observed also in earlier studies of previously (Eerdekens et al. (2009), Liao et al. (2011), Williams et al. (2011) and Hakola et al. (2012)). These samples were not used in the further analysis.

Of the MTs-α-pinene Ppinene had clearly the highest concentrations showed the highest concentration (50% of the MT sum) of the measured MTs (50% of the MT sum) followed by  $3\Delta_a^3$ -carene, β-pinene and limonene (Figure 1 and Table 1). The MT distribution was very similar for nighttime (photosynthetically active radiation (PAR) <\_50 µmol s-1 m-2) and daytime (PAR

5 >\_50 µmol s-1 m-2)-values, only 1,8-cineol and linalool had showed a bitslightly higher fractions during the day. A sSimilar MT distributions haswashave also been observed at the site (Hakola et al. 2012, Yassaa et al. 2012) observed and here resembles the one of the emissions of local trees (Bäck et al. 2012, Hakola et al. 2006 and 2017).

The dDiurnal variability in theof MT concentrations at the site wasis driven by the vertical mixing; low values were measured during the day when mixing was highest and the highest values during nights with the lowest mixing (Figures Figs. 2 and 3). This has been observed also in earlier studies of MTs at this boreal forest site (Hakola et al. 2012 and Kontkanen et al. 2016 and Hakola et al. 2012). Similar diurnal variation was found by Bouvier-Brown et al. (2009) in a ponderosa pine forest in the Sierra Nevada Mountains of California. However, this observation of MTs is opposite in contrast to the diurnal variation of in MT concentrations measured in the Amazon tropical rain forest by Yanez-Serrano et al. (2017). Light dependent emission found in the Amazon rain forestia (Jardine et al. 2015) could explain this. In boreal forests emissions are strongly temperature\_

dependent and may also continue <del>also</del> during nights with only <del>with</del> lower rates if temperature is sufficiently high <del>enough</del> (e.g. Hakola et al. 2006).

The dDiurnal variation of in concentrations was highest in June concomitant with the highest variation of mixing layer heightthe MLH (Figure Figs. 2 and 3). Mean mixing layer heights The MLHs during the day (at 12:00-16:00) in June and July

- 20 heightthe MLH (Figure Figs. 2 and 3). Mean mixing layer heightsThe MLHs during the day (at 12:00-16:00) in June and July were 1605 and 819 m, respectively. w. While during the night (at 00:00-4:00) mean mixing heights in June and July they were 87m and <\_60 m, respectively. Also The monthly mean mixing layer heightMLH, which roughly describes the mean dilution volume of the emissions, was two-also twice astimes higher during our measurements in June than in July (Table1). Since the lidar that was used for the measurements of mixing layer heightsthe MLHs, is notwas unable to detect mixing layer
- 25 heightsMLHs <60 m, we also used also temperature difference between the heights 125 m and 4.2 m to roughly describe the vertical mixing. The correlation of monthly mean diurnal variation of in MT sum concentrations with temperature differences at the site was high (R2MT=0.85 in July, Figure S1a). [Also individually measured values\_also showed had-relatively good favourable correlation with temperature difference (R2MT=0.46 in July, Figure S1bS2b). 1,8-cineol\_Cineol and linalool were did not following this general diurnal pattern of MTs giving indicatingon of different sources. 1,8-cineol\_Cineol is the only
- 30 MT which is known to have also show clearly light-dependent emissions from Scots pines growing at the site (Hakola et al. 2006).

**3.1.2 Concentrations of sesquiterpenes**

At the moment there is very littleFew data are available on atmospheric SQT concentrations and also emission data are also is much sparser than for MTs. In our measurements SQTs had similar showed seasonal variation similar to that of the as MTs, but their concentrations were much lower (Table 1). SQTs are very reactive and therefore their contribution to the local chemistry can still be significant. The highest 30-\_minute mean for the SQT sum (103 pptv) was detected on 25th-of-July at

5 3:15am coinciding with high temperature and a shallow mixing layer. The concentrations of SQTs did not increased during the sawmill episode in contrast to that observed for MTs.SQTs were not increased during the sawmill episodes of MTs. They are were more reactive and, if emitted, they are were probably depleted during the transport from the sawmill to the site.

Of the SQTs β-caryophyllene Ccaryophyllene showedhad the highest concentrations among the SQTs measured followed by
 longicylene, β-farnesene and four unidentified SQTs detected only in July and August, but in summer also longicylene, β-farnesene and 4 unidentified SQTs were detected. ((TableFigure 1). Night-time (PAR<50 µmol s-1 m-2) and daytime (PAR>50 µmol s-1 m-2) distributions of SQT concentrations were very similar, only β-farnesene had ashowed slightly higher fraction during the day. β-caryophyllene is emitted by the-local pines and spruces (Hakola et al. 2006 and, 2017) as well as from the forest floor (Hellén et al. 2006, Mäki et al. 2017, Bourtsoukidis et al., 2018). Aaltonen et al. (2011) and Mäki et al. (2017)
 have also detected longicyclene in forest floor emissions. Laboratory studies have shown stress related emissions of β-

- farmesene (Petterson 2007, Blande et al. 2009, Petterson 2007), and β-farmesene was also detected it has been detected as well in local spruce emissions by Hakola et al. (2017).
- The diurnal variation of in most SQTs was similar to the variability variation in of MTs, and the concentrations were largely driven by the vertical mixing (Figure Figs. 2 and 3). As for the MTs correlation of the monthly mean diurnal variation of in the SQT concentrations with temperature difference between the heights of 125 m and 4.2 m was high (R2SQT=0.90 in July7 Figure S1a) and alsowhile individual measured values had also showed relatively favourablegood correlation with the temperature difference (R2SQT=0.48 in July, Figure Fig. S1bS2b). Based on the modelling studies by Zhou et al. (2017) also in addition to mixing a higher chemical sink than for MTs during the day may have an effectaffect -onlocal SQT concentrations
- 25 during the day. This was supported here by the higher relative diurnal variation of in SQTs compared tothan in MTs. The only exception was, β-farnesene, which had-showed almost as high concentrations during the day as in theat night, indicating different sources than for other-different SQTs and MTs. An opposite constracting diurnal variation was found by Bouvier-Brown et al. (2009) at in a ponderosa pine forest in California and they suggested suggesting that the sources of β-farnesene is havingare mainly light dependent sources.

30

**3.1.3 Isoprene and 2-methyl-3-buten-2-ol concentrations**

The iHsoprene and 2-methyl-3-buten-2-ol (MBO) concentrations were low (Table 1). Low concentrations of isoprene have also been observed in previous studies (Table S3). In our study, the mMonthly means in 2016 were 0.3-18 pptv for isoprene and 0.1\_-30 pptv for MBO.\_ Low levels were expected since the main local trees-trees (Scots pine and Norway spruce) are MT emitters and have show only minor emissions of isoprene and MBO (Tarvainen et al. 2005, Hakola et al. 2006 and 2017). The hHighest daily means were measured in July and August together with MTs and SQTs. The emissions of ilsoprene is known to haveare light\_light\_dependent emissions (Ghirardo et al. 2010) while MBO emissions from local trees are mainly temperature temperature-dependent (Hakola et al. 2006).

The diurnal variation of in MBO was coincideding with the variation inof MTs2 with high values during the night and low values during the day. This was expected, due to the temperature\_dependent emission of MBO. For isoprene4 clear change5 in diurnal variation wereas observed between early summer (April-June) and late summer (July\_-September) (Fig\_ure 2). In May and June when emissions are still low due to the early growing season, lower daytime values were detected, but in July and August7 the daytime concentrations were clearly higher due to high light\_dependent emissions. Previously, the gradually increasing isoprene emissions have been associated witheonnected to the foliage growth period, and start when the effective temperature sum (ETS) reaches a threshold value. For example, ., e.g. in tea-leafed-leaved willow (*Salix phylicifolia* L.), the lower emissions of isoprene were found when the ETS < 400 degree days (Hakola et al. 1998). During our measurements in 2016, the ETS reached a value 400 on 23e4 -of-June.

**3.1.4 Concentrations of reaction products of terpenes**

5

- Methacrolein (MACR) is a reaction product of isoprene, but the monthly means averageof MACR concentrations didare not
   following the concentrations of its precursor isoprene (Table 1). MACR is known to have alsomay also have anthropogenic sources (Biesenthal and Shepson, 1997) and in spring and autumn, when lifetimes are longer and biogenic emissions lower than in summer, anthropogenic influence is was expected to be higher. In their studies Biesenthal and Shepson (1997) found that the MACR concentrations near Vancouver were not explained by the photochemical source, while in Toronto MACR correlated with carbon monoxide (CO) suggesting they originate in traffic emissions. In our study, the monthly mean
   concentrations of MACR (4.8 and 3.3 ppt, respectively) were ca. 30% of the isoprene concentration in July and August when

- 2.5 concentrations of MACR (4.8 and 3.3 ppt, respectively) were ca. 30% of the isoprene concentrations of MACR (4.8 and 3.3 ppt, respectively) were ca. 30% of the isoprene concentration. This is elose similar to the yields of 25 % and 24 % measured in chamber experiments by Paulson et al. (1992) and Atkinson (1994), respectively.
- 30 Nopinone is a reaction product of β-pinene and its monthly mean concentration followed the variation of thein MTs (Table 1). During the summer months its concentration was 7--13 % of the concentration of its precursor β-pinene. In reaction chamber studies, the yields of nopinone in ozoneQ2 reactions of β-pinene have been 19--23 % (Grossjean et al. 1993a; Hakola et al. 1994 and-, Winterhalter et al. 2000) and in OH radical reactions 25--30-37 % (Calvert et al. 2011, Kaminski et al. 2017). Further

reactions of the products affect also the ambient air concentrations and therefore chamber yields are not directly comparable to the concentrations.

The mMean diurnal variation of in nopinone was followeding the variations of in its precursor ( $\beta$ -pinene) in April-\_June, but

- 5 in July-\_September high values were also observed also during the day (Figure 4). Nopinone is known to be produced both from in OH radical and O3 reactions of  $\beta$ -pinene (Hakola et al. 1994), but it is destroyed only in the OH radical reactions. Since the yields from the NO3 radical reactions are not available from in the literature, they cannot be considered. The NO3 reactions would increase the production especially at during the night. DAlso deposition may also have an effect (Zhou et al. 2017), but it was not taken into account here. The Production rate (PR), destruction rate (DR) and net formation rate (NFR) of nopinone
- 10 were calculated, using the Eqs.equations 1-to-3. The nNopinone yields used for OH radical and O3 reactions obtained from Hakola et al. (1994) were 0.27 and 0.23, respectively.
- Change in the nopinone diurnal variation is explained by the balance between its sources and sinks. The cConcentrations closely followed elosely the NFR variation (Fig.gure 4). Nopinone is a rather stable molecule, and has 5 times lower OH radical reactivity than  $\beta$ -pinene and in contrast to  $\beta$ -pinene, it is does not reacting with O3. The rResults indicate that in May
- 15 and June, when there is was already high light intensity and high OH radical concentrations, but the emissions of β-pinene are were still low due to lower temperatures and an early growing season, the nopinone produced may have nopinone reacteds away during the day, while and higher values are were measured during the night, when there are no OH radicals, but nopinone is still produced from O3 reactions of β-pinene. In July and August higher emissions and faster-more rapid reactions of β-pinene with OH radicals results resulted in higher daytime concentrations of nopinone. In September the emissions are were
- 20 already lower, but also the OH radical concentrations and MLHs the mixing layer heights were also are lower, while and higher nopinone concentrations wereare still detected during the day.

Reaction product of limonene, 4-acetyl-1-methylcyclohexene (4-AMCH), had-showed\_very low concentrations and was detected only in June and July (Table 1). The NFR for 4-AMCH was calculated by the same methods as for nopinone using
the equations Eqs.1--to-3. The rReaction rates of 4-AMHC-AMCH with the OH radical and O3 wereare only 25 % and 20 % lower than for its precursor. The 4-AMCH yields used here for the OH radical and O3 reactions obtained from the Hakola et al. (1994) were 0.20 and 0.04, respectively. In a study by Grosjean et al. (1993) the yield from the O3 reaction was 0.02. The concentrations measured concentrations in the present study did not follow the diurnal variation of in the NFR especially in July, when the highest concentrations were measured (Figure 4). However, production rate (PR) had showed aq similar diurnal

30 pattern similar as to that of the concentrations. Studies on limonene reactions by Grosjean et al. (1993) and Hakola et al. (1994) did not take into account that 4-AMCH reacts almost as rapidlyfast with the oxidants as limonene and that the real yields could have been higher. When we increased the yields in our calculations, better agreement wascould be achieved. In fig.ure 4d the yields for the OH radical and O3 reactions were increased by factors of 2 and 3, respectively. In August, the sensitivity of the

instrument was less than 50 % of the sensitivity in June/July, due to faultya worse tuning of the MS and 4-AMCH was not detected even though calculations-would indicated higher concentrations than in June.

**3.1.5 Concentrations of volatile organic acids**

5

The VOAs showed had higher concentrations than terpenoids (Tables 1 and 2). Their atmospheric lifetimes (Calvert et al. 2011, Hellén et al. 2017) are much longer and therefore they can accumulate in the atmosphereir and be transported for longer distances. They wereare expected to have both biogenic and anthropogenic sources and they are also produced in the atmosphereir by the reactions of other VOCs (Ciccioli and Mannozzi, 2007). In this-the present\_study, the highest concentrations of VOAs in 2016 were already measured already in June (Table 2). This was at least partly due to the different measurement periods for the GC-MS2 and GC-MS3 in June and July. The dDays, when the VOAs were measured with the GC-MS2, were at the end of thein late June when the temperature (18 °C) and PAR were higher compared to the than VOC measurements with GCMS3 in June (temperature 12 °C). TAlso the MT sum measured together with the VOAs also showedhad the highest monthly mean in June.

15

20

25

The daily means of  $C_3$ -- $C_7$  VOAs had-were more highlyer correlatedion with the MT sum (R=0.6-0.85) than with anthropogenic compounds e-g-such as toluene (R=0.0-0.31), indicating the biogenic origin of these compounds, either direct or through secondary production in the airatmosphere. Only acetic acid showed some correlation with aromatic hydrocarbons (R=0.2-0.48). Since the lifetime of acetic acid is longest (1-2 weeks; Calvert et al. 2011) it is was expected to be more influenced by the long-range transported anthropogenic emissions.

The dDaily means of hexanoic acid had-were very highly correlatedion not only with 1-hexanol ( $R_{=}0.97$ ), but also with other  $C_6$  compounds often referred tocalled as green leaf volatiles (GLVs) i.eg. hexanal ( $R_{=}0.82$ ) and cis-3-hexenol ( $R_{=}0.83$ ). This indicates that also hexanoic acid could also be a GLV or that it is produced from GLVs in the airatmosphere. Correlation of the daily means of hexanoic acid with pentanoic and propanoic acids was also high ( $R_{=}0.89$  and 0.80, respectively) and

with the MT sum relatively high (R = 0.77).

For smaller acids (acetic, propanoic and butanoic) daytime maxima were observed, especially in July/August, but for pentanoic and hexanoic acids higher concentrations were observed during theat nights (Fig.ure 2). The mean diurnal variation of in

30 VOAs was not as strong as for the MTs and SQTs. Both direct biogenic emissions and production in the atmosphere are-were expected to be higher during the day, but since the mixing layer is-was also higher the VOAs wereare more diluted. However, due to the longer lifetimes of these acids, the mixing effect during the day is-was not as strong as for fast-reacting terpenes since the also dilution-background air may also have contained comparable amounts of these acids. -Tand theyse acids may even be transported from distant sources or produced in the upper parts of the mixing layer from the reactions of other VOCs.

TheAlso losses due to OH reactions during the day and dry/wet deposition during the-nights were alsoare expected to affect the concentrations (Calvert et al. 2011). In canopy scale flux measurements by proton transfer reaction-time of flight (PTR-TOF) devices downward fluxes of acetic acid have been detected, especially during theat nights (Schallhart et al. 2018).

**5 3.1.6 Concentrations of C5-C10 aldehydes**

 $C_5$ -- $C_{10}$  aldehydes can be directly emitted or they can be produced in the air atmosphere through oxidation of other compounds. Generally, the emissions are much lower than those of smaller aldehydes. Low emissions have been measured, e.g. from grasslands, but emissions of trans-2-hexenal, <del>and also</del> 2-hexenylacetate and 2-hexenol from <del>wounded damaged</del> and stressed plants can be significant (Fall 1999 and Hakola et al., 2001). Possanzini et al., (2000) found that larger aldehydes (heptanal, octanal) were emitted from citrus plants when exposed to ozoneQ2. There is also some evidence that e.g.-nonanal can be produced when ozoneQ2 attacks the fatty acids on leaf or needle surfaces (Bowman et al., 2003). Hakola et al. (2017) also measured also-C4--C10 aldehyde emissions from Norway spruce and found <del>out</del> that their magnitudey were about the same magnitude-s were similar to that of as-MT emissions during late summer.

The cConcentrations of  $C_5-C_{10}$  aldehydes were low; their monthly means remained <10 pptv (Table 1). In the measurements of Hellén et al. (2004) at the same site in March and April 2003, the concentrations were slightly higher (122-16 pptv) but within the same order of magnitudesimilar. The dDiurnal variations of in  $C_5-C_{10}$  aldehydes were followeding the variation of in isoprene, with low daytime values in June and high values in July and August (Figure 2).

20

10

15

The dDaily means of hexanal were highly correlated with MTs and SQTs (R\_=0.90) in summer (June-August). The dDaily means of nonanal and decanal have showed the highest correlation with  $\beta$ -farnesene in summer. Since  $\beta$ -farnesene emissions are related to stress, this could also have indicated stress-related sources for the emissions<del>m too</del>.

25 In July 24-, hour samples for analysis of carbonyls with a liquid chromatograph (LC, Praplan et al. 2018 in preparation) were collected concomitant with the GC\_MS3 measurements, similarly to Praplan et al. (2017). The concentrations of <C5 carbonyls were also obtained fFrom those samples also concentrations of <C5 carbonyls were obtained. The JJuly means for formaldehyde, acetaldehyde, acetone and butanal were 430, 270, 1820 and 50 pptv, respectively. The concentrations of these smaller carbonyls are were much higher than for the C5-cC10 aldehydes. This is-was expected, due to their longer lifetimes and larger emissions (Hellén et al. 2004).

**Table 1:** Mean concentrations of VOCs measured in summer (June-August) in 2011, 2015 and 2016 (GC-MS1 and GC-MS3) and monthly mean concentrations (pptv) in April--November 2016 with mean temperature (T), mean photosynthetically active radiation (PAR), mean mixing layer height (MLH), mean mixing layer heightMLH between 0:00- and 4:00 (MLH00.04 local

| N = number of meas                         | urements | and DL= |      | n iimit, - | = missi | ng value. |      |      |      |      |      |      |
|--------------------------------------------|----------|---------|------|------------|---------|-----------|------|------|------|------|------|------|
| Summer (Jun-Aug)                           |          |         |      |            |         |           |      | 2    | 016  |      |      |      |
| pptv                                       | DL       | 2011    | 2015 | 2016       | Apr     | May       | Jun  | Jul  | Aug  | Sep  | Oct  | Nov  |
| Ν                                          | -        | 267     | 244  | 240        | 163     | 244       | 61   | 114  | 125  | 246  | 84   | 187  |
| T (°C)                                     | -        | 15.3    | 14.1 | 14.7       | 2.9     | 13.3      | 12.0 | 17.8 | 14.3 | 11.1 | 2.0  | -1.6 |
| PAR (µmols -1 m -2 ) | -        | -       | 382  | 389        | 227     | 413       | 492  | 374  | 301  | 229  | 38   | 15   |
| MLH (m a.g.l.)                             | -        | -       | -    | -          | 315     | 485       | 615  | 301  | 294  | 192  | 136  | 120  |
| MLH 00-04 (m a.g.l.)            | -        | -       | -    | -          | 62      | 36        | 87   | 22   | 53   | 34   | 95   | 125  |
| MLH 12-16 (m a.g.l.)            | -        | -       | -    | -          | 783     | 1262      | 1605 | 819  | 883  | 567  | 222  | 138  |
| Isoprene                                   | 5.1      | 102     | 74   | 11         | 0.3     | 5.2       | 5.0  | 17.7 | 11.0 | 4.6  | 1.5  | 1.4  |
| MBO                                        | 3.5      | -       | 9.3  | 14         | 0.1     | 6.3       | 7.7  | 29.8 | 3.7  | 1.9  | 0.8  | 0.2  |
| α-Pinene                                   | 1.1      | 192     | 248  | 224        | 10      | 110       | 136  | 365  | 173  | 72   | 4.8  | 7.7  |
| Camphene                                   | 1.0      | 23      | 20   | 20         | 3.8     | 18        | 17   | 30   | 14   | 15   | 2.6  | 1.9  |
| β-Pinene                                   | 0.2      | 53      | 35   | 37         | 1.0     | 18        | 20   | 60   | 31   | 15   | 0.8  | 0.7  |
|                                            | 0.8      | 85      | 79   | 86         | 3.8     | 51        | 65   | 136  | 58   | 24   | 1.2  | 2.3  |
| p-Cymene                                   | 0.6      | 8       | 18   | 11         | 2.8     | 12.2      | 16.3 | 6.9  | 10   | 10   | 2.1  | 2.2  |
| 1,8-Cineol                                 | 0.9      | 10      | 19   | 9.7        | 0.4     | 6.2       | 8.9  | 14   | 5.8  | 4.5  | 1.5  | 1.3  |
| Limonene                                   | 1.0      | 23      | 14   | 30         | 0.6     | 6.2       | 9.2  | 62   | 21   | 9.7  | 0.4  | 0.7  |
| Terpinolene                                | 1.2      | 2       | 0.4  | 2.6        | 0.0     | 0.0       | 0.0  | 6.4  | 1.3  | 0.0  | 0.0  | 0.0  |
| Linalool                                   | 1.6      | -       | 5.6  | 0.8        | 0.0     | 1.1       | 1.1  | 1.0  | 0.3  | 0.1  | 0.4  | 0.2  |
| Myrcene                                    | 0.5      | -       | 4.2  | 5.3        | 0.3     | 1.7       | 2.0  | 10   | 3.8  | 1.4  | 0.2  | 0.2  |
| Bornylacetate                              | 0.6      | 0.6     | 0.6  | 1.1        | 0.0     | 0.7       | 0.7  | 1.8  | 0.7  | -    | 0.2  | 0.1  |
| MT SUM                                     | -        | 398     | 442  | 427        | 22      | 223       | 274  | 689  | 318  | 151  | 13   | 17   |
| Longicyclene                               | 0.3      | -       | -    | 0.2        | 0.0     | 0.3       | 0.1  | 0.4  | 0.1  | 0.1  | 0.0  | 0.0  |
| β-Farnesene                                | 0.9      | -       | -    | 1.2        | 0.0     | 0.0       | 0.5  | 2.8  | 0.3  | 0.0  | 0.0  | 0.0  |
| β-Caryophyllene                            | 0.8      | -       | -    | 7.8        | 0.0     | 2.6       | 3.2  | 16   | 4.5  | 4.0  | 0.0  | 0.1  |
| SQT1                                       | 0.4      | -       | -    | 0.5        | 0.0     | 0.0       | 0.0  | 1.4  | 0.2  | 0.0  | 0.0  | 0.0  |
| SQT2                                       | 0.4      | -       | -    | 1.1        | 0.0     | 0.0       | 0.0  | 2.7  | 0.7  | 0.0  | 0.0  | 0.0  |
| SQT3                                       | 0.6      | -       | -    | 0.5        | 0.0     | 0.0       | 0.0  | 1.4  | 0.2  | 0.0  | 0.0  | 0.0  |
| SQT4                                       | 0.7      | -       | -    | 1.4        | 0.0     | 0.0       | 0.0  | 3.8  | 0.4  | 0.0  | 0.0  | 0.0  |
| SQT SUM                                    | -        | -       | -    | 13         | 0.1     | 2.9       | 3.9  | 28   | 6.4  | 4.1  | 0.09 | 0.06 |
| Nopinone                                   | 0.8      | -       | -    | 3.8        | 0.7     | 2.1       | 1.6  | 7.7  | 2.0  | 1.2  | 0.5  | 0.5  |
| 4-AMCH                                     | 0.9      | -       | -    | 0.2        | 0.0     | 0.0       | 0.2  | 1.9  | 0.0  | 0.0  | 0.0  | 0.0  |
| MACR                                       | 0.3      | -       | -    | 3.8        | 4.1     | 3.8       | 3.4  | 4.8  | 3.3  | 5.0  | -    | -    |
| Pentanal                                   | 0.9      | -       | 41   | 6.8        | 5.2     | 11        | 6.1  | 8.1  | 6.3  | 3.8  | -    | -    |
| Hexanal                                    | 0.4      | -       | 30   | 9.1        | 3.5     | 9.1       | 6.5  | 12   | 8.6  | 3.2  | -    | -    |
| Octanal                                    | 1.8      | -       | 3.2  | 4.8        | 1.5     | 4.9       | 4.2  | 6.2  | 4.0  | 0.3  | -    | -    |
| Nonanal                                    | 0.8      | -       | 27   | 7          | 1.8     | 3.0       | 5.4  | 8.4  | 7.0  | 0.0  | -    | -    |
| Decanal                                    | 1.8      | -       | 19   | 7.1        | 0.4     | 3.0       | 6.8  | 10   | 4.1  | 0.5  | -    | -    |
| trans-2-Hexenal                            | 1.6      | -       | 4.6  | 1.1        | 0.0     | 1.4       | 0.3  | 2.3  | 0.7  | 1.5  | -    | -    |

wintertime UTC+2) and mean mixing layer height MLH between 12:00 and -16:00 (MLH12-16) during the VOC measurements.

N\_=\_number of measurements and DL\_=\_detection limit, '-'\_=\_missing valu

1

a.g.l. = above ground level, MBO = 2-methyl-3-butenol, SQT = sesquiterpene, MT = monoterpene, 4-AMCH = 4-acetyl-1methylcyclohexene, MACR = methacrolein

 Table 2: Mean concentrations of studied-VOCs measured in summer (June-August) 2015 and 2016 (GC-MS2) and monthly

 5
 mean concentrations (pptv) in February-September 2016 with mean temperature (T) and photosynthetic radiation (PAR). N\_= number of measurements-and, DL\_=detection limit, '-' = missing value.

|                                            |     | Summer(Jun-Aug) |      |      | 2016 |      |      |      |      |      |      |
|--------------------------------------------|-----|-----------------|------|------|------|------|------|------|------|------|------|
| pptv                                       | DL  | 2015            | 2016 | Feb  | Mar  | Apr  | May  | Jun  | Jul  | Aug  | Sep  |
| Ν                                          | -   | 615             | 218  | 43   | 56   | 92   | 240  | 31   | 81   | 106  | 218  |
| T (°C)                                     | -   | 14.0            | 16.1 | -5.8 | -3.9 | 5.2  | 12.4 | 18.1 | 17.7 | 14.4 | 10.9 |
| PAR (µmols -1 m -2 ) | -   | 388             | 371  | 85   | 59   | 253  | 448  | 483  | 347  | 368  | 252  |
| MT SUM                                     | -   | 324             | 350  | 5.4  | 4.5  | 55   | 129  | 568  | 502  | 171  | 204  |
| Acetic acid                                | 280 | 1799            | 978  | 1530 | 714  | 1172 | 899  | 1723 | 1395 | 564  | 1418 |
| Propanoic acid                             | 22  | 127             | 76   | 46   | 26   | 55   | 51   | 225  | 90   | 29   | 87   |
| Butanoic acid                              | 14  | 45              | 68   | 37   | 14   | 41   | 37   | 114  | 76   | 56   | 74   |
| Pentanoic acid                             | 5   | 16              | 44   | 16   | 4.0  | 10   | 8    | 88   | 62   | 25   | 32   |
| Hexanoic acid                              | 7   | 15              | 11   | 3.7  | 1.3  | 1.9  | 0.5  | 35   | 15   | 2.3  | 3.3  |
| Heptanoic acid                             | 16  | 3.5             | 5    | 1.0  | 0.4  | 0.3  | 0.0  | 26   | 4.4  | 0.0  | 0.2  |
| Isopropanol                                | 11  | 122             | 228  | 9.4  | 1107 | 50   | 23   | 61   | 124  | 361  | 1097 |
| 1-Butanol                                  | 6   | 138             | 365  | 169  | 48   | 306  | 339  | 508  | 304  | 311  | 277  |
| 1-Pentanol                                 | 19  | 2.6             | 2.9  | 0.0  | 0.0  | 0.7  | 0.6  | 9.3  | 2.9  | 0.0  | 4.9  |
| 1-Hexanol                                  | 3   | 1.5             | 2.2  | 0.0  | 0.0  | 0.2  | 0.2  | 5.2  | 3.1  | 0.1  | 1.0  |
| 1-Penten-3-ol                              | 2   | 0.5             | 0.8  | 0.0  | 0.0  | 0.0  | 0.1  | 0.5  | 2.0  | 0.0  | 0.1  |
| trans-3-Hexen-1-ol                         | 5   | 0.5             | 1.3  | 0.0  | 0.0  | 0.3  | 0.1  | 0.0  | 3.3  | 0.0  | 0.0  |
| cis-3-Hexen-1-ol                           | 4   | 0.0             | 0.4  | 0.0  | 0.0  | 0.0  | 0.0  | 0.9  | 0.4  | 0.0  | 0.1  |
| trans-2-Hexen-1-ol                         | 13  | 1.3             | 0.0  | 0.0  | 0.0  | 0.0  | 0.0  | 0.0  | 0.0  | 0.0  | 0.0  |
| cis-2-Hexen-1-ol                           | 12  | 0.2             | 0.0  | 0.0  | 0.0  | 0.0  | 0.0  | 0.0  | 0.0  | 0.0  | 0.0  |
| 1-Octen-3-ol                               | 3   | 0.1             | 0.1  | 0.0  | 0.0  | 0.0  | 0.0  | 0.2  | 0.2  | 0.0  | 0.2  |
| Butylacetate                               | 39  | 7.9             | 0.6  | 0.0  | 0.0  | 1.5  | 0.0  | 2.2  | 0.2  | 0.1  | 0.6  |
| Hexylacetate                               | 8   | 0.0             | 0.0  | 0.0  | 0.0  | 0.0  | 0.0  | 0.0  | 0.0  | 0.0  | 0.0  |
| cis-3-Hexenylacetate                       | 6   | 0.8             | 0.2  | 0.0  | 0.0  | 0.0  | 0.0  | 0.5  | 0.3  | 0.0  | 0.0  |
| trans-2-Hexenylacetate                     | 7   | 0.0             | 0.0  | 0.0  | 0.0  | 0.0  | 0.0  | 0.0  | 0.0  | 0.0  | 0.0  |

 There has been discussion on the The formation of aldehydes in GC and PTR-MS instruments from organic peroxides has been

 10
 discussed (Rivera-Rios et al. 2014). However, we measured methacrolein MACR and hexanal with both\_LC-UV and the GC\_

 MS3 in July 2016 and the results were at comparable levelscomparable (methacrolein-MACR 4.7 and 4.8 pptv and hexanal

8.4 and 12 pptv, respectively). For pentanal even higher concentrations were obtained by LC-UV (July mean 45 ppt). In 2015 when the GC\_MS1 was used, the aldehyde concentrations were clearly higher than in 2016 (Table 1) and it is possible that the production from organic peroxides in the GC\_MS1 mayeould explain the difference. This indicates that the hypothesis by Rivera-Rios et al. (2014) might be truemay hold for some GC instruments, but it is still unclear under which circumstances.

**3.1.7 Concentrations of alcohols and acetates**

C4-C8 alcohols and acetates (including GLVs) have generally show very low concentrations; the monthly means were mostly below the detection limits (Table 2). The only exceptions were 1-butanol and isopropanol, both of which are bothwere used in
instrumentation at the site and have therefore showed higher concentrations from leaks and exhaust lines. As for the other BVOCs the highest concentrations of most alcohols and acetates were measured in summer. Most of the alcohols and acetates measured alcohols and acetates—were GLVs, which are emitted due to herbivory or pathogen infection by almost every green plant (Scala et al. 2013) or due to physical damage of plants (Hakola et al. 2001).

**15 **3.2** Correlation of concentrations with temperature**

The mMonthly and daily means of most of the studied BVOCs examined were found to becorrelateds exponentially correlated with temperature. This temperature dependence is described in more further detail for different compound groups as well as for individual BVOCs in the following sections.

**20**

5

**3.2.1 Correlation of MT-monoterpene concentrations with temperature**

The mMonthly mean MT concentrations showedhad very strong exponential correlation with temperature (R2=0.92, Fig.ure 5a). The site is dominated by Scots pines, which have temperature and light-dependent emissions of MTs (Tarvainen et al. 2005). Correlation of photosynthetic active radiation (The PAR was highly correlated) with monthly mean MT concentrations was also high (R2Apr.Nov 2016=0.73), but correlation was clearly lower than with temperature.

The dDaily means of MTs also correlated well with temperature (R2Apr-Nov 2016=\_0.83 and R2Jun-Aug 2016=\_0.88, Table 3 and Figure-Fig. 5b). The hHigh correlation with temperature observed indicates exponential correlation of mean concentrations
with temperature indicates that temperature has a major effect on the seasonality of the concentrations and emissions and processes controlling them. In an earlier \_previous\_study by Lappalainen et al. (2009) lower correlation (R2=0.50) with temperature was found for the PTR-MS data. However, they used only daytime medians. In our study 24-hour averages starting at 8:00 (UTC+2) and ending next day at 8:00 (UTC+2) have beenwere used.

Temperature dependence of monoterpene MT emissions are is often described by the Guenther algorithm (Guenther et al., 1993):

(<del>9</del>10)

**$E=E_S \times exp(\beta (T-T_S))$**

5

10

, where  $E_s$  is the standardized emission potential ( $\mu$ g gdw-1\_dry weight (dw) h-1), T is the leaf temperature (°C), Ts is the standard temperature of 30 °C and  $\beta$  is the temperature sensitivity (°C-1) of the emissions. Often the value 0.09 °C-1 is used for  $\beta$  to describe monoterpene MT emissions. In our monthly and daily mean concentration data, the temperature sensitivity was clearly higher ( $\beta$ =0.20 °C-1, Fig\_tre 5 and Table 3). The temperature also affects also the vertical mixing of air, and a lower mixing after warm sunny days is one-probablye reason for increased the temperature sensitivity of the concentrations. Even though the value 0.09 °C-1 is often used for  $\beta$  to model emissions, it is known to vary (Hakola et al. 2006). HAlso here the temperature sensitivity of the daily mean MT concentration for the summer months ( $\beta$ =0.27 °C-1, June--August) was also

higher than for the entirewhole growing season (β=0.20 °C-1, Apr-Nov).

- 15 To study-determine the temperature sensitivity of the individual MTs, data from the GC-MS3 was-were used. Exponential correlations of the monthly means with temperature were found to haveshowed that R2>0.91 (Table 3 and Figure S2S3) for all monoterpenoids except 1,8-cineol (R2=0.77), p-cymene (R2=0.72), bornylacetate (R2=0.71) and linalool (R2=0.25). Tarvainen et al. (2005) found that in Scots pine emissions, 1,8-cineol was the only MT, which that was both light and temperature\_dependent while the others were only temperature\_dependent. *p*-Cymene has been detected e.g. in Norway spruce emissions (Hakola et al. 2017), but it also has also anthropogenic sources (Hakola et al. 2012). Linalool is known to be emitted
- by trees as a result of biotic stress (Petterson, 2007, Blande et al. 2009). Bornylacetate, linalool and 1,8-cineol have-showed very low concentrations, which also resulteds in higher uncertainty. For the MTs with high ( $R^2>0.91$ ) temperature correlation, the  $\beta$ -values of the monthly means varied between 0.15 and 0.26 °C-1 being lowest for camphene and highest for  $\beta$ -pinene.

**25 3.2.2 Correlation of SQT sesquiterpene concentrations with temperature**

As for the MTs, the also monthly and daily means of the SQTs also showedhad very strong exponential correlation with temperature (Table 3, Figure S3S4). The temperature sensitivity of the SQTs was even higher than for MTs. The SQT emissions from Norway spruce (Hakola et al., 2017) and Scots pine (Tarvainen et al., 2005) wereare closely correlated with

30 temperature, but the SQT emissions may also have been influenced by light (Duhl et al. 2008). The Especially daily mean  $\beta$ caryophyllene concentrations showedhad very high exponential correlation with temperature ( $R^2_{Jun-Aug}=0.96$ ) supporting only temperature\_dependent emissions. The mMonthly means of the SQT sum (consisting mainly of  $\beta$ -caryophyllene) had also showed very high exponential correlation with temperature ( $R^2=0.97$ ), indicating that also-seasonality is also driven by the temperature. For the other SQTs, the correlations were lower than for  $\beta$ -caryophyllene. Low concentrations with higher Formatted: German (Switzerland) Formatted: German (Switzerland) measurement uncertainty and e.g. light\_ and stress\_related emissions may have significantly aeffecteds\_on-the correlations.  $\beta$ -Farnesene is is known to be emitted due to the biotic stress (Kännaste et al., 2009) and it has been shown to increases simultaneously with linalool in the emissions of Norway spruce and Scots pines (Hakola et al. 2006 and 2017). However, the linalool and  $\beta$ -farnesene concentrations did not correlate in our data. Bouvier-Brown et al. (2009) suggested that at least in a ponderosa pine forest  $\beta$ -farnesene emissions are-may be both temperature\_ and light\_dependent.

5

10

**Table 3:** Correlation of VOC concentrations with temperature at SMEAR II in 2016, intercept ( $\underline{\alpha}$ a) of temperature dependence curve, temperature sensitivity ( $\beta$ ) and temperature correlations ( $\mathbb{R}^2$ ) of monthly (April-November) and daily (June-August) mean concentrations and mixing layer height (MLH)--scaled concentration of individual measurements points ( $\mathbb{C}_{MLH}$ ). The fFitted curves were exponent functions  $y_=\underline{\alpha}e^{\beta x}$ , where  $y_=$  concentration or MLH scaled concentration,  $x_=$  temperature and  $\beta$  = temperature sensitivity.

| · · · · · · · · · · · · · · · · · · · | Month     | ly mean (A | or-Nov)        | Daily     | mean (lun | -Λιισ)         | Com (Apr-Nov)                  |      |      |  |
|---------------------------------------|-----------|------------|----------------|-----------|-----------|----------------|--------------------------------|------|------|--|
|                                       | and and a | в (С-1)    | R 2 | any and a | В (C-1)   | R 2 | $\beta = \beta (C^{-1}) = R^2$ |      |      |  |
| Isoprene                              | 0.76      | 0.16       | 0.74           | 0.23      | 0.24      | 0.84           | 00 0                    | 0 10 | 0.52 |  |
| MBO                                   | 0.70      | 0.10       | 0.74           | 0.23      | 0.24      | 0.84           | 30                             | 0.13 | 0.52 |  |
| a Dinono                              | 6.07      | 0.20       | 0.00           | 1.02      | 0.31      | 0.70           | 750                            | 0.22 | 0.04 |  |
| a-Pillelle
Camphono                | 0.07      | 0.25       | 0.95           | 1.95      | 0.50      | 0.05           | 110                            | 0.17 | 0.05 |  |
|                                       | 2.34      | 0.15       | 0.97           | 0.04      | 0.21      | 0.00           | 110                            | 0.10 | 0.05 |  |
| p-Pinene                              | 0.68      | 0.26       | 0.97           | 0.45      | 0.27      | 0.82           | 80                             | 0.19 | 0.65 |  |
| 3 Δ 2 -Carene   | 1.86      | 0.25       | 0.95           | 0.59      | 0.30      | 0.88           | 190                            | 0.19 | 0.66 |  |
| p -Cymene                      | 2.44      | 0.10       | 0.72           | 16.53     | -0.04     | 0.07           | 60                             | 0.20 | 0.58 |  |
| 1,8-Cineol                            | 0.88      | 0.15       | 0.77           | 0.52      | 0.18      | 0.71           | 120                            | 0.16 | 0.64 |  |
| Limonene                              | 0.45      | 0.25       | 0.91           | 0.14      | 0.34      | 0.91           | 40                             | 0.21 | 0.61 |  |
| Linalool                              | 0.09      | 0.12       | 0.25           | 0.01      | 0.23      | 0.69           | 30                             | 0.13 | 0.48 |  |
| Myrcene                               | 0.19      | 0.20       | 0.93           | 0.03      | 0.33      | 0.77           | 30                             | 0.18 | 0.64 |  |
| Bornylacetate                         | 0.04      | 0.20       | 0.43           | 0.005     | 0.34      | 0.71           | 30                             | 0.12 | 0.56 |  |
| MT sum                                | 14.38     | 0.22       | 0.96           | 5.57      | 0.27      | 0.88           | 1500                           | 0.18 | 0.67 |  |
| MACR                                  | 4.02      | 0.00       | 0.00           | 0.21      | 0.17      | 0.86           | 170                            | 0.12 | 0.24 |  |
| Nopinone                              | 0.44      | 0.12       | 0.86           | 0.07      | 0.25      | 0.80           | 70                             | 0.14 | 0.55 |  |
| Longicyclene                          | -         | -          | -              | 0.003     | 0.28      | 0.69           | 10                             | 0.13 | 0.4  |  |
| β-Farnesene                           | -         | -          | -              | 0.003     | 0.37      | 0.83           | 140                            | 0.09 | 0.13 |  |
| β-Caryophyllene                       | 0.04      | 0.34       | 0.87           | 0.019     | 0.37      | 0.96           | 50                             | 0.18 | 0.51 |  |
| Other SQTs                            | -         | -          | -              | 0.006     | 0.41      | 0.70           | 30                             | 0.17 | 0.31 |  |
| SQT sum                               | 0.07      | 0.32       | 0.96           | 0.006     | 0.49      | 0.95           | 40                             | 0.21 | 0.50 |  |
| Pentanal                              | 4.2       | 0.04       | 0.23           | 1.73      | 0.09      | 0.70           | 280                            | 0.13 | 0.28 |  |
| Hexanal                               | 2.24      | 0.09       | 0.66           | 1.60      | 0.11      | 0.90           | 190                            | 0.16 | 0.38 |  |
| Octanal                               | 0.65      | 0.11       | 0.22           | 1.12      | 0.09      | 0.36           | 100                            | 0.15 | 0.26 |  |
| Nonanal                               | 0.31      | 0.14       | 0.07           | 1.88      | 0.08      | 0.70           | 100                            | 0.15 | 0.26 |  |
| Decanal                               | 0.19      | 0.21       | 0.67           | 0.84      | 0.13      | 0.43           | 50                             | 0.20 | 0.36 |  |
| trans-2-Hexenenal                     | 0.006     | 0.37       | 0.82           | 0.07      | 0.19      | 0.57           | 90                             | 0.31 | 0.14 |  |
| Acetic acid                           | 947       | 0.01       | 0.02           | 107.3     | 0.13      | 0.22           | 57224                          | 0.12 | 0.27 |  |

| Propanoic acid                                                                       |  | 31.1 | 0.06 | 0.20 | 2.08  | 0.20 | 0.39 | 2728 | 0.14 | 0.36 |  |
|--------------------------------------------------------------------------------------|--|------|------|------|-------|------|------|------|------|------|--|
| Butanoic acid                                                                        |  | 27.7 | 0.06 | 0.49 | 23.45 | 0.06 | 0.31 | 2207 | 0.13 | 0.33 |  |
| Pentanoic acid                                                                       |  | 3.48 | 0.15 | 0.58 | 0.84  | 0.23 | 0.65 | 573  | 0.16 | 0.41 |  |
| Hexanoic acid                                                                        |  | 0.27 | 0.20 | 0.42 | 0.05  | 0.30 | 0.30 | 232  | 0.16 | 0.43 |  |
| MBO = 2-methyl-3-butenol, MACR = methacrolein, MT = monoterpene, SQT = sesquiterpene |  |      |      |      |       |      |      |      |      |      |  |

**3.2.3 Correlation of isoprene and 2-methyl-3-butenolMBO concentrations with temperature**

Isoprene emissions are both light and temperature dependent (Guenther et al. 1993, Ghirardo et al. 2010). Here correlation of the isoprene daily mean concentrations with light and the temperature activity factor (Guenther et al. 1993) was slightly lower (R2=0.74) than for the temperature only (R2=0.84, Fig. ure S4S5). However, the difference in R2 is not high small and since the concentrations were low and close to the detection limits, no clear conclusions can be made based ondrawn from this. The 2-

10 Methyl-3-butenol (MBO) was somewhat better correlated with light and the temperature activity factor ( $R^2 = 0.76$ ) than with temperature only (R2=0.70). This is in contrast with to the Scots pine emissions, in which the where MBO has been found to be- was only temperature--dependent (Tarvainen et al. 2005).

Even though the diurnal variation inof most MT, SQT and MBO concentrations are did not following the ambient temperature, 15 isoprene has showed the highest concentrations during the day, while the and 30 -minute mean concentrations have were exponentially correlatedion with the ambient temperature (Figure S65,  $R^2$ =0.64). Due to the close link between isoprene production and light, isoprene is produced and emitted from trees only during the light hours and is therefore detected in the air atmosphere only during the day while the MBO, MTs and SQTs are also emitted from storage pools inside the needles or leaves also-during the night and due to lower vertical mixing the ambient air concentrations are higher at nightthen (Ghirardo 20 et al. 2010).

**3.2.5 Correlation of terpenoid reaction product concentrations with temperature**

The nNopinone concentrations showed very clear exponential correlation with temperature ( $R^2_{daily} = 0.80$ ) due to the 25 temperature dependence of its precursor (β-pinene) and more rapidfaster production on warm and sunny summer days. The temperature sensitivity of the nopinone daily means ( $\beta$ =0.25 °C-1) is similarelose to the sensitivity of its precursor  $\beta$ -pinene ( $\beta$  $= 0.27 \ ^{\circ}C^{-1}$ ).

MACR, which is a reaction product of isoprene, washes as highly correlated  $(R^2 = 0.86)$  with temperature in summer as its precursor isoprene ( $R^2 = 0.84$ ) but the temperature sensitivity was slightlyis a bit lower ( $\beta_{isoprene} = 0.24 \text{ °C}^{-1}$  and  $\beta_{MACR} = 0.17$ °C-1). Similar to its precursor, the 30 minute mean concentrations of MACR have also showed low exponential correlation Formatted: Superscript

30

5
with temperature ( $R^2=0.32$ , Figure  $\frac{55}{50}$ ), but the monthly means of MACR didare not correlateing with temperature ( $R^2<0.01$ , Table 3). MACR has also direct anthropogenic sources (Biesenthal and Shepson, 1997) and in spring and autumn when biogenic emissions are lower, the influence of these sources is expected to be more important also due to longer lifetimes of VOCs in the atmosphere.

**3.2.6 Correlation of oxygenated volatile organic compoundOVOC concentrations with temperature**

Since the concentrations of most  $C_5$ - $C_{10}$  aldehydes are-were very close to the detection limits, the results are more scattered; but still clearly showing strong correlation with the temperature. The hHighest correlation of the daily means in summer (Jung-10 -August) was found for hexanal ( $R^2 = 0.90$ ) and lowest for octanal ( $R^2 = 0.36$ ) and decanal ( $R^2 = 0.43$ , Table 3 and Fig.ure S5S7). The temperature sensitivities of the aldehydes ( $\beta = 0.08$ --0.13 °C-1) were clearly lower than for terpenoids ( $\beta = 0.18$ --0.67 °C-1). Aldehydes have direct biogenic emissions (Seco et al. 2007, Hakola et al. 2017), but they are also produced in the atmosphere by the oxidation of other VOCs. The correlation of the daily mean concentrations of trans-2-hexenal with light and the temperature activity factor (Guenther et al. 1993) was higher ( $R^2 = 0.71$ ) than just only with temperature ( $R^2 = 0.57$ ), indicating a light-dependent source.

15

5

25

As for the isoprene and its reaction product (MACR), the diurnal variation of in pentanal and hexanal concentrations have were also correlation correlated with temperature and temperature sensitivities for the 30 minute mean concentrations  $(\beta_{\text{pentanal}}\beta_{\text{pentanal}}=0.07 \text{ °C}^{-1} \text{ and } \beta_{\text{hexanal}}=0.08 \text{ °C}^{-1}$ , Figure S55S6b), are close to the similar to that of MACR ( $\beta_{\text{MCAR}}=0.06 \text{ °C}^{-1}$

20 1). This indicates that photochemical reactions could be an important source for these compounds as well.

A weak correlation with temperature was also found also for the VOAs, but it was lower than for most other VOCs studied VOCs (Table 3). Due to the long lifetime of VOAs, the background concentrations and anthropogenic sources are-were expected to have higher more of an effect on the concentrations, and therefore their effect of local temperature dependent emissions and production in the air atmosphere remains unclear. Correlation of the daily means was highest for pentanoic acid (R2=0.65, Table 3, Fig. ure S6S7). The temperature sensitivity of the butanoic acid daily means ( $\beta = 0.06 \,^{\circ}C^{-1}$ ) was lower than for the other VOAs. The bButanol concentrations at the site are-were strongly affected by the emissions from the particle counters used at the site and it is were expected to produce butanoic acid.

**3.2.7 Seasonality of temperature correlations 30**

The vVariation inof the daily mean concentrations is best explained by the temperature in summer (Table 4). Also-The temperature sensitivity of the MT, SQT and isoprene concentrations are-were highest during the summer months and lower in autumn and spring. In summer, the emissions from trees wereare expected to play a major role, but in spring and autumnthe relative impact of other emissions (e.g. sawmill emissions) increasesincreased.

In May, values lower than expected by the overall temperature dependence were detected (Figure Fig. 5b). This is most
probably explained by the beginning of the growing season (mean ETS<200) with lower emission potentials (Hakola et al. 2001, and 2012). In autumn (September-November), when values were more scattered (Figure Fig. 5b), the emissions from fresh leaf litter are-were expected to contribute have-significantly contribution to the concentrations (Hellén et al., 2006, Aaltonen et al., 2011). Mäki et al., 2017). Then MT sum was correlating (Figure S7) a bit better with soil humus layer temperature (R2=0.87) than with ambient temperature (R2=0.80), which also indicates the soil related sources. During the colder months, when biogenic emissions are low, also anthropogenic closeby sawmillemissions from a closebynear the sawmill were expected to showahave higher relative influence. This is detectedwas indicated by higher MT concentrations in November than expected by the general temperature correlation (Figure Fig. 5b). However, if daily means of all studied months are plotted together, correlation with temperature is relatively high (R2MT=0.83, R2SQ1=0.67 R2ISOPERME=0.68, Table 4). these higher concentrations were still within the measurement uncertainty.

15

**Table 4:** Characterization of the temperature dependence of the isoprenoid concentrations with intercept ( $\alpha$ +), temperature sensitivity ( $\beta$ ) and correlation (R2) of the daily mean concentrations of MT sum and SQT sum measured at SMEAR II in different months in 2016. N=number of daily means. The fFitted curves were exponent functions y\_=  $\alpha$ +e $\beta$ x, where y\_= concentration, x\_=temperature and  $\beta$ =temperature sensitivity.

|         | MT sum |            |                       |                | SQT sum |            |                       |                | Isoprene |            |          |                |
|---------|--------|------------|-----------------------|----------------|---------|------------|-----------------------|----------------|----------|------------|----------|----------------|
|         | Ν      | α ə | β (°C -1 ) | R 2 | Ν       | α ə | β (°C -1 ) | R 2 | N        | α ə | β (°C⁻¹) | R 2 |
| Apr     | 9      | 8.16       | 0.18                  | 0.43           | 5       | 0.17       | -0.05                 | 0.11           | 2        | -          | -        | -              |
| May     | 13     | 1.19       | 0.33                  | 0.67           | 13      | 0.10       | 0.24                  | 0.82           | 13       | 0.31       | 0.19     | 0.70           |
| Jul     | 8      | 1.19       | 0.26                  | 0.92           | 8       | 0.09       | 0.33                  | 0.91           | 8        | 0.09       | 0.29     | 0.87           |
| Aug     | 7      | 6.53       | 0.37                  | 0.82           | 7       | 0.001      | 0.63                  | 0.92           | 7        | 0.14       | 0.31     | 0.94           |
| Sep     | 15     | 29.2       | 0.12                  | 0.18           | 15      | 1.48       | 0.05                  | 0.02           | 15       | 0.89       | 0.14     | 0.60           |
| Oct     | 5      | 13.4       | -0.08                 | 0.16           | -       | -          | -                     | -              | 5        | 1.3        | -0.08    | 0.05           |
| Nov     | 9      | 11.7       | 0.11                  | 0.69           | -       | -          | -                     | -              | -        | -          | -        | -              |
| Apr-Nov | 66     | 10.64      | 0.20                  | 0.83           | 48      | 0.06       | 0.31                  | 0.67           | 50       | 0.50       | 0.18     | 0.68           |

20

**3.2.8 Temperature sensitivities vs. vapour pressures**

The temperature sensitivities (β-values) of the most abundant terpenoids were found to be dependent on their vapour pressures 25 (FigureFig. 6). Vapour pressures have been estimated with the AopWinTM module of the EPITM software suite (https://www.epa.gov/tsca-screening-tools/epi-suitetm-estimation-program-interface, EPA, U.S.A) were used. The -yapour

pressures used in the calculations are given in Supplementary Table S1. Higher  $\beta$ -values were found for the terpenes with lower vapour pressure, higher boiling point and higher carbon number. This indicates that temperature sensitivity is driven by the volatility of the compounds. In addition to the temperature sensitivities of the monthly means shown in figureFig. 6, also the summertime daily means of the terpenes had also showed the same dependence on vapour pressures. However, camphene,

5 p-cymene, 1,8-cineol and linalool did not show this dependence neither for the monthly or daily means. For these compounds, the temperature sensitivity was lower than expected, based on the volatility. These differences, as previously mentioned, may have been due to the concentrations of camphene and p-cymene affected by the emissions of the nearby sawmill, while 1,8-cineol also showed light-dependent emissions and linalool is emitted from plants, due to stressFor these compounds temperature sensitivity was lower than expected based on the volatility. The possible reasons for these differences are as
10 mentioned also in the previous sections: the concentrations of camphene and p cymene are affected by the emissions of the eloseby sawmill, 1,8-cineol has also light dependent emissions and linalool is emitted from plants due to stress.

Even though the VOAs were lesshad lower correlatedion with temperature than terpenes (Table 3), the dependence of temperature sensitivity on vapour pressures was also found for all other VOAs, except butanoic acid, which is-was expected
to be produced from the 1-butanol used in other instruments at the site. For C5--C10 aldehydes, only monthly means had showed this dependence. The sSummertime daily means of aldehydes had-were more highlyer correlatedion with temperature, but still β-values still did not follow the vapour pressures.

These dependencies can be used to estimate the kind type of compound that could explain the missing reactivity found by total reactivity measurements or to assist in the identification of compounds in direct mass spectrometric methods.

**3.2.9. Simple proxies for estimating local biogenic volatile organic compound BVOC concentrations**

Kontkanen et al. (2016) have developed MT proxies, which that are used for calculating concentrations of the MT sum at the SMEAR II. The proxies are based on the temperature-controlled emissions from the forest ecosystem, the dilution caused by the mixing within the boundary layer and different-various oxidation processes. Our data shows that the monthly and daily means of both the sum and individual MTs and most other BVOC concentrations can be described relatively well, using only the temperature (Tables 3 and 4, FigureFig. 5) and a simplified proxy for the daily or monthly mean concentrations would be

 $30 \quad [BVOC(monthly or daily)_i]_{proxy} = \alpha e^{\beta T}$ (1011)

where  $\underline{\alpha}$  and  $\beta$  are empirical coefficients found from thein T table 3 and obtained from the correlation of monthly and daily mean concentrations with temperature (Figure Fig. 75) and T is the ambient temperature.

However, for describing the diurnal variation in mixing of air has tomust be taken into account. To roughly describe the dilution caused by the vertical mixing we multiplied the concentrations with mixing layer heights the MLHs ( $C_{MLH}$ =[VOC] x MLH) and studied the correlation of these  $C_{MLH}$  values with temperature (Table 3 and FigureFig. 7). All individual measured data points available from the year 2016 were used except the cases when in which MLH was below the detection limit of the lidar

5 LIDAR (< 60 m). Therefore, the highest values during the most stable nights are missing. The correlation of the CMLH values with temperature was best for the MTs (R2MTsum=0.67). The modelling study of Zhou et al. (2017) showed that the variation of monoterpenein MT concentrations is was mainly driven by the emissions and mixing, while for faster\_reacting SQTs also oxidation also plays a role. For oxygenated compounds also production in the air atmosphere and deposition also have an effectaffect on-local concentrations, and therefore correlation of the CMLH values with temperature are was lower than for the 10 MTs (Table 3). In our case the proxy for the concentration of the MT or SQT sum or an individual compound (BVOCi,), when MLH> 60m, would be

$$[BVOC_i]_{proxy} = \frac{ae^{\beta T}}{MLH} \tag{4442}$$

15 where  $\underline{\alpha}$  and  $\beta$  are empirical coefficients found from the in Ttable 3 and obtained from the correlation of concentrations multiplied by the mixing layer heightMLH (CMLH) with temperature (FigureFig. 7), T is the ambient temperature and MLH is the mixing layer height.

**3.3 Importance of studied the biogenic volatile organic compounds BVOCs for local atmospheric chemistry**

20

25

**3.3.1 Reactivity of measured the biogenic volatile organic compounds BVOCs**

To describe the effects of different dvarious compounds and compound groups on the oxidation capacity of air the atmosphere we calculated the OH, NO3 and O3 reactivities for the BVOCs studied BVOCs using the measured concentrations and reaction rates with different oxidants (Eq. 4).

The OH reactivity of the MTs wasis clearly higher than for the any other VOC group at this boreal forest site, showing the importance of the MTs for the local OH chemistry (Figure Fig. 8a). The OH reactivity of monoterpenes the MTs wasis 10 times higher than the reactivity of for SQTs, even in July when the SQT concentrations were highest. The OH reactivity of the other compound groups was-were minor (ea.app. 4% in July). Based on additional measurements in July, also the contribution of

30 C\_scarbonyls <C\_s(formaldehyde, acetaldehyde, acetone and butanal) was minor. However, even when reactivities of all the BVOCs, anthropogenic VOCs and other OH-reactive compounds measured at the site were added up, the OH reactivity was much lower (<50 %) than the total reactivity measured at the site by Sinha et al. (2010), Nölscher et al. (2012) and Praplan et al. (2018, in preparation). Based on additional measurements in July, also the contribution of <C\_s carbonyls (formaldehyde, acetaldehyde).

acetaldehyde, acetone and butanal) was minor. In a previous modelling study at the site, the OH reactivity of the MTs was also highest, but the other VOCs showed almost as high a contribution (Mogensen et al., 2011). These other VOCs included 415 compounds mainly consisting of second- or higher-order organic reaction products, but not SQTs. Even with these reaction products, app. 50--70% of the measured total OH reactivity was still missingremained unexplained, However, even adding up the reactivity of all the BVOCs, anthropogenic VOCs and other OH reactive compounds measured at the site, OH reactivity is

5 the reactivity of all the BVOCs, anthropogenie VOCs and other OH reactive compounds measured at the site, OH reactivity is much lower (<50 %) than the total reactivity measured at the site by Sinha et al. (2010), Nölscher et al. (2012) and Praplan et al. (2018 in preparation).

Since O3 is reacteding only with unsaturated VOCs, only isoprene, most MTs, SQTs and unsaturated alcohols of the measured
VOCs only isoprene, most MTs, SQTs and unsaturated alcohols contributed to the O3 reactivity. From May to September, the
SQTs had major contributed greatlyion to the O3 reactivity (FigureFig. 8b). Even though the MT concentrations were appare
ea. 50 times higher than the SQT concentrations, the O3 reactivity given by theof SQTs is was about 3-three times higher than
the reactivitythat of the MTs. Hakola et al. (2017) also showed the high crucial importance of the SQTs for the O3 reactivity in the spruce emissions. This indicates that the SQTs-and, escpecially β-caryophyllene (FigureFig. S8dS8d) have much higher
effects for example on local ozoneO3 deposition destruction than MTs. Several studies have shown that the O3 deposition

fluxes measured ozone deposition fluxes-cannot be explained by stomatal and known non-stomatal sinks modelled-stomatal and known non-stomatal sinks, such as reactions with the VOCs measured <del>VOCs</del> in the gas phase (Clifton et al. 2017; Wolfe et al. 2011a; Rannik et al. 2012. Clifton et al. 2017). Higher than expected impact of the SQTs could explain at least part of the discrepancy.

25

Also-The NO3 radicals alsoare mainly reacteding with the unsaturated VOCs, and the MTs\_have-clearly highest-contributed mostion to the NO3 reactivity of BVOCs at the site (FigureFig. 8c). Of the SQTs only  $\beta$ -caryophyllene was considered, since the reaction rate coefficients were not available for the others. However,  $\beta$ -caryophyllene had\_showed\_the highest concentrations of all the SQTs, but still did not have-significantly aeffect on the NO3 reactivity. Liebmann et al. (2018) measured the total NO3 reactivity at the site in September 2016 and the BVOCs measured at the same time explainede 70% of the reactivity during the night but only 40 % during the day.

Similar to concentrations of the individual MTs- $\alpha$ -Ppinene had the highest-contributed mostion to the OH, O3 and NO3 reactivity-reactivities, similar to the concentrations of the individual MTs (FigureFig. S8S8). However, limonene and

30 terpinolene both have relatively fast rate coefficients with the OH radicals and O2 therefore, despite being present at lower concentrations, they can greatly impact the formation of secondary products. the importance of limonene and especially terpinolene for the local chemistry was clearly higher than their contribution to the concentrations. In addition, limonene has shows a higher SOA yield than MTs generally (Lee et al. 2006) and, therefore, it wasis expected to be more important for SOA production than the concentrations indicate. Of the individual SQTs β-caryophyllene had played a major role in

20

contributingcontribution to OH reactivity, while and for the  $O_3$  reactivity it had the highest contributed the most (>60 % in June-August) contribution of all the VOCs measured <del>VOCs</del> (FigureFig. \$858).

**3.3.2. Oxidation products and secondary organic aerosols SOA**

Oxidation of VOCs, under various environmental conditions, produces a variety of gas- and particle phase products that are relevant for atmospheric chemistry and SOA production. To describe this we calculated production rates of oxidation products (OxPRs) from the isoprene, MT, SQT and OVOC reactions as described in section 2.5.

- 10 More oxidation products were produced from SQTs than from MTs (FigureFig. 8d). The contribution of OVOCs, aromatic hydrocarbons and isoprene was very low. SQTs were very important especially during summer nights (FigureFig. 8 and 9). In Thedaytime contributions of MTs and SQTs were equal-similar during all other months except in July when the SQTs were predominateding even in the middle of themid-day. In addition, photo-oxidation of SQTs in smog chamber experiments has been shown to generally resulted in a-much greater aerosol yields than MTs (Hoffmann et al., 1996; Griffin et al., 1999, Lee
- 15 et al. 2006) and therefore they are were expected to have a strongly influence on SOA production. However, these production rates OxPRs described very local situations and even though the rapidfast reactions of SQTs showed very have very strong local effects also MTs also reacted relatively fast rapidly producing secondary products in on a regional scale. The gGlobal emissions of SQTs have been estimated tomay be about 20 % of the MT emissions (Guenther et al. 2012), but this is probably a low-end estimate, since evidence for additional unaccounted SQTs and their oxidation products clearly exists (Yee et al. 2018).
- 20

5

Often-α-pinene is often used as a proxy for all BVOCs, but as shown in FigureFig. 9, the contribution of α-pinene to the total of-oxidation reactions was relatively low (enapp. 20 %). The most important individual reaction producing-generating oxidation products at the site was the reaction of β-caryophyllene with O3. For SQTs the contributions of the OH and NO3 reactions were very low1 especially during the summer months (<2 %). FAlso for MTs, the O3 reactions were also the most important, while the OH radicals had-contributed about 30 %-contribution during summer days and the NO3 reactions were important atim nighttime. Peräkylä et al. (2014) stated that for MTs, nighttime oxidation is dominated by the NO3 radicals

- whereas daytime oxidation is dominated by the O3. However, like as in our study, O3 was also predominateding also during the summer nights. If we take into account for that also emissions and concentrations also beingare highest during the summer, ozoneO3 becomes the most important oxidant for the OxPR production of oxidation products of MTs at night as well. For
- 30 SQTs2 O3 oxidation is clearly dominatesing the first step of the reactions. However, the reaction products of MTs and SQTs, that have lost all their double bonds, continued to react with OH and NO3 and their total contribution is was expected to be higher. It has been suggested that during At nighttime reaction products of MTs may build-up in the atmosphere and are oxidized oxidised after sunrise with OH radicals2 promoting particle growth (Peräkylä et al. 2014). Our results suggest that this also applies also for SQTs.

**6.4. Conclusion**

5

10

We have measured an exceptionally large dataset of VOCs in boreal forest, including terpenoid compounds (isoprene, MTs, SQTs), aldehydes, alcohols and organic acids during 26 months in over a 3three years period. The measurements revealed that of the terpenoids, MTs had showed the highest concentrations at the site, but we were also able to measure also highly reactive SQTs, such as  $\beta$ -caryophyllene and other SQTs in the ambient air due, to the availability of an instrument with improved sensitivity. Our results indicate that in addition to terpenoids, also most of the VOAs, aldehydes and alcohols have a biogenic origins either from direct emissions or by production from the other BVOCs in the air-atmosphere through oxidation reactions.

Temperature was the major factor controlling the concentrations of BVOCs in the air of a boreal forest. Both monthly and daily mean concentrations of MTs had showed very strong exponential correlation with temperature (R2monthly=0.92 and R2daily
 =\_0.88). The SQT concentrations were even more strongly correlated with temperature and had showed higher temperature sensitivity than the MTss, eEspecially monthly mean concentrations in 2016 were highly correlated with temperature (R2= 0.97). The rResults also indicate that in spring and even more in autumn, also other sources (e.g. needle and leaf litter) other than temperature-dependent emissions from the main local trees have highgreatly impact on-MT and SQT concentrations.

- 20 The temperature sensitivities of the most abundant terpenoids, aldehydes and VOAs within the same class of compounds were dependent on vapor pressures. This knowledge can be used to characterize the missing reactivity found in forests during total reactivity studies (Yang et al., 2016) and to help withaid in identification of the masses in direct mass spectrometric measurements of BVOCs and their reaction products.
- 25 We also evaluated the eaffect of that effect affect different BVOCs have have on the local atmospheric chemistry, and although the MTs dominated the OH and NO3 radical chemistry, Due to the very rapid rate coefficient of O3 with SQTs, a relatively small concentration (50 times lower than MTs) of SQTs can greatly impact O3 deposition.SQTs had a major impact on ozone chemistry, even though SQT concentrations are 30 times lower than MTs. The SQTs were also generated producing more oxidation
- 30 products than the MTs. Since the products of SQTs are also less volatile than the MT oxidation products, and SQTs wereare expected to have even higher impact on local SOA production. Both MT and SQT oxidation was dominated by ozoneQ3 especially during summer. Oxidation of other VOC groups had showed very minor contributions to the formation of oxidation products at the site. Our results clearly indicate that SQTs have tomust be considered in local SOA studies for example.g. in when interpreting the results from direct mass spectrometric measurements or modelling SOA formation and growth.

**Acknowledgements**

The research was supported by the Academy of Finland via the Academy research Research fellow Fellow project (Academy

5 of Finland, project 275608) and the Center of Excellence in Atmospheric Sciences (grant no. 307331). The dData providers of SmartSmear AVAA portal are gratefully acknowledged.

| Formatted: No underline, English (United States) |
|--------------------------------------------------|
| Formatted: English (United States)               |
| Formatted: No underline, English (United States) |
| Formatted: English (United States)               |
| Formatted: English (United States)               |
| Formatted: English (United States)               |
| Formatted: No underline                          |
| Formatted: No underline                          |
| Formatted: English (United States)               |
| Formatted: English (United States)               |
| Formatted: No underline                          |
| Formatted: English (United States)               |
| Formatted: English (United Kingdom)              |

Hoffmann, T., Odum, J., Bowman, F., Collins, D., Klockow, D., Flagan, R. C., and Seinfeld, J. H.: Aerosol formation potential of biogenic hydrocarbons, J. Aerosol Sci., 27, S233–S234, 1996.

Holopainen, J.: Multiple functions of inducible plant volatiles. Trends in Plant Science, 9, 529-533, 2004.

Ilvesniemi, H., Levula, J., Ojansuu, R., Kolari, P., Kulmala, L., Pumpanen, J., Launiainen, S., Vesala, T., and Nikinmaa, E.:

- 5 Long-term measurements of the carbon balance of a boreal Scots pine dominated forest ecosystem, Boreal Environ. Res, 14, 731-753, 2009.
  - Jardine, K., Yanez Serrano, A., Arneth, A., Abrell, L., Jardine, A., van Haren, J., Artaxo P., Rizzo, L.V., Ishilda, F.Y., Karl, T., Kesselmeier, J., Saleska, S. and Huxman, T.: Within-canopy sesquiterpene ozonolysis in Amazonia, J. Geophys. Res., 116, D19301, doi:10.1029/2011JD016243, 2011.
- 10 Jardine, A. B., Jardine, K. J., Fuentes, J. D., Martin, S. T., Martins, G., Durgante, F., Carneiro, V., Higuchi, N., Manzi, A. O. and Chambers, J. Q.: Highly reactive light-dependent monoterpenes in the Amazon. Geophys. Res. Lett., 42: 1576–1583. doi: 10.1002/2014GL062573, 2015.
  - Jones, C. E., Hopkins, J. R., and Lewis, A. C.: In situ measurements of isoprene and monoterpenes within a south-east Asian tropical rainforest, Atmos. Chem. Phys., 11, 6971-6984, https://doi.org/10.5194/acp-11-6971-2011, 2011.
- 15 Junninen, H., Lauri, A., Keronen, P., Aalto, P., Hiltunen, V., Hari, P., and Kulmala, M.: Smart-SMEAR: on-line data exploration and visualization tool for SMEAR stations, Boreal Environ. Res., 14, 447–457, 2009.

Kaminski, M., Fuchs, H., Acir, I. H., Bohn, B., Brauers, T., Dorn, H. P., Häseler, R., Hofzumahaus, A., Li, X., Lutz, A., Nehr,
 S., Rohrer, F., Tillmann, R., Vereecken, L., Wegener, R., and Wahner, A.: Investigation of the \_-pinene photooxidation by
 OH in the atmosphere simulation chamber SAPHIR, Atmos. Chem. Phys., 17, 6631-6650, doi:10.5194/acp-17-6631-2017,
 2017.

- 20
  - Kim, S., Karl, T., Helmig, D., Daly, R., Rasmussen, R., and Guenther, A.: Measurement of atmospheric sesquiterpenes by proton transfer reaction-mass spectrometry (PTR-MS), Atmos. Meas. Tech., 2, 99-112, https://doi.org/10.5194/amt-2-99-2009, 2009.
- 25 Kim, S., Karl, T., Guenther, A., Tyndall, G., Orlando, J., Harley, P., Rasmussen, R., and Apel, E.: Emissions and ambient distributions of Biogenic Volatile Organic Compounds (BVOC) in a ponderosa pine ecosystem: interpretation of PTR-MS mass spectra, Atmos. Chem. Phys., 10, 1759-1771, https://doi.org/10.5194/acp-10-1759-2010, 2010.
  - Kesselmeier, J., U. Kuhn, S. Rottenberger, T. Biesenthal, A. Wolf, G. Schebeske, M. O. Andreae, P. Ciccioli, E. Brancaleoni, M. Frattoni, S. T. Oliva, M. L. Botelho, C. M. A. Silva, and T. M. Tavares, Concentrations and species composition of
- 30 atmospheric volatile organic compounds (VOCs) as observed during the wet and dry season in Rondo^nia (Amazonia), J. Geophys. Res., 107(D20), 8053, doi:10.1029/2000JD000267, 2002
  - Kontkanen, J., Paasonen, P., Aalto, J., Bäck, J., Rantala, P., Petäjä, T., and Kulmala, M.: Simple proxies for estimating the concentrations of monoterpenes and their oxidation products at a boreal forest site, Atmos. Chem. Phys., 16, 13291-13307, https://doi.org/10.5194/acp-16-13291-2016, 2016.

- Kulmala, M., Kontkanen, J., Junninen, H., Lehtipalo, K., Manninen, H. E., Nieminen, T., Petaja, T., Sipila, M., Schobesberger,
  S., Rantala, P., Franchin, A., Jokinen, T., Jarvinen, E., Aijala, M., Kangasluoma, J., Hakala, J., Aalto, P. P., Paasonen, P.,
  Mikkila, J., Vanhanen, J., Aalto, J., Hakola, H., Makkonen, U., Ruuskanen, T., Mauldin, R. L., Duplissy, J., Vehkamaki,
  H., Back, J., Kortelainen, A., Riipinen, I., Kurten, T., Johnston, M. V., Smith, J. N., Ehn, M., Mentel, T. F., Lehtinen, K.
- 5 E. J., Laaksonen, A., Kerminen, V. M., and Worsnop, D. R.: Direct observations of atmospheric aerosol nucleation, Science, 339, 943–946, doi:10.1126/science.1227385, 2013.
  - Kännaste, A., Vongvanich, N., Borg-Karlson, A.-L.: Infestation by a Nalepella species induces emissions of a and bfarnesenes, (-)-linalool and aromatic compounds in Norway spruce clones of different susceptibility to the large pine weevil. Arthropod-Plant Interactions, 31–41, 2008.
- 10 Kännaste, A., Nordenhem, H., Nordlander, G., and Borg-Karlson, A.-K.: Volatiles from a Mite-Infested Spruce Clone and their Effects on PineWeevil Behavior, J. Chem. Ecol., 35, 1262–1271, 2009.
  - Lappalainen, H. K., Sevanto, S., Bäck, J., Ruuskanen, T. M., Kolari, P., Taipale, R., Rinne, J., Kulmala, M., and Hari, P.: Daytime concentrations of biogenic volatile organic compounds in a boreal forest canopy and their relation to environmental and biological factors, Atmos. Chem. Phys., 9, 5447-5459, https://doi.org/10.5194/acp-9-5447-2009, 2009.
- 15
- Lee, A., A. H. Goldstein, J. H. Kroll, N. L. Ng, V. Varutbangkul, R. C. Flagan, and J. H. Seinfeld: Gas-phase products and secondary aerosol yields from the photooxidation of 16 different terpenes, J. Geophys. Res., 111, D17305, doi:10.1029/2006JD007050, 2006.
- Liao, L., Dal Maso, M., Taipale, R., Rinne, J., Ehn, M., Junninen, H., Äijälä, M., Nieminen, T., Alekseychik, P., Hulkkonen,
- 20 M., Worsnop, D. R., Kerminen, V.-M. & Kulmala, M.: Monoterpene pollution episodes in a forest environment: indication of anthropogenic origin and association with aerosol particles, Boreal Environ. Res., 4, 2011. Liebmann, J., Karu, E., Sobanski, N., Schuladen, J., Ehn, M., Schallhart, S., Quéléver, L., Hellen, H., Hakola, H., Hoffmann,
  - Liebmann, J., Karu, E., Sobanski, N., Schuladen, J., Enn, M., Schalhart, S., Quelever, L., Hellen, H., Hakola, H., Hormann, T., Williams, J., Fischer, H., Lelieveld, J., and Crowley, J. N.: Direct measurement of NO3 radical reactivity in a boreal forest, Atmos. Chem. Phys., 18, 3799-3815, https://doi.org/10.5194/acp-18-3799-2018, 2018.
- 25 Manninen, A. J., O'Connor, E. J., Vakkari, V. and Petäjä, T.: A generalised background correction algorithm for a Halo Doppler lidar and its application to data from Finland, Atmos. Meas. Tech., 9(2), 817–827, doi:10.5194/amt-9-817-2016, 2016.
  - Mellouki, A., Wallington, T. J., and Chen, J.: Atmospheric chemistry of oxygenated volatile organic compounds: Impacts on air quality and climate, Chem. Rev., 115, 3984–4014, doi:10.1021/cr500549n, 2015.
- 30 Mogensen, D., Smolander, S., Sogachev, A., Zhou, L., Sinha, V., Guenther, A., Williams, J., Nieminen, T., Kajos, M. K., Rinne, J., Kulmala, M., and Boy, M.: Modelling atmospheric OH-reactivity in a boreal forest ecosystem, Atmos. Chem. Phys., 11, 9709-9719, doi:10.5194/acp-11-9709-2011, 2011.
  - Mäki, M., Heinonsalo, J., Hellén, H., and Bäck, J.: Contribution of understorey vegetation and soil processes to boreal forest isoprenoid exchange, Biogeosciences, 14, 1055-1073, https://doi.org/10.5194/bg-14-1055-2017, 2017.

|   |    | Estimating the Turbulent Kinetic Energy Dissipation Rate from a Vertically Pointing Doppler Lidar, and Independent              |                                     |
|---|----|---------------------------------------------------------------------------------------------------------------------------------|-------------------------------------|
|   |    | Evaluation from Balloon-Borne In Situ Measurements, J. Atmos. Oceanic Technol., 27(10), 1652 1664,                              |                                     |
|   |    | doi:10.1175/2010JTECHA1455.1, 2010. Ng, N. L., Brown, S. S., Archibald, A. T., Atlas, E., Cohen, R. C., Crowley, J. N.,         |                                     |
|   | 5  | Day, D. A., Donahue, N. M., Fry, J. L., Fuchs, H., Griffin, R. J., Guzman, M. I., Herrmann, H., Hodzic, A., Iinuma, Y.,         |                                     |
|   |    | Jimenez, J. L., Kiendler-Scharr, A., Lee, B. H., Luecken, D. J., Mao, J., McLaren, R., Mutzel, A., Osthoff, H. D., Ouyang,      |                                     |
|   |    | B., Picquet-Varrault, B., Platt, U., Pye, H. O. T., Rudich, Y., Schwantes, R. H., Shiraiwa, M., Stutz, J., Thornton, J. A.,     |                                     |
|   |    | Tilgner, A., Williams, B. J., and Zaveri, R. A.: Nitrate radicals and biogenic volatile organic compounds: oxidation,           |                                     |
|   |    | mechanisms, and organic aerosol, Atmos. Chem. Phys., 17, 2103-2162, https://doi.org/10.5194/acp-17-2103-2017, 2017.             |                                     |
|   | 10 | Nölscher, A. C., Williams, J., Sinha, V., Custer, T., Song, W., Johnson, A. M., Axinte, R., Bozem, H., Fischer, H., Pouvesle,   |                                     |
|   |    | N., Phillips, G., Crowley, J. N., Rantala, P., Rinne, J., Kulmala, M., Gonzales, D., Valverde-Canossa, J., Vogel, A.,           | Formatted: English (United Kingdom) |
|   |    | Hoffmann, T., Ouwersloot, H. G., Vilà-Guerau de Arellano, J., and Lelieveld, J.: Summertime total OH reactivity                 |                                     |
|   |    | measurements from boreal forest during HUMPPA-COPEC 2010, Atmos. Chem. Phys., 12, 8257-8270, doi:10.5194/acp-                   |                                     |
|   |    | 12-8257-2012, 2012.                                                                                                      |                                     |
|   | 15 | O'Connor, E. J., Illingworth, A. J., Brooks, I. M., Westbrook, C. D., Hogan, R. J., Davies, F. and Brooks, B. J.: A Method for  |                                     |
|   |    | Estimating the Turbulent Kinetic Energy Dissipation Rate from a Vertically Pointing Doppler Lidar, and Independent              |                                     |
|   |    | Evaluation from Balloon-Borne In Situ Measurements, J. Atmos. Oceanic Technol., 27(10), 1652-1664,                              |                                     |
|   |    | doi:10.1175/2010JTECHA1455.1, 2010.                                                                                             |                                     |
|   |    | A                                                                                                                               | Formatted: English (United Kingdom) |
| Ļ | 20 | Paulson, S. E., R. C. Flagan, and J. H. Seinfeld: Atmospheric photooxidation of isoprene, 2, The ozone-isoprene reaction, Int.  |                                     |
|   |    | J. Chem. Kinet., 24, 103-125, 1992.                                                                                             |                                     |
|   |    | Pearson, G., Davies, F. and Collier, C.: An Analysis of the Performance of the UFAM Pulsed Doppler Lidar for Observing the      |                                     |
| l |    | Boundary Layer, J. Atmos. Oceanic Technol., 26(2), 240-250, doi:10.1175/2008JTECHA1128.1, 2009.                                 |                                     |
|   |    | Peräkylä, O., Vogt, M., Tikkanen, OP., Laurila, T., Kajos, M. K., Rantala, P. A., Patokoski, J., Aalto, J., Yli-Juuti, T., Ehn, | Formatted: English (United Kingdom) |
|   | 25 | M., Sipilä, M., Paasonen, P., Rissanen, M., Nieminen, T., Taipale, R., Keronen, P., Lappalainen, H. K., Ruuskanen, T. M.,       |                                     |
|   |    | Rinne, J., Kerminen, VM., Kulmala, M., Bäck, J., and Petäjä, T.: Monoterpenes' oxidation capacity and rate over a boreal        |                                     |
|   |    | forest: temporal variation and connection to growth of newly formed particles, Boreal Environ. Res., 19, 293-310, 2014.         |                                     |
|   |    | Petterson, M.: Stress related emissions of Norway spruce plants, Licentiate thesis, KTH Royal Institute of technology,          |                                     |
| l |    | Stockholm, ISBN-13: 978-91-7178-644-9, 2007.                                                                                    |                                     |
|   | 30 | Petäjä, T., Mauldin, R. L., Kosciuch, E., McGrath, J., Nieminen, T., Paasonen, P., Boy, M., Adamov, A., Kotiaho, T., and        | Formatted: English (United States)  |
|   |    | Kulmala, M.: Sulfuric acid and OH concentrations in a boreal forest site, Atmospheric. Chemistry, and Physics, 9, 7435-         |                                     |
|   |    | 448, doi: 10.5194/acp-9-7435-2009, 2009.                                                                                        |                                     |
|   |    | Possanzini, M., Di Palo, V., Brancaleoni, E., Frattoni, M. and Ciccioli, P.: A train of carbon and DNPH-coated catridges for    | Formatted: English (United States)  |
|   |    |                                                                                                                                 |                                     |

O'Connor, E. J., Illingworth, A. J., Brooks, I. M., Westbrook, C. D., Hogan, R. J., Davies, F. and Brooks, B. J.: A Method for

| Praplan, A. P., Pfannerstill, E. Y., Williams, J. and Hellén, H.: OH reactivity of the urban air in Helsinki, Finland, durin | 3 |
|------------------------------------------------------------------------------------------------------------------------------|---|
| winter. Atmos. Environ., 169, 150 - 161, doi: 10.1016/j.atmosenv.2017.09.013, 2017.                                          |   |

Praplan A. P., Tykkä T., Boy M. and Hellén H.: Long-term total OH reactivity measurements in a boreal forest. Manuscript in preparation, 2018.

5

15

Rannik, Ü., Altimir, N., Mammarella, I., Bäck, J., Rinne, J., Ruuskanen, T. M., Hari, P., Vesala, T., and Kulmala, M.: Ozone deposition into a boreal forest over a decade of observations: evaluating deposition partitioning and driving variables, Atmos. Chem. Phys., 12, 12165-12182, https://doi.org/10.5194/acp-12-12165-2012, 2012.

Rivera-Rios, J. C., Nguyen, T. B., Crounse, J. D., Jud, W., St. Clair, J. M., Mikoviny, T., Gilman, J. B., Lerner, B. M.,

10 Kaiser, J. B., de Gouw, J., Wisthaler, A., Hansel, A., Wennberg, P. O., Seinfeld, J. H. and Keutsch, F. N.. Conversion of hydroperoxides to carbonyls in field and laboratory instrumentation: Observational bias in diagnosing pristine versus anthropogenically controlled atmospheric chemistry, Geophys. Res. Lett., 41, 8645–8651, doi:10.1002/2014GL061919, 2014.

Rohrer, F., and Berresheim, H.: Strong correlation between levels of tropospheric hydroxyl radicals and solar ultraviolet radiation, Nature, 442, 184-187,doi:10.1038/nature04924, 2006

Scala, A., Allmann, S., Mirabella, R., Haring, M.A. and Schuurink, R.C.: Green leaf volatiles: A Plant's multifunctional weapon against herbivores and pathogens. Int. J. Mol. Sci., International Journal of Molecular Sciences, 14, 17781-17811, doi:10.3390/ijms140917781, 2013.

Seco, R., Penuelas, J. and Filella, I.: Short-chain oxygenated VOCs: Emission and uptake by plants and atmospheric sources,
 sinks, and concentrations. Atmos. Environ., 41, 2477-2499, 2007.

Schallhart, S., Rantala, P., Kajos, M. K., Aalto, J., Mammarella, I., Ruuskanen, T. M., and Kulmala, M.: Temporal variation of VOC fluxes measured with PTR-TOF above a boreal forest, Atmos. Chem. Phys., 18, 815-832, https://doi.org/10.5194/acp-18-815-2018, 2018.

Sindelarova, K., Granier, C., Bouarar, I., Guenther, A., Tilmes, S., Stavrakou, T., Müller, J.-F., Kuhn, U., Stefani, P., and Knorr, W.: Global data set of biogenic VOC emissions calculated by the MEGAN model over the last 30 years, Atmos.

- Chem. Phys., 14, 9317-9341, https://doi.org/10.5194/acp-14-9317-2014, 2014.
   Sinha, V., Williams, J., Lelieveld, J., Ruuskanen, T., Kajos, M., Patokoski, J., Hellen, H., Hakola, H., Mogensen, D., Boy, M., Rinne, J., and Kulmala, M.: OH Reactivity Measurements within a Boreal Forest: Evidence for Unknown Reactive Emissions, Environ. Sci. Technol., 44, 6614–6620, doi:10.1021/es101780b, 2010.
- 30 Tarvainen, V., Hakola, H., Hellén, H., Bäck, J., Har, P., Kulmala, M.: Temperature and light dependence of the VOC emissions of Scots pine. Atmos. Chem. Phys., 5, 6691-6718, 2005.

Tunved, P., Hansson, H.-C., Kerminen, V.-M., Ström, J., Dal Maso, M., Lihavainen, H., Viisanen, Y., Aalto, P. P., Komppula, Formatted: English (United Kingdom) M., and Kulmala, M.: High natural aerosol loading over boreal forests, Science, 312, 261–263, 2006.

[revised manuscript text omitted]